# Theoretical Analysis of Weak-to-Strong Generalization

**Hunter Lang**
MIT CSAIL

**David Sontag**
MIT CSAIL

**Aravindan Vijayaraghavan**
Northwestern University

## Abstract

Strong student models can learn from weaker teachers: when trained on the predictions of a weaker model, a strong pretrained student can learn to correct the weak model's errors and generalize to examples where the teacher is not confident, even when these examples are excluded from training. This enables learning from cheap, incomplete, and possibly incorrect label information, such as coarse logical rules or the generations of a language model. We show that existing weak supervision theory fails to account for both of these effects, which we call *pseudolabel correction* and *coverage expansion*, respectively. We give a new bound based on *expansion* properties of the data distribution and student hypothesis class that directly accounts for pseudolabel correction and coverage expansion. Our bounds capture the intuition that weak-to-strong generalization occurs when the strong model is unable to fit the mistakes of the weak teacher without incurring additional error. We show that these expansion properties can be checked from finite data and give empirical evidence that they hold in practice.

## 1 Introduction

Weakly-supervised learning allows practitioners to train models with possibly-incorrect, easy-to-obtain *pseudo*labels instead of accurate and expensive ground-truth labels. For example, suppose the goal is to classify documents based on whether they have positive or negative sentiment. Instead of employing humans to label examples $x_i$ with positive/negative sentiment labels $y_i$, weak supervision enables models to learn from simple rules, such as: if 'incredible' $\in x_i$, sentiment = positive. Called *programmatic weak supervision*, a wealth of literature has shown how to aggregate rules, often called "labeling functions", into individual pseudolabels that can be used to train a model [e.g., 51, 49, 63, 21]. In typical pipelines, this consists of fine-tuning a pre-trained neural network [71]. This method has met with a huge amount of empirical success in natural language processing [51, 43, 28, 36, 70] computer vision [74, 62], and verticals such as healthcare [25, 26, 20, 67].

Another emerging trend in weak supervision is to use the zero-shot or few-shot outputs of a large language model (LLM) as pseudolabels for training another language model—the student model often outperforms its noisy "teacher", and this technique even works when the teacher is a less powerful model than the student, distinguishing it from classical knowledge distillation [65, 17, 18, 35, 2, 24, 33, 66, 39, 10]. Good data selection can be critical, and several approaches carefully choose a "confident" subset of the pseudolabels [e.g. 35, 18, 39, 66]. The increasing prevalence of LLM use in crowdwork [64] highlights the importance of better understanding this learning method.

In both cases, the pseudolabels $\tilde{y}$ are given by the *pseudolabeler* or *teacher model*, which is some function of the input example $x$. The pseudolabeler may make errors ($\tilde{y}(x) \neq y(x)$), and it may not *cover* every point—there are some points where the teacher abstains from providing a weak label. In the examples above, these are points not covered by the rules and points where the teacher LLM is not confident, respectively. *A priori*, it seems that a powerful enough classifier should exactly fit the pseudolabeler on the covered data and have trivial performance on the uncovered data. However, this is not what happens in practice—the empirical success of weak supervision is due to two surprising, related phenomena that together comprise *weak-to-strong generalization:* (a) *Pseudolabel correction:*

The performance of the model exceeds the performance of the pseudolabels used to train it; and (b) *Coverage expansion:* The model performs well even on the portion of example space $\mathcal{X}$ that is not covered by pseudolabels. These empirical outcomes are key to the success of weak supervision.

Surprisingly, the existing theoretical literature on programmatic weak supervision does not address either phenomenon—the majority of weak supervision theory literature either focuses on how to adeptly combine the outputs of multiple "weak rules" into a single pseudolabel $\tilde{y}_i$, [51, 49, 50, 21], or treats learning from weak supervision as learning from noisy labels [e.g. 46, 59, 15], but as we discuss in Section 2, this framing does not capture the setting we consider, where the pseudolabels used to train one model are the outputs of another model.

In this work, we give a theoretical analysis of weakly-supervised learning that provably accounts for the effects of pseudolabel correction and coverage expansion. Our results use a natural *expansion* condition on the population data distribution. Informally, expansion implies that "bad" points (points with incorrect pseudolabels, or points with no pseudolabel at all) have many "good" neighbors (points with correct pseudolabels). If the learned student model is relatively robust on the neighborhoods of interest, then making a mistake on a bad point means making many mistakes on good points as well. This allows us to prove a relationship between the student model's error on the weak labels (the training objective) and the student model's error on the true labels (the desired objective). Our assumptions and bounds in Section 4 formalize this intuition. Section 5 details a procedure for checking our expansion conditions from finite data, and in Section 6 and Appendix E, we give empirical evidence that these conditions hold on real data.

To the best of our knowledge, our results provide the first error bounds for programmatic weak supervision with realistic assumptions. We show that our bounds generalize and connect several existing results from the co-training, self-training, and distribution shift literature [e.g., 6, 3, 69, 12] and adapt them to the weak supervision setting. For example, we show in Appendix C.1 that Theorem 4.2 generalizes the co-training results of Blum and Mitchell [6]. We discuss these generalizations in detail in Sections 2 and 4 and Appendix C. Our result in Section 5 is the first among these works to prove that the expansion assumptions can, in principle, be checked using finite data. Unlike most prior work in this space, our experiments in Section 6 attempt to check whether the expansion assumptions hold in practice. While our experiments are limited in scope, our attempt to systematically check expansion in a practical scenario is a major departure from previous work.

Finally, prior work with expansion assumptions similar to ours (in the co-training [6, 3], self-training [69], and distribution shift [12] literature) requires that the classifiers are either perfectly robust [6, 3] or adversarially robust [69, 12] for their bounds to apply. Empirical results suggest that adversarial training has fairly limited value for improving coverage expansion and pseudolabel correction [38], and that these two effects still occur for student models that are not adversarially trained [71, 10]. To close this gap, we make a connection to the literature on *robust expansion* [27, 34, 42] and prove error bounds for student models that are merely "robust on average." Unlike prior work, these bounds allow for the presence of adversarial examples for every input point.

## 2   Related Work

Ratner et al. [51, 49, 50], Fu et al. [21] focus on how to combine the outputs of multiple "weak rules" into a single pseudolabel $\tilde{y}(x)$ for each covered example, a problem with a long history in the crowdsourcing literature [e.g. 16, 30, 29]. However, empirical results indicate that this is not the important aspect of weak supervision: most methods for combining weak rules fail to significantly outperform majority vote once the final classifier is trained [72]. Works in this literature that *do* provide error bounds for the student (e.g., Fu et al. [21], Ratner et al. [51]) either fail to capture weak-to-strong generalization effects, as shown in Section 3, or make difficult-to-justify assumptions—for example, Ratner et al. [51] assumes that $y(\mathbf{x})$ and $\mathbf{x}$ are conditionally independent given $\tilde{y}(\mathbf{x})$. If this were true, there would be no gain from training a classifier, since $\tilde{y}(\mathbf{x})$ already captures all the information that $\mathbf{x}$ contains about $y(\mathbf{x})$. Work that treats learning from weak supervision as a noisy label learning problem [e.g. 45, 46, 59, 15] does not capture the types of weak supervision we consider. When the supervision comes from weaker model (be it rule-based or a weaker LLM), there is no exogenous noise process that corrupts the training labels. There are simply some points that deterministically get the wrong labels and some points with no label. This rules out common noise models like class-conditional noise [46] and Tsybakov noise [60], and is arguably not appropriate to model as instance-dependent noise [59, 15], since for each $x$ the noise is deterministically 0 or 1.

Burns et al. [10] conducted a large empirical study showing widespread weak-to-strong generalization effects when training a strong language model on the generations of a weaker model. Our results are a step toward a theoretical understanding of these effects. Several works have used expansion to give provable guarantees in other settings where models are learning from each other. Balcan et al. [3] use expansion to analyze co-trained [6] classifiers. Our expansion assumption is similar to their "left-right" expansion, but we generalize beyond the multi-view setup of co-training and account for error propagation. Wei et al. [69] give provable guarantees for self-training under expansion assumptions similar to ours. We provide a different pseudolabel correction bound, tighter guarantees for coverage expansion under weaker assumptions, and generalize both results to classifiers that are not adversarially robust. Cai et al. [12] use expansion to prove general guarantees for pseudolabel correction, semi-supervised learning, and unsupervised domain adaptation, but their results require the student to be very adversarially robust. Compared to all these expansion works, we also outline a rigorous theoretical framework for checking expansion on finite data and provide more empirical evidence for our assumptions. We discuss more related work in Appendix A.

## 3 Setup and Shortcomings of Existing Bounds

**Notation.** $\mathbf{x}$ refers to a random variable with distribution $\mathcal{D}$ and italicized letters $\boldsymbol{x}$ refer to realizations of $\mathbf{x}$. We will assume for ease of exposition that the input space $\mathcal{X}$ is a discrete[1] (but possibly very large) set, such as all vectors in $\mathbb{R}^d$ up to a fixed numerical precision. For $A \subset \mathcal{X}$ we use $\bar{A} = \mathcal{X} \setminus A$. We assume there is a ground-truth function of interest, $y : \mathcal{X} \to \mathcal{Y} = \{1, \ldots, k\}$, and a *pseudolabeler* $\tilde{y} : \mathcal{X} \to \mathcal{Y} \cup \{\varnothing\}$, which assigns to each point $\boldsymbol{x}$ either a label in $\mathcal{Y}$ or the special "abstention" symbol $\varnothing$. The function $\tilde{y}$ can also be thought of as the *teacher model*, but we are primarily concerned with instances where the teacher is much less capable than the "student" it will be used to train.

Define $S = \{\boldsymbol{x}|\tilde{y}(\boldsymbol{x}) \neq \varnothing\}$ to be the *covered* subset of $\mathcal{X}$, i.e., the subset of $\mathcal{X}$ that has a pseudolabel, and let $T = \{\boldsymbol{x}|\tilde{y}(\boldsymbol{x}) = \varnothing\} = \mathcal{X} \setminus S$ be the uncovered set. This notation serves to emphasize that training occurs on a (pseudolabeled) *source* subset $S$, and then evaluation occurs on the union of $S$ and the (uncovered) *target* $T$. Let $\{\mathcal{X}_i\}$ be a partition of $\mathcal{X}$ such that within each $\mathcal{X}_i$, the ground-truth label is constant. For example, we could set $\mathcal{X}_i = \{\boldsymbol{x}|y(\boldsymbol{x}) = i\}$ to be the set of points with ground-truth label $i$. We will use this definition of $\mathcal{X}_i$ for convenience, but all our results hold for more general partitions. Each of $S$ and $T$ can further be partitioned as $S_i = S \cap \mathcal{X}_i$, $T_i = T \cap \mathcal{X}_i$. Finally, each $S_i$ can be further partitioned into the correctly-pseudolabeled examples $S_i^{good} = \{\boldsymbol{x} \in S_i|\tilde{y}(\boldsymbol{x}) = y(\boldsymbol{x})\}$ and the incorrectly-pseudolabeled examples $S_i^{bad} = S_i \setminus S_i^{good}$. Let $\alpha_i := \mathbb{P}(S_i^{bad}|S_i)$ be the error rate of $\tilde{y}$ on $S_i$. We assume $0 < \alpha_i < \frac{1}{2}$ for all $i$.

**Problem Setup.** For two classifiers $f, g : \mathcal{X} \to \mathcal{Y}$ and a set $U \subset \mathcal{X}$, we use $\text{err}(f, g|U)$ to represent $\mathbb{P}(f(\mathbf{x}) \neq g(\mathbf{x})|\mathbf{x} \in U)$, their probability of disagreement conditioned on $\mathbf{x}$ falling in the set $U$. Here the probability is over $\mathbf{x} \sim \mathcal{D}$; this will often be omitted for notational convenience. We will be particularly interested in classifiers obtained by minimizing the error on the non-abstaining weak labels over the strong model hypothesis class $\mathcal{F}$, i.e., (approximate) solutions to $\arg\min_{f \in \mathcal{F}} \text{err}(f, \tilde{y}|S)$. The ultimate goal is to obtain upper bounds on the error $\text{err}(f, y|\mathcal{X})$ for such classifiers. That is, we want to upper bound the error of a classifier $f$ on the *true labels* over the *entire* input space $\mathcal{X}$.

There are two key challenges. First, the classifier is trained using $\tilde{y}$, not $y$, and $\tilde{y}$ may have arbitrary errors that are not captured by any well-studied noise model such as class-conditional or Tsybakov noise [60]—we've assumed $\tilde{y}$ and the true labels $y$ are both deterministic functions of the input, so there is no exogenous noise process that corrupts the training labels.

Second, we care about the performance of $f$ on the entire space $\mathcal{X}$, but we only train on the *covered* samples from $S \subset \mathcal{X}$. Again, since $\tilde{y}$ is an arbitrary deterministic function, our samples from $S$ are not distributed according to $\mathcal{D}$, and $S$ and $T$ have no overlap, ruling out approaches like importance-weighting. The following example elaborates on the issues at play and illustrates the shortcomings of existing bounds in the weak supervision literature.

### 3.1 Shortcomings of Existing Bounds: Illustrative Example

A special case of weak-to-strong generalization is training a strong pretrained model on the outputs of very coarse rules. Following the example from Section 1, suppose our goal is to obtain a sentiment

---

[1]This is done to simplify some of the arguments, as in HaoChen et al. [22]. As shown there, the results can be generalized to continuous $\mathcal{X}$ with additional regularity conditions. Our bounds have no dependence on $|\mathcal{X}|$.

classifier, so $\mathcal{X}$ is the space of text documents, $\mathcal{Y} = \{-1, 1\}$ and $\tilde{y}$ is given by the following rules: if 'incredible' $\in \boldsymbol{x}$, $\tilde{y}(\boldsymbol{x}) = +1$. If 'horrible' $\in \boldsymbol{x}$, $\tilde{y}(\boldsymbol{x}) = -1$. Otherwise, $\tilde{y}(\boldsymbol{x}) = \varnothing$.

Assume for simplicity that "incredible" and "horrible" never co-occur, so $\tilde{y}$ is well-defined. This example shows that the student model hypothesis class $\mathcal{F}$ and the training procedure both play a vital role in weak-to-strong generalization. Suppose $\mathcal{F}$ is the class of bag-of-words classifiers, and we obtain a student $f$ by minimizing $\text{err}(f, \tilde{y}|S)$. Without modifications to the training procedure (such as L2 regularization), $f$ may place a large positive weight on "incredible", a large negative weight on "horrible", and zero weight on all other tokens. This model has zero error on the weak labels, so it exactly minimizes the training objective. It reproduces the pseudolabels on the covered set and has trivial performance on the uncovered set, so there is no weak-to-strong generalization. On the other hand, if we were to instead train a linear probe on top of a SentenceBERT [52] representation, we would obtain a model that improves over $\tilde{y}$ on $S$ (the *covered* set of documents containing either "horrible" or "incredible") and has reasonable performance on $T$ (the uncovered set). Section 6 contains precise results for this example, but the critical (seemingly obvious) aspect is that the student representation and the training details matter for achieving weak-to-strong generalization.

The following proposition (proven in Appendix B.3) illustrates how existing error bounds in the programmatic weak supervision literature, which do not account for training details and the student hypothesis class, are unable to capture pseudolabel correction and coverage expansion.

**Proposition 3.1.** *Suppose the label marginals for the above example satisfy $\mathbb{P}(\mathrm{y} = y) = \frac{1}{2}$ for $y \in \{-1, 1\}$, and assume that the weak label error rates $\alpha_{-1} = \alpha_1 = \alpha$, and that the weak labels cover each class equally often: $\mathbb{P}(\tilde{y} = \varnothing | \mathrm{y} = y) = \mathbb{P}(\tilde{y} = \varnothing)$. Let $\tilde{f} = \min_{f \in \mathcal{F}} \text{err}(f, \tilde{y}|S)$ be the classifier minimizing the weak label error on the covered set. Then the bound from Fu et al. [21, Theorem 3] simplifies (in our notation) to: $\text{err}(\tilde{f}, y) \leq \mathbb{P}(S) \cdot 4\alpha(1 - \alpha) + \mathbb{P}(T)$.*

The first term accounts for the error of $\tilde{f}$ on the covered set $S$. The weak labels themselves have error $\alpha$ on $S$, but the bound for $\tilde{f}$ is $4\alpha(1 - \alpha) > \alpha$ whenever $\alpha < \frac{3}{4}$, so Fu et al. [21]'s bound does not allow for pseudolabel correction in this example. The second term accounts for the error of $\tilde{f}$ on the uncovered set $T$. A random guess achieves error $\frac{1}{2}$ on $T$, but the bound charges every point in $T$ as an error, so it also does not account for coverage expansion or even the performance of random guessing on $T$.

Expansion assumptions similar to ours have also been studied in the context of self-training [69] and domain adaptation [12]. The following proposition shows that while the results of Wei et al. [69] can be adapted to weakly-supervised learning, our bounds capture the full weak supervision setup better, since Wei et al. [69, Theorem 4.3] was not designed to deal with partial coverage $\mathbb{P}(\mathbf{x} \in S) < 1$, so applying it to weakly-supervised learning still requires fairly large coverage.

**Proposition 3.2** (informal). *Suppose the coverage $\mathbb{P}(\mathbf{x} \in S)$ in the example above is less than $2/3$. Then the bound from Wei et al. [69, Theorem 4.3] does not apply since directly adapting it to the weak supervision setting requires $\mathbb{P}(\mathbf{x} \in S) \geq 2/3$.*

Empirically, coverage expansion and pseudolabel correction can both occur in the low-coverage regime [38]. Finally, as mentioned in Section 1, Wei et al. [69] assumes the classifier is adversarially robust. In contrast, we provide bounds that directly account for coverage expansion, place no restrictions on the amount of weak label coverage, and allow for the presence of many adversarial examples. As the example in this section suggests is necessary, the model hypothesis class and the training details (in particular, the *robustness* of the model) play a central role in our bounds. The following definitions attempt to capture these properties.

### 3.2 Definitions

**Definition 1** (Neighborhood). *Let $\mathcal{N}$ be a* neighborhood function *that maps each point $\boldsymbol{x}$ to a set of points $\mathcal{N}(\boldsymbol{x}) \subset \mathcal{X}$ that we call the neighborhood of $\boldsymbol{x}$. We will assume that $\mathcal{N}$ satisfies $\boldsymbol{x} \in \mathcal{N}(\boldsymbol{x}') \iff \boldsymbol{x}' \in \mathcal{N}(\boldsymbol{x})$, i.e., that the neighborhoods are symmetric. We can extend $\mathcal{N}$ to a function of sets as $\mathcal{N}(A) = \bigcup_{\boldsymbol{x} \in A} \mathcal{N}(\boldsymbol{x})$. Examples to keep in mind are $\mathcal{N}(\boldsymbol{x}) = \{\boldsymbol{x}' : ||\varphi(\boldsymbol{x}) - \varphi(\boldsymbol{x}')|| \leq r\}$ for some representation $\varphi : \mathcal{X} \to \mathbb{R}^d$, or, in the case of text inputs $\boldsymbol{x}$, the set of fluent paraphrases of $\boldsymbol{x}$. However, our results work with any definition of $\mathcal{N}$.*

**Definition 2** ($\eta$-robust). *For an arbitrary classifier $f$ and point $\boldsymbol{x}$, define $r(f, \boldsymbol{x}) = \mathbb{P}(f(\mathbf{x}') \neq f(\boldsymbol{x}) | \mathbf{x}' \in \mathcal{N}(\boldsymbol{x}))$ as the probability $f$ gives different labels to $\boldsymbol{x}$ and a random neighbor $\mathbf{x}'$ of $\boldsymbol{x}$.*

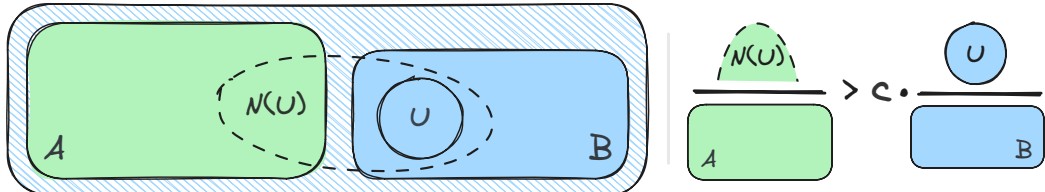

Figure 1: Relative expansion (Definition 3) on the sets $(A, B)$. Expansion requires that certain subsets $U \subset B$ have neighborhoods $\mathcal{N}(U)$ such that $\mathbb{P}(\mathcal{N}(U)|A) \geq c\mathbb{P}(U|B)$. These probabilities are represented graphically on the right-hand-side as the fractions $|\mathcal{N}(U) \cap A|/|A|$ and $|U|/|B|$.

A classifier $f : \mathcal{X} \to \mathcal{Y}$ is said to be $\eta$-robust at a point $\boldsymbol{x}$ if $r(f, \boldsymbol{x}) \leq \eta$. Define $R_\eta(f) = \{\boldsymbol{x} : r(f, \boldsymbol{x}) \leq \eta\}$ as the set of $\eta$-robust points for $f$.

If $\eta = 0$, $f$ is $\eta$-robust at $\boldsymbol{x}$ if and only if $f$ is adversarially robust over $\mathcal{N}(\boldsymbol{x})$, so this definition generalizes adversarial robustness. By Markov's inequality, any classifier $f$ with:

$$\mathbb{E}_{\mathbf{x}\sim\mathcal{D}, \mathbf{x}'\sim\mathcal{D}|\mathcal{N}(\mathbf{x})}[f(\mathbf{x}) \neq f(\mathbf{x}')] \leq \gamma \tag{1}$$

is $\eta$-robust on a set of probability at least $1 - \gamma/\eta$ (see Lemma B.1). The requirement (1) is significantly more natural than adversarial robustness: a classifier satisfies (1) whenever it gives most points the same labels as most of their neighbors. We refer to (1) as "average-case robustness".

**Definition 3** (Expansion). *Fix sets $A, B \subset \mathcal{X}$. We say the distribution $\mathbb{P}_\mathbf{x}$ satisfies $(c, q)$-expansion on $(A, B)$ if for all sets $U \subset B$ with $\mathbb{P}(U|B) > q$, $\mathbb{P}(\mathcal{N}(U)|A) > c\mathbb{P}(U|B)$.*

Figure 1 shows examples of Definition 3 graphically. Intuitively, a pair of sets $A$ and $B$ satisfy this definition if large subsets of $B$ correspond/*expand* (via the neighborhood $\mathcal{N}$) to large subsets of $A$. Definition 3 requires *all sets $U$* with large enough probability in $B$ to expand to $A$. However, this is unnecessarily strong. Our theorems will only need certain *structured* sets, corresponding to elements of the student hypothesis class, to expand. This is captured by Definition 4 and is the key to our results in Section 5 on checking the expansion property.

**Definition 4** (Expansion of a set collection). *Fix sets $A, B \subset \mathcal{X}$ and suppose $\mathcal{M}$ is a collection of subsets of $B$. Then we say $\mathcal{M}$ satisfies $(c, q)$-expansion on $(A, B)$ if all sets $U \in \mathcal{M}$ with $\mathbb{P}(U|B) > q$ satisfy $\mathbb{P}(\mathcal{N}(U)|A) > c\mathbb{P}(U|B)$.*

## 4 Error Bounds for Weakly-Supervised Classifiers

In this section, we upper bound the *gold* error of $f$ on the covered and uncovered sets—$\mathrm{err}(f, y|S)$ and $\mathrm{err}(f, y|T)$—with expressions involving the *weak* error $\mathrm{err}(f, \tilde{y}|S)$ of $f$ on the covered set, expansion parameters, and robustness parameters. These bounds give a theoretical justification for why empirical risk minimization using the weak labels leads to weak-to-strong generalization. We condition on subsets $S_i \subset S$ (for pseudolabel correction) or pairs of subsets $S_i \subset S, T_i \subset T$ (for coverage expansion). This subset-wise conditioning allows for different parts of the distribution to expand in different amounts, yielding tighter bounds, an approach also followed by Cai et al. [12].

### 4.1 Adversarially Robust Models

We begin by stating our theorems for the case when the classifier of interest is $\eta$-robust with $\eta = 0$ on most points $\boldsymbol{x}$. That is, we assume that for most points $\boldsymbol{x}$, the classifier is adversarially robust over $\mathcal{N}(\boldsymbol{x})$. This follows the assumptions in related work from other domains [69, 12]. We describe our results for the significantly more general average-case-robust classifiers in Section 4.2.

**Expanding Families.** We first define the families $\mathcal{M}$ and $\mathcal{M}'$ of sets that must expand according to Definition 4. Let $\mathcal{F}$ be the hypothesis class of the strong model and for each $f \in \mathcal{F}$, define $R(f) = R_0(f) = \{\boldsymbol{x} : r(f, \boldsymbol{x}) = 0\}$ to be the set of adversarially robust points for $f$. For $B \subset \mathcal{X}$ and $f \in \mathcal{F}$, define $U(B, f) = \{\boldsymbol{x} \in B : f(\boldsymbol{x}) \neq y(\boldsymbol{x})\}$ as the set of points in $B$ where $f$ makes a mistake on the true label $y$. Now define $\mathcal{M}(B, \mathcal{F})$ to be the class of *robust* mistakes sets: $\mathcal{M}(B, \mathcal{F}) = \{U(B, f) \cap R(f) : f \in \mathcal{F}\}$. Similarly, define $\mathcal{M}'(B, \mathcal{F}) = \{(B \setminus U(B, f)) \cap R(f) : f \in \mathcal{F}\}$ as the family of robust *non*-mistakes on $B$.

**Pseudolabel Correction.** Here we relate the gold error of the student model $f$ on a covered subset, $\text{err}(f, y|S_i)$, to the weak error of $f$ on that set, $\text{err}(f, \tilde{y}|S_i)$. The goal is to allow for the *correction* of some of the incorrect weak labels ($S_i^{bad}$): we want our bounds for $\text{err}(f, y|S_i)$ to be less than $\alpha_i$, the error rate of the weak labels. Expansion between the points with correct pseudolabels, $S_i^{good}$, and points with incorrect pseudolabels, $S_i^{bad}$, implies that there are many bad points with good neighbors. If the classifier is suitably robust on the neighborhoods, pseudolabel correction can occur. The bound in this section makes this intuition quantitative.

**Theorem 4.1** (Pseudolabel correction). *Suppose $\mathcal{M}'(S_i^{good}, \mathcal{F})$ satisfies $(c, q)$-expansion on the sets $(S_i^{bad}, S_i^{good})$ for $q < \frac{3}{4}(1 - 2\alpha)$. Consider an arbitrary classifier $f \in \mathcal{F}$ such that $\mathbb{P}(f(\mathbf{x}) \neq \tilde{y}(\mathbf{x})$ or $f$ not robust at $\mathbf{x}|S_i) \leq \frac{1 - \alpha + 3c\alpha}{4}$. Then the true error of $f$ on $S_i$ satisfies:*

$$\text{err}(f, y|S_i) \leq \frac{2\alpha_i}{1 - 2\alpha_i}\mathbb{P}(\overline{R(f)}|S_i) + \text{err}(f, \tilde{y}|S_i) + \alpha_i\left(1 - \frac{3}{2}c\right).$$

The expansion condition intuitively states that "good" sets (elements of $M'(S_i^{good}, \mathcal{F})$) must have suitably many neighbors with the wrong pseudolabel (elements of $S_i^{bad}$). The trivial error bound obtained via the triangle inequality is $\text{err}(f, y|S_i) \leq \text{err}(f, \tilde{y}|S_i) + \alpha_i$. The bound in Theorem 4.1 has almost the same form, but the multiplicative term on $\alpha_i$ allows it to be much tighter than the trivial bound. Theorem 4.1 allows for pseudolabel correction because the right-hand-side can be much less than $\alpha_i$ (the error of the weak teacher) when $c$ is large and $\text{err}(f, \tilde{y}|S_i)$ and $\mathbb{P}(\overline{R(f)}|S_i)$ are small. While Wei et al. [69, Theorem 4.3] also gives pseudolabel correction guarantees for adversarially robust classifiers, Theorem 4.1 is a different bound with several desirable properties that Wei et al. [69, Theorem 4.3] lacks—we compare the two in detail in Appendix C.2 and also show how to generalize Wei et al. [69]'s results to average-case-robustness.

The following proposition shows that the expansion conditions in Theorem 4.1 rule out strong models that can exactly fit the weak model.

**Proposition 4.1.** *Suppose there exists $f \in \mathcal{F}$ such that for all $\boldsymbol{x} \in S$, $f(\boldsymbol{x}) = \tilde{y}(\boldsymbol{x})$. Then the $(c, q)$-expansion conditions of Theorem 4.1 are not satisfied.*

*Proof.* The family of sets that must satisfy $(c, q)$-expansion between $S_i^{bad}$ and $S_i^{good}$ is $\mathcal{M}' = \{R(f) \cap (S_i^{good} \setminus \text{mistakes}(f)) : f \in \mathcal{F}\}$. Choose $f$ such that $f = \tilde{y}$ when restricted to $S$, since we assumed this is a valid choice. $S_i^{good}$ is defined as the set where $\tilde{y}$ makes no mistakes (and the true label is $i$), so $S_i^{good} \setminus \text{mistakes}(f) = S_i^{good}$, and thus $R(f) \cap S_i^{good} \in \mathcal{M}'$. Let $N = \{\boldsymbol{x} \in S_i^{good} : \mathcal{N}(\boldsymbol{x}) \cap S_i^{bad} \neq \emptyset\}$ be the subset of $S_i^{good}$ with neighbors in $S_i^{bad}$. Consider an arbitrary point $\boldsymbol{x} \in N$, so there exists $\boldsymbol{x}' \in \mathcal{N}(\boldsymbol{x}) \cap S_i^{bad}$. Since $\boldsymbol{x}$ and $\boldsymbol{x}'$ are both in $S_i$, $y(\boldsymbol{x}) = y(\boldsymbol{x}')$. Then we have $f(\boldsymbol{x}) = \tilde{y}(\boldsymbol{x}) = y(\boldsymbol{x}) = y(\boldsymbol{x}') \neq \tilde{y}(\boldsymbol{x}') = f(\boldsymbol{x}')$, so $f(\boldsymbol{x}) \neq f(\boldsymbol{x}')$. Thus $\boldsymbol{x} \notin R(f)$. This shows $N \subset \overline{R(f)}$. Hence

$$R(f) \cap S_i^{good} = R(f) \cap \left((S_i^{good} \cap N) \cup (S_i^{good} \cap \bar{N})\right)$$

$$\subset R(f) \cap \left((S_i^{good} \cap \overline{R(f)}) \cup (S_i^{good} \cap \bar{N})\right)$$

$$= R(f) \cap (S_i^{good} \cap \bar{N}).$$

The latter set is made up entirely of points with no neighbors in $S_i^{bad}$, so

$$\mathbb{P}(\mathcal{N}(R(f) \cap S_i^{good})|S_i^{bad}) \leq \mathbb{P}(\mathcal{N}(R(f) \cap (S_i^{good} \cap \bar{N}))|S_i^{bad}) = 0.$$

But for expansion to hold, we needed $\mathbb{P}(\mathcal{N}(R(f) \cap S_i^{good})|S_i^{bad}) > c\mathbb{P}(R(f) \cap S_i^{good}|S_i^{good})$. □

Proposition 4.1 shows the strong model hypothesis class enters our error bounds indirectly via the (data-dependent) expansion parameter. Using a richer class for the strong model may decrease the amount of expansion, since it may make it easier to exactly fit the weak labels. At the same time, the error of the strong model on the weak labels also appears as a term in the bounds ($\text{err}(f, \tilde{y}|S_i)$), and a richer class might decrease this term, so these two terms capture a tradeoff. This makes our bounds

flexible enough to capture empirical results showing that whether a richer class or more restricted class works best for the strong model depends on the problem [73].

**Coverage Expansion.** In this section, we relate the error of $f$ on an uncovered subset, $\text{err}(f, y|T_i)$, to the weak error of $f$ on the corresponding covered subset, $\text{err}(f, \tilde{y}|S_i)$. The goal is to give a nontrivial error bound on these points even though we see none of them during training. Expansion from $T_i$ to $S_i^{good}$ implies that subsets of $T_i$ have enough correctly-pseudolabeled neighbors. If the student model is robust on $\mathcal{N}$, this is already enough to prove an error bound for $T_i$. However, we *also* assume that subsets of $T_i$ have enough *incorrectly*-pseudolabeled neighbors. Intuitively, this means that subsets of $T_i$ have the "correct" number of neighbors in $S_i^{good}$ *and* the "correct" number of neighbors in $S_i^{bad}$. This implies a regular structure in the $S_i$–$T_i$ neighborhood connections that allows us to prove a much tighter error bound. Our empirical results suggest that this structure is present in real-world examples. We prove a weaker bound that only assumes expansion from $T_i$ to $S_i^{good}$ in Appendix B.

**Theorem 4.2** (Error bound for uncovered points). *Suppose $\mathcal{M}(T_i, \mathcal{F})$ satisfies $(c, q)$-expansion on $(S_i^{good}, T_i)$, and $\mathcal{M}'(T_i, \mathcal{F})$ satisfies $(c, q)$-expansion on $(S_i^{bad}, T_i)$. Consider an arbitrary classifier $f \in \mathcal{F}$ that fits the weak labels well on $S_i$ and is fairly robust on $T_i$: $\text{err}(f, \tilde{y}|S_i) + \mathbb{P}(\overline{R(f)}|T_i) < c(1 - q - \alpha_i)$ Then the true error of $f$ on $T_i$ satisfies:*

$$\text{err}(f, y|T_i) \leq \left(1 + \frac{\alpha_i}{1 - 2\alpha_i}\right) \mathbb{P}(\overline{R(f)}|T_i) + \max\left(q, \frac{\text{err}(f, \tilde{y}|S_i) - c\alpha_i}{c(1 - 2\alpha_i)}\right).$$

To qualify for the bound, $f$ must fit the weak labels well on $S_i$, so $\text{err}(f, \tilde{y}|S_i)$ is small, and be adversarially robust at most points on $T_i$, so $\mathbb{P}(\overline{R(f)}|T_i)$ is small. We show in Appendix C that the original co-training setup of Blum and Mitchell [6] satisfies the assumptions of Theorem 4.2 with $c = 1$, $q = 0$, and $\mathbb{P}(\overline{R(f)}|T_i) = 0$. Theorem 4.2 exactly recovers the bounds of Blum and Mitchell [6], Lang et al. [37] in this case, so Theorem 4.2 is a direct generalization of Blum and Mitchell [6] but without the restrictive assumptions regarding multi-view data and conditional independence. In Appendix B, we prove a generalization of Theorem 4.2 that allows $T_i$ to expand to $S_i^{good}$ and $S_i^{bad}$ at different rates (i.e., different expansion parameters).

It is not immediately clear from Theorem 4.2 how the *coverage* $\mathbb{P}(S)$—the probability that the weak labeler does not abstain—affects the bounds. As with the role of strong model hypothesis class, this enters the picture implicitly through the parameter measuring expansion between $S_i$ and $T_i$. Informally, for a fixed neighborhood $\mathcal{N}$, it may be easier to have expansion from $T_i$ to $S_i$ when $S_i$ is larger, since for each uncovered point in $T_i$, there are more possible covered neighbors in $S_i$. On the other hand, increasing the coverage by including more points in $S_i$ might also affect the weak label accuracy parameters $\alpha_i$, which also appear in the bounds. In practice, there is not a consistent tradeoff between coverage and performance, as explored recently in, e.g., Lang et al. [37], so our bounds do not prescribe a functional dependence of the error on the amount of coverage and instead allow that dependence to enter through data-dependent parameters.

## 4.2 Relaxing Robustness Requirements

The previous bounds in this section assumed the student is adversarially robust at most points. Here, we considerably generalize this requirement so our results apply to any classifier $f$ that satisfies (1), i.e., any classifier that gives *most* points the same label as *most* of their neighbors. This requires several additional definitions and goes beyond the assumptions made in other work with expansion-based error bounds, which assume *adversarial* robustness at most [69, 12] or (effectively) all [6] points.

To allow for this relaxed assumption on the classifier, we assume a more robust version of expansion, aptly called *robust expansion* [27, 34, 42]. To define robust expansion, we start by defining a graph over examples with edges induced by the neighborhood $\mathcal{N}$ and weights given by the underlying probability measure. HaoChen et al. [22] study this graph in the context of contrastive pretraining.

**Definition 5** (Example graph). *Let $G = (\mathcal{X}, E)$ be a graph with one node for each element of $\mathcal{X}$ (we assumed $\mathcal{X}$ is a possibly very large, but finite, set), and connect two nodes $(\boldsymbol{x}, \boldsymbol{x}')$ if $\boldsymbol{x} \in \mathcal{N}(\boldsymbol{x}')$ or, equivalently, if $\boldsymbol{x}' \in \mathcal{N}(\boldsymbol{x})$, with an edge weight of $w(\boldsymbol{x}, \boldsymbol{x}') := \mathbb{P}(\boldsymbol{x})\mathbb{P}(\boldsymbol{x}')\mathbb{1}[\boldsymbol{x} \in \mathcal{N}(\boldsymbol{x}')]$.*

Definition 3 (regular, non-robust expansion) is very sensitive to removal of a few edges from the example graph. A set $U \subset B$ may have $\mathbb{P}(\mathcal{N}(U)|A)$ large, but only because a small fraction of the

edges (by probability mass) are connected to many $\boldsymbol{x} \in A$ with $\mathbb{P}(\boldsymbol{x}|A)$ large. If we ignored these small-probability edges, the neighborhood would be much smaller. Figure 2 (appendix) shows an example. The *robust neighborhood* tries to address this issue:

**Definition 6** ($\eta$-robust neighborhood size). *Let $A, U \subset \mathcal{X}$. The size of the $\eta$-robust neighborhood of $U$ in $A$ is: $P_{1-\eta}(U, A) := \min_{V \subset \mathcal{X}}\{\mathbb{P}(V|A) : w(V, U) \geq (1 - \eta)w(\mathcal{N}(U), U)\}$.*

$P_{1-\eta}(U, A)$ is the probability of the "smallest" subset of $A$ that still captures at least a $1 - \eta$ fraction of the edge weight incident on $U$. When $\eta = 0$, we have $P_1(U, A) = \mathbb{P}(\mathcal{N}(U)|A)$, so this recovers the size of the non-robust neighborhood. In the pathological example described above, we would have $\mathbb{P}(\mathcal{N}(U)|A)$ large, but $P_{1-\eta}(U, A)$ small for some $\eta > 0$. We are now ready to give a "robustified" definition of expansion, which is identical to Definition 4 except that it requires the *robust* neighborhood, rather than the regular neighborhood, to be large.

**Definition 7** (Robust expansion). *Fix sets $A, B \subset \mathcal{X}$ and suppose $\mathcal{M}$ is a collection of subsets of $B$. $\mathcal{M}$ satisfies $(c, q, \eta)$-robust expansion on $(A, B)$ if for all $U \in \mathcal{M}$ with $\mathbb{P}(U|B) > q$, $P_{1-\eta}(U, A) > c\mathbb{P}(U|B)$. This exactly recovers Definition 3 when $\eta = 0$.*

The following (informal) theorem shows that Theorems 4.1 and 4.2 hold for average-case-robust classifiers when we replace expansion with robust expansion and $R(f)$ with $R_\eta(f)$. We state and prove formal versions of Theorems 4.1 and 4.2 for average-case-robust classifiers in Appendix B.

**Theorem 4.3** (Informal). *Theorems 4.1 and 4.2 hold exactly with $(c, q, \eta)$-expansion instead of $(c, q)$-expansion and $R_\eta(f)$ instead of $R(f)$.*

By Markov's inequality, for any $\eta > 0$ a classifier $f$ with $\mathbb{E}_{\mathbf{x} \sim \mathcal{D}|A, \mathbf{x}' \sim \mathcal{D}|\mathcal{N}(\mathbf{x})}[f(\mathbf{x}) \neq f(\mathbf{x}')] \leq \gamma$ has $\mathbb{P}(\overline{R_\eta(f)}|A) \leq \frac{\gamma}{\eta}$. Theorem 4.3 shows that by assuming the data distribution follows a slightly more "regular" structure (robust expansion), we can give guarantees for average-case-robust classifiers. This generalization is important since it matches with empirical results: adversarial training and adversarial robustness are *not* required for weak-to-strong generalization to occur [73, 38, 10], and most empirical work on weak supervision does not include adversarial training in the pipeline [71].

# 5 Checking Expansion

We now outline a statistical theory for checking the expansion properties of the population distribution from finite data. This is possible because, as described in Section 4, our results do not actually require *all sets* to expand—rather, they only require expansion for a class of sets that is generated by the student hypothesis class. This means we can check expansion on a finite dataset and control the generalization of our estimate using the complexity of the hypothesis class. The purpose of checking expansion is not (currently) algorithmic—the goal of the procedures described in this section is to give empirical evidence that our assumptions hold in the real world and that our bounds correlate with actual occurrences of pseudolabel correction and coverage expansion. Exactly checking the expansion of *all subsets* is coNP-complete [7]; whether our notion of expansion with respect to a certain family of sets $\mathcal{M}$ can be checked efficiently is an interesting direction for future research. We show that expansion can at least be checked *statistically* (i.e., from finite data), if not efficiently.

For a fixed choice of $q$, the (non-robust) expansion of a set family $\mathcal{M}$ between sets $A$ and $B$ is: $c = \min_{U \in \mathcal{M}: \mathbb{P}(U|B) > q} \frac{\mathbb{P}(\mathcal{N}(U)|A)}{\mathbb{P}(U|B)}$. Suppose we have two samples $\mathcal{S}_A = \{(\boldsymbol{x}_i, y(\boldsymbol{x}_i))\}_{i=1}^{n_A}$ with $\mathbf{x} \sim \mathbb{P}(\cdot|A)$, and $\mathcal{S}_B = \{(\boldsymbol{x}_i, y(\boldsymbol{x}_i))\}_{i=1}^{n_B}$ with $\mathbf{x} \sim \mathbb{P}(\cdot|B)$. For a fixed $U$, the denominator is straightforward to estimate using $\mathcal{S}_B$ as: $\mathbb{P}(U|B) \approx \frac{1}{n_B} \sum_{\boldsymbol{x}_i \in \mathcal{S}_B} \mathbb{1}[\boldsymbol{x}_i \in U]$. Estimating the numerator is less straightforward: due to finite sampling, $\mathcal{S}_A \times \mathcal{S}_B$ may contain no pairs $(\boldsymbol{x}, \boldsymbol{x}')$ with $\boldsymbol{x} \in \mathcal{N}(\boldsymbol{x}')$. That is, the empirical neighbor graph may be empty even when the population distribution expands (see Wei et al. [69] for a more thorough discussion). This is a major difference between our assumptions and similar work that uses expansion-like assumptions to analyze the performance of label-propagation algorithms that use the empirical graph, such as Pukdee et al. [48]. To overcome this, we assume we have access to a neighborhood oracle $n : A \to B$ that for each $\boldsymbol{x} \in A$ returns a point $n(\boldsymbol{x}) \in B$ such that $n(\boldsymbol{x}) \in \mathcal{N}(\boldsymbol{x})$. We assume nothing about the distribution of $n(\boldsymbol{x})$ values (i.e., we do not assume that they are drawn from $\mathbb{P}(\cdot|B)$, merely that $\mathbb{P}(n(\boldsymbol{x})|B) > 0$). We describe how to construct $n$ in a practical scenario in Section 6.

The neighborhood oracle makes estimating the expansion numerator more straightforward, since if $n(\boldsymbol{x}) \in U$, then by construction, $\boldsymbol{x} \in \mathcal{N}(U)$. Formally, $\mathbb{P}(\mathcal{N}(U)|A) \geq \mathbb{P}(n(\mathbf{x}) \in U|A)$, where the

Table 1: Measured expansion and error bounds for the covered sets $S_i$. Expansion values for the family of sets $\mathcal{M}'(S_i^{good}, \mathcal{F})$ are measured using the heuristic described in Section 5 and shown in the $(S_i^{bad}, S_i^{good})$ *exp.* column. This column shows our heuristic finds expansion in practice. Pseudolabel error $\alpha_i = \mathbb{P}(\tilde{y} \neq y | S_i)$. Worst-case error of trained classifier $f$ on the weak labels $\tilde{y}$, $\text{err}(f, \tilde{y}|S_i)$, across 5 independent training runs. This column shows the student can't exactly fit the teacher labels using this representation. Value of the error upper bound in Theorem 4.1 (specifically, the tighter version, B.1), computed using the numbers from the other columns (details in Appendix E). For label $i = 0$, the bound being strictly less than the error $\alpha_i$ of the teacher $\tilde{y}$ suggests pseudolabel correction may occur. Finally, the actual worst-case error of trained classifier $f$ on the *true* labels $y$, $\text{err}(f, y|S_i)$, across 5 independent training runs, shows pseudolabel correction *does* occur for label $i = 0$.

| **Model** | $i$ | $(S_i^{bad}, S_i^{good})$ exp. | $\alpha_i$ | $\text{err}(f, \tilde{y}|S_i)$ | Bound val | $\text{err}(f, y|S_i)$ |
|---|---|---|---|---|---|---|
| SentenceBERT | 0 | 0.848 | 0.11 | 0.12 | 0.05 | 0.04 |
| | 1 | 0.497 | 0.33 | 0.29 | 0.37 | 0.35 |

quality of $n(\boldsymbol{x})$ determines the tightness of this bound. This inequality is valid for any $n : A \to B$ as long as $\boldsymbol{x} \in \mathcal{N}(n(\boldsymbol{x}))$. Now we can estimate: $\mathbb{P}(n(\mathbf{x}) \in U | A) \approx \frac{1}{n_A} \sum_{\boldsymbol{x}_i \in \mathcal{S}_A} \mathbb{1}[n(\boldsymbol{x}_i) \in U]$. Putting it all together, we can form our empirical estimate of the expansion by solving $\hat{c} = \min_{U \in \mathcal{M}} \frac{\frac{1}{n_A} \sum_{\boldsymbol{x}_i \in \mathcal{S}_A} \mathbb{1}[n(\boldsymbol{x}_i) \in U]}{\frac{1}{n_B} \sum_{\boldsymbol{x}_i \in \mathcal{S}_B} \mathbb{1}[\boldsymbol{x}_i \in U]}$ subject to: $\frac{1}{n_B} \sum_{\boldsymbol{x}_i \in \mathcal{S}_B} \mathbb{1}[\boldsymbol{x}_i \in U] \geq q - \epsilon$, where $\epsilon$ is chosen appropriately to account for empirical error in estimating the probability $\mathbb{P}(U|B)$. The following theorem, proven in Appendix D, shows that the expansion on the population distribution can't be too much smaller than the expansion on the empirical distribution.

**Theorem 5.1** (Expansion generalization, informal). *For arbitrary $U \in \mathcal{M}$, define the population and empirical expansion estimates as: $c(U) := \mathbb{P}(n(\mathbf{x}) \in U | \mathbf{x} \in A)/\mathbb{P}(\mathbf{x} \in U | \mathbf{x} \in B)$ and $\hat{c}(U) := \frac{1}{n_A} \sum_{i=1}^{n_A} \mathbb{1}[n(\boldsymbol{x}_i) \in U]/\frac{1}{n_B} \sum_{j=1}^{n_B} \mathbb{1}[\boldsymbol{x}_i \in U]$. Then for any $\delta \in (0, 1]$, with probability at least $1 - \delta$, $\sup_{U \in \mathcal{M}} \hat{c}(U) - c(U) \leq \widetilde{\mathcal{O}}(\sqrt{\text{VC}(\mathcal{M})/n_A})$, where $\widetilde{\mathcal{O}}$ hides constants and log factors in $\text{VC}(\mathcal{M})$, $n_A + n_B$, and $1/\delta$.*

**Heuristic approximation.** While Theorem 5.1 gives a rigorous statistical theory for checking expansion from finite data, it is unfortunately still intractable to compute the set with the worst expansion on the empirical data, i.e., to solve $\hat{U} = \arg\min_{U \in \mathcal{M}} \hat{c}(U)$. Instead, our experiments in Section 6 use a simple randomized heuristic for approximating this minimization. If the learning algorithm $\mathcal{A} : S^m \to \mathcal{F}$ is deterministic conditioned on the observed training data, we can simplify our hypothesis class $\mathcal{F}$ of interest to those $f \in \mathcal{F}$ such that there exists a training sample $\mathcal{S} \subset S$ with $f = \mathcal{A}(\mathcal{S})$. Since each $f \in \mathcal{F}$ generates a set $U(f) \in \mathcal{M}$, given a training sample $\mathcal{S}$, we can compute $f = \mathcal{A}(\mathcal{S})$, then use our "test" sample $\mathcal{S}_A, \mathcal{S}_B$ to compute $\hat{c}(U(f))$. Repeating this procedure for many samples $\mathcal{S}$ and choosing the smallest value approximates $\min_{U \in \mathcal{M}} \hat{c}(U)$. Table 1 shows the expansion measurements for different hypothesis classes on a weakly-supervised classification task inspired by the example in Section 3. Appendix 6 contains more results and a much more detailed description of the setup and process of checking expansion, but these results indicate that expansion is present and correlates with performance.

## 6 Experiments

**Setup.** We explore training linear classifiers on top of the contrastively-fine-tuned SentenceBERT embeddings[2] [52]. As shown in Muennighoff et al. [44], training simple classifiers on top of these complex pretrained representations leads to very competitive performance. We study binary sentiment prediction for movie reviews on the IMDb dataset [41], continuing with the example from Section 3. For the teacher model, we use a very coarse rule [49] based on the presence of the unigrams "incredible" and "horrible". Let $C(w, \boldsymbol{x})$ be the number of times word $w$ appears in input $\boldsymbol{x}$. The weak label $\tilde{y}(\boldsymbol{x})$ is 1 when $C(\texttt{incredible}, \boldsymbol{x}) > C(\texttt{horrible}, \boldsymbol{x})$, 0 when $C(\texttt{horrible}, \boldsymbol{x}) > C(\texttt{incredible}, \boldsymbol{x})$, and $\varnothing$ otherwise. This counts the occurrences of "horrible" and "incredible" and assigns the binary label corresponding to the word that occurs strictly more often, and abstains otherwise.

---

[2]HuggingFaceHub model ID `sentence-transformers/all-mpnet-base-v2`

**Neighborhood function and oracle.** We set $\mathcal{N}(\boldsymbol{x})$ to be the examples obtainable from $\boldsymbol{x}$ by sampling from a pretrained paraphrase model. As described in Section 5, to measure expansion between sets $A$ and $B$, we need a neighborhood oracle $n : A \to B$ that, given $\boldsymbol{x} \in A$, returns a point $\boldsymbol{x}' \in \mathcal{N}(\boldsymbol{x}) \cap B$. Our results require us to measure expansion between $(S_i^{good}, T_i)$, $(S_i^{bad}, T_i)$, and $(S_i^{bad}, S_i^{good})$. For $\boldsymbol{x} \in S_i^{good}$ (resp. $\boldsymbol{x} \in S_i^{bad}$), we generate a target point $\boldsymbol{x}' \in \mathcal{N}(\boldsymbol{x}) \cap T$ by rejection sampling from a pretrained paraphrase model. Because $\tilde{y}$ takes a simple form, we can efficiently approximate this step by setting the logits of tokens "horrible" and "incredible" to $-\infty$ during decoding so they are never generated. For $\boldsymbol{x} \in S_i^{bad}$, to generate a neighbor $\boldsymbol{x}'' \in \mathcal{N}(\boldsymbol{x}) \cap S_i^{good}$, we prompt GPT-4 to paraphrase a randomly chosen sentence from $\boldsymbol{x}' \in \mathcal{N}(\boldsymbol{x}) \cap T_i$ and include the *correct* word in its paraphrase, so that $\boldsymbol{x}'' \in S_i^{good}$. We rejection sample until this constraint is satisfied. Figure 3 (appendix) shows examples of these procedures.

**Expansion results.** Table 1 measures the expansion of the set family $\mathcal{M}'(S_i^{good}, \mathcal{F})$ on the sets $(S_i^{bad}, S_i^{good})$ using the procedure from Section 5. Theorem 4.1 shows this is related to pseudolabel correction. For the SentenceBERT model, the measured expansion is high and the student fits the weak labels well, but doesn't overfit to the teacher labels (i.e., $\text{err}(f, \tilde{y}|S_i) > 0$). For label 0, our bound indicates that pseudolabel correction should be present, since the value for our error bound is less than $\alpha_i$. There is indeed pseudolabel correction on this label: $\text{err}(f, y|S_0) < \alpha_0$. For label 1, our bound indicates that pseudolabel correction may not occur—the bound value is greater than $\alpha_i$ since the the measured expansion is lower for this label and the error of the student on the weak labels is higher. As suggested by the bound, there is no pseudolabel correction: $\text{err}(f, y|S_1) > \alpha_1$. Our expansion-based theory can therefore differentiate between cases where pseudolabel correction does and does not occur. Table 4 (appendix) shows the measured expansion values between the set pairs $(S_i^{good}, T_i)$ and $(S_i^{bad}, T_i)$, which Theorem 4.2 shows are related to the amount of coverage expansion. For example, for label 1, $T_i$ expands to both $S_i^{good}$ ($c = 0.75$) *and* to $S_i^{bad}$ ($c = 0.55$). The fact that both expansions are nonzero gives evidence for the structure assumed Theorem 4.2. In this case, our coverage expansion bounds show that the student model has nontrivial performance on the uncovered sets $T_i$—for example, for label 1, the worst-case value of $\text{err}(f, y|T_1)$ in all the training runs is 0.29, and the value of our bound is 0.33. Appendix E contains more details on how the models are trained and the bounds are computed.

## 7 Limitations & Conclusion

In this work, we proved error bounds based on *expansion* properties of the data distribution and student hypothesis class that directly allow for weak-to-strong generalization, gave a statistical theory for checking these expansion properties, and gave empirical evidence that they hold in practice. Our empirical procedure for finding the worst-expanding set generated by our hypothesis class is ultimately still a heuristic, and our experiments are limited in scope. However, Sections 5 and 6 go beyond prior work by testing our assumptions more carefully. Finally, this work contains no new weak supervision *algorithms* (e.g., new training methods) for improving weak-to-strong generalization. While our work does not propose new weak supervision algorithms, we believe our theory suggests a framework for encouraging weak-to-strong generalization effects: find a neighborhood structure and student hypothesis class pair that expands, then find the student model $f \in \mathcal{F}$ that minimizes the error on the weak labels while staying as robust as possible on the neighborhoods. Both expansion and contrastive pre-training are related to spectral properties of the underlying neighborhood graph [22]. Can we improve the performance of weakly-supervised learning by imbuing the contrastive pretraining objective with knowledge of the pseudolabeler $\tilde{y}$? Burns et al. [10] show that weak-to-strong generalization effects are much stronger in classification problems than in other settings such as reward modeling. Adapting our analysis techniques to reward modeling and using our results to design algorithms for weak-to-strong reward model training are interesting directions for future work.

## Acknowledgments

Thanks to Hussein Mozannar and Ilker Demirel for feedback on drafts of this paper. DS and HL were partially supported by NSF AitF award CCF-1723344, AV was supported by NSF grants EECS2216970 and CCF-2154100, and HL was supported by the NDSEG fellowship. Finally, this project was partially supported by an OpenAI "Superalignment Fast" grant.

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

# A   Additional Related Work

**Programmatic Weak Supervision.** Pukdee et al. [48] study the performance of label propagation for weak supervision. Unlike our work, they assume expansion with respect to the empirical sample graph and study the performance of a particular learning algorithm (label propagation). Chen et al. [13] show how to propagate weak labels using the embeddings of a strong pretrained model to provably improve performance, but they also focus on the empirical graph, and not on guarantees for the classifiers trained on the weak labels. Unlike most other work on weak supervision, which tries to give separate algorithms and guarantees for (i) the procedure for creating a pseudolabel out of multiple labeling functions and (ii) the student training, Sam and Kolter [55], Rühling Cachay et al. [54] consider direct end-to-end weak supervision. However, they do not give theoretical error guarantees for the student models. Cabannnes et al. [11] give theoretical guarantees for a different flavor of weak supervision, when the "teacher" gives a set of labels that contains the ground-truth but may also contain other incorrect labels. Robinson et al. [53] show that when only a few gold labels are available, weak labels from a teacher model can speed up the learning process.

**Domain Adaptation.** What we call "coverage expansion" is very related to some work on domain adaptation that can still apply when the source and target distributions do not overlap, such as Ben-David et al. [4], Blitzer et al. [5]. Our expansion assumptions for coverage expansion, which require the expansion of certain families of sets (generated by the student hypothesis class), are qualitatively very related to the $\mathcal{F}\Delta\mathcal{F}$ distance [4]. Assuming the $\mathcal{F}\Delta\mathcal{F}$ distance is small essentially says that the mistake set of any hypothesis in $\mathcal{F}$ must have similar probability in both the source and target distribution, so the target error can't be much higher than the source error. Kifer et al. [31] showed that this distance can be statistically estimated from a finite sample. In an imprecise sense, our results suggest that the $\mathcal{F}\Delta\mathcal{F}$ distance is small when every hypothesis $f \in \mathcal{F}$ is robust on the neighborhoods $\mathcal{N}$ and the distribution has good expansion. We also have one unified set of assumptions that leads to guarantees for both coverage expansion/domain adaptation *and* pseudolabel correction, and our coverage expansion bounds account for the error of the teacher model on the source domain. Abbe et al. [1] also give theoretical guarantees for learning boolean functions when part of the data domain is completely unseen during training.

**Knowledge Distillation, Pseudolabel correction.** The observation that a distilled model can outperform its teacher (i.e., what Burns et al. [10] calls weak-to-strong generalization) goes back at least to Buciluǎ et al. [9]. In the context of knowledge distillation, Stanton et al. [58] show that even when student models have the capacity to match the teacher, they don't match them exactly. They give empirical examples of this phenomenon and argue that it can be due to optimization issues during student training. Clearly, this effect is critical for weak-to-strong generalization, and our expansion and robustness assumptions effectively try to capture the structure (whether implicit, due to the optimization process, or explicit, due to the choice of student hypothesis class) that rules out this exact-teacher-fitting behavior. In the context of self-training, Chen et al. [14], Kumar et al. [32], Oymak and Cihad Gulcu [47], Frei et al. [19] all (effectively) give pseudolabel correction guarantees under certain distributional assumptions, such as when the data are distributed according to a Gaussian Mixture Model [47, 19]. In contrast, like other work with expansion assumptions [3, 69, 12, 22], our results do not assume the input data follows any specific distributional form. Indeed, Wei et al. [69] showed that distributions similar to the Gaussian Mixture Models studied in these works satisfy a notion of expansion very similar to the one we study here.

# B   Error bound proofs

We directly prove the "average-case-robust" versions of Theorems 4.1 and 4.2. Theorems 4.1 and 4.2 directly follow from the equivalence between expansion and robust expansion, and adversarial robustness and $\eta$-robustness, when $\eta = 0$. For convenience, we reproduce the definitions of the example graph, $\eta$-robustness, and robust expansion here. Figure 2 shows examples of good and bad robust expansion.

We begin by proving some lemmas that will be useful for both the pseudolabel correction and coverage expansion bounds.

**Definition** (Example graph). *Let $G = (\mathcal{X}, E)$ be a graph with one node for each element of $\mathcal{X}$ (we assumed $\mathcal{X}$ is a possibly very large, but finite, set), and connect two nodes $(\boldsymbol{x}, \boldsymbol{x}')$ if $\boldsymbol{x} \in \mathcal{N}(\boldsymbol{x}')$ or, equivalently, if $\boldsymbol{x}' \in \mathcal{N}(\boldsymbol{x})$, with an edge weight of $w(\boldsymbol{x}, \boldsymbol{x}') := \mathbb{P}(\boldsymbol{x})\mathbb{P}(\boldsymbol{x}')\mathbb{1}[\boldsymbol{x} \in \mathcal{N}(\boldsymbol{x}')]$.*

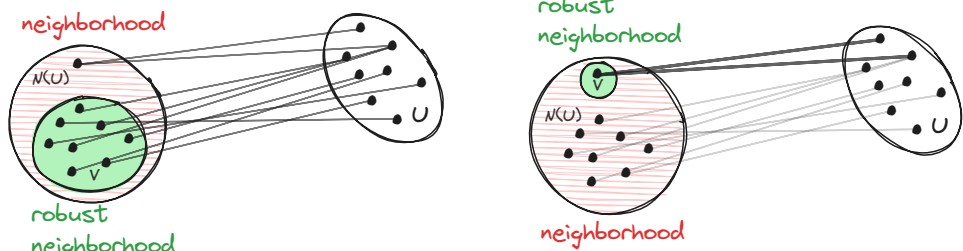

Figure 2: Examples of good (left) and bad (right) robust expansion. In both cases, there is a core subset $V \subset \mathcal{N}(U)$ that accounts for most of the edge weight incident on $U$ (at least a $1 - \eta$ fraction, for some small $\eta$). The robust expansion is good when every such subset has large probability.

**Definition** ($\eta$-robust). *For an arbitrary classifier $f$ and point $\boldsymbol{x}$, define $r(f, \boldsymbol{x}) = \mathbb{P}(f(\mathbf{x}') \neq f(\boldsymbol{x})|\mathbf{x}' \in \mathcal{N}(\boldsymbol{x}))$ as the probability $f$ gives different labels to $\boldsymbol{x}$ and a random neighbor $\mathbf{x}'$ of $\boldsymbol{x}$. A classifier $f : \mathcal{X} \to \mathcal{Y}$ is said to be $\eta$-robust at a point $\boldsymbol{x}$ if $r(f, \boldsymbol{x}) \leq \eta$. For an arbitrary classifier $f$, let $R_\eta(f) = \{\boldsymbol{x} : r(f, \boldsymbol{x}) \leq \eta\}$ be the set of $\eta$-robust points for $f$.*

**Definition** ($\eta$-robust neighborhood size). *Let $A, U \subset \mathcal{X}$. The size of the $\eta$-robust neighborhood of $U$ in $A$ is: $P_{1-\eta}(U, A) := \min_{V \subset \mathcal{X}}\{\mathbb{P}(V|A) : w(V, U) \geq (1 - \eta)w(\mathcal{N}(U), U)\}$.*

**Definition** (Robust expansion). *Fix sets $A, B \subset \mathcal{X}$ and suppose $\mathcal{M}$ is a collection of subsets of $B$. $\mathcal{M}$ satisfies $(c, q, \eta)$-robust expansion on $(A, B)$ if for all $U \in \mathcal{M}$ with $\mathbb{P}(U|B) > q$, $P_{1-\eta}(U, A) > c\mathbb{P}(U|B)$. This exactly recovers Definition 4 when $\eta = 0$.*

The following lemma shows that a classifier that is robust on average must also be $\eta$-robust on a set of large probability.

**Lemma B.1.** *Fix a set $A \subset \mathcal{X}$ and a classifier $f$, and suppose*

$$\mathbb{E}_{\mathbf{x} \sim \mathcal{D}|A, \mathbf{x}' \sim \mathcal{N}(\mathbf{x})}[f(\mathbf{x}) \neq f(\mathbf{x}')] \leq \gamma$$

*for some $\gamma > 0$. Then for any $\eta > 0$, $\mathbb{P}(\overline{R_\eta(f)}|A) \leq \frac{\gamma}{\eta}$.*

*Proof.* Rewriting the condition on $f$ slightly,

$$\mathbb{E}_{\mathbf{x} \sim \mathcal{D}|A}[\mathbb{P}_{\mathbf{x}'}(f(\mathbf{x}) \neq f(\mathbf{x}')|\mathbf{x}' \in \mathcal{N}(\mathbf{x}))] \leq \gamma.$$

Recall that $r(\boldsymbol{x}, f) = \mathbb{P}_{\mathbf{x}'}(f(\boldsymbol{x}) \neq f(\mathbf{x}')|\mathbf{x}' \in \mathcal{N}(\boldsymbol{x}))$. So we have $\mathbb{E}_{\mathbf{x} \sim \mathcal{D}|A}[r(\mathbf{x}, f)] \leq \gamma$. Markov's inequality implies that for any $\eta > 0$,

$$\mathbb{P}(r(\mathbf{x}, f) > \eta|A) \leq \frac{\mathbb{E}_{\mathbf{x} \sim \mathcal{D}|A}[r(\mathbf{x}, f)]}{\eta} \leq \frac{\gamma}{\eta}.$$

Since $R_\eta(f) = \{\boldsymbol{x} : r(\boldsymbol{x}, f) \leq \eta\}$, we thus have

$$\mathbb{P}(\overline{R_\eta(f)}|A) \leq \frac{\gamma}{\eta},$$

which concludes the proof. $\square$

**Good and bad edges.** Consider an arbitrary classifier $f$ and arbitrary set $U \subset \mathcal{X}$ and fix $\boldsymbol{x}' \in U$, $\boldsymbol{x} \in \mathcal{N}(U)$. We say the pair $(\boldsymbol{x}, \boldsymbol{x}')$ is *bad* if $f(\boldsymbol{x}) \neq f(\boldsymbol{x}')$; otherwise the pair is *good*. Let $\widetilde{\mathcal{N}}(U)$ be the subset of $\mathcal{N}(U)$ reachable by good edges. Formally, $\widetilde{\mathcal{N}}(\boldsymbol{x}') = \{\boldsymbol{x} \in \mathcal{N}(\boldsymbol{x}') : (\boldsymbol{x}, \boldsymbol{x}') \text{ good}\}$ and $\widetilde{\mathcal{N}}(A) = \cup_{\boldsymbol{x}' \in A}\widetilde{\mathcal{N}}(\boldsymbol{x}')$. We are suppressing the dependence of $\widetilde{\mathcal{N}}$ on the classifier $f$ for notational convenience. The following lemma guarantees that if $f$ is $\eta$-robust on all points in $U$, bad edges do not account for much of the weight between $\mathcal{N}(U)$ and $U$ in the example graph (Definition 5). This is the key to our average-case-robustness results, since it implies that if the robust expansion is large and $f$ is $\eta$-robust on $U$, the neighborhood $\widetilde{\mathcal{N}}(U)$ of points reachable *by good edges* must be large.

**Lemma B.2.** *Consider an arbitrary set $U \subset \mathcal{X}$ and suppose that for all $\boldsymbol{x} \in U$, $f$ is such that $r(f, \boldsymbol{x}) \leq \eta$. I.e., $U \subset R_\eta(f)$. Then:*

$$w(\widetilde{\mathcal{N}}(U), U) \geq (1 - \eta)w(\mathcal{N}(U), U).$$

*Proof.* Since every $\boldsymbol{x}' \in U$ is in $R_\eta(f)$, we have:

$$
\begin{aligned}
\eta \geq r(f, \boldsymbol{x}') &= \mathbb{P}_{\mathbf{x}}(f(\mathbf{x}) \neq f(\boldsymbol{x}') | \mathbf{x} \in \mathcal{N}(\boldsymbol{x}')) \\
&= \frac{\mathbb{P}_{\mathbf{x}}(f(\mathbf{x}) \neq f(\boldsymbol{x}'), \mathbf{x} \in \mathcal{N}(\boldsymbol{x}'))}{\mathbb{P}(\mathbf{x} \in \mathcal{N}(\boldsymbol{x}'))} \\
&= \frac{\sum_{\boldsymbol{x} \in \mathcal{X}} \mathbb{1}[f(\boldsymbol{x}) \neq f(\boldsymbol{x}')] \mathbb{1}[\boldsymbol{x} \in \mathcal{N}(\boldsymbol{x}')] \mathbb{P}(\boldsymbol{x})}{\sum_{\boldsymbol{x} \in \mathcal{X}} \mathbb{1}[\boldsymbol{x} \in \mathcal{N}(\boldsymbol{x}')] \mathbb{P}(\boldsymbol{x})} \\
&= \frac{\sum_{\boldsymbol{x} \in \mathcal{X}} \mathbb{1}[f(\boldsymbol{x}) \neq f(\boldsymbol{x}')] \mathbb{1}[\boldsymbol{x} \in \mathcal{N}(\boldsymbol{x}')] \mathbb{P}(\boldsymbol{x}) \mathbb{P}(\boldsymbol{x}')}{\sum_{\boldsymbol{x} \in \mathcal{X}} \mathbb{1}[\boldsymbol{x} \in \mathcal{N}(\boldsymbol{x}')] \mathbb{P}(\boldsymbol{x}) \mathbb{P}(\boldsymbol{x}')} \\
&= \frac{\sum_{\boldsymbol{x} \in \mathcal{X}} \mathbb{1}[f(\boldsymbol{x}) \neq f(\boldsymbol{x}')] w(\boldsymbol{x}, \boldsymbol{x}')}{\sum_{\boldsymbol{x} \in \mathcal{X}} w(\boldsymbol{x}, \boldsymbol{x}')},
\end{aligned}
$$

so for all $\boldsymbol{x}' \in U$ we have:

$$
\sum_{\boldsymbol{x} \in \mathcal{X}} \mathbb{1}[(\boldsymbol{x}, \boldsymbol{x}') \text{ bad}] w(\boldsymbol{x}, \boldsymbol{x}') \leq \eta \sum_{\boldsymbol{x} \in \mathcal{X}} w(\boldsymbol{x}, \boldsymbol{x}').
$$

Because $w(\boldsymbol{x}, \boldsymbol{x}') > 0 \iff \boldsymbol{x} \in \mathcal{N}(\boldsymbol{x}')$, we can replace the summations over all of $\mathcal{X}$ with the sum over $\mathcal{N}(\boldsymbol{x}')$, so:

$$
\sum_{\boldsymbol{x} \in \mathcal{N}(\boldsymbol{x}')} \mathbb{1}[(\boldsymbol{x}, \boldsymbol{x}') \text{ bad}] w(\boldsymbol{x}, \boldsymbol{x}') \leq \eta \sum_{\boldsymbol{x} \in \mathcal{N}(\boldsymbol{x}')} w(\boldsymbol{x}, \boldsymbol{x}'). \tag{2}
$$

Now we can simplify:

$$
\begin{aligned}
w(\widetilde{\mathcal{N}}(U), U) &= \sum_{\boldsymbol{x}' \in U} \sum_{\boldsymbol{x} \in \widetilde{\mathcal{N}}(\boldsymbol{x}')} w(\boldsymbol{x}, \boldsymbol{x}') \\
&\geq \sum_{\boldsymbol{x}' \in U} \sum_{\boldsymbol{x} \in \mathcal{N}(\boldsymbol{x}')} \mathbb{1}[(\boldsymbol{x}, \boldsymbol{x}') \text{ good}] w(\boldsymbol{x}, \boldsymbol{x}'),
\end{aligned}
$$

where the inequality is because some elements of $\widetilde{\mathcal{N}}(U)$ might be reachable by a mixture of good and bad edges. The left-hand-side counts both, and the right-hand-side only counts the contribution of the good edges. Note that $\mathbb{1}[(\boldsymbol{x}, \boldsymbol{x}') \text{ good}] = 1 - \mathbb{1}[(\boldsymbol{x}, \boldsymbol{x}') \text{ bad}]$. Plugging this in gives:

$$
\begin{aligned}
w(\widetilde{\mathcal{N}}(U), U) &\geq \sum_{\boldsymbol{x}' \in U} \sum_{\boldsymbol{x} \in \mathcal{N}(\boldsymbol{x}')} (1 - \mathbb{1}[(\boldsymbol{x}, \boldsymbol{x}') \text{ bad}]) w(\boldsymbol{x}, \boldsymbol{x}') \\
&= \sum_{\boldsymbol{x}' \in U} \sum_{\boldsymbol{x} \in \mathcal{N}(\boldsymbol{x}')} w(\boldsymbol{x}, \boldsymbol{x}') - \sum_{\boldsymbol{x}' \in U} \sum_{\boldsymbol{x} \in \mathcal{N}(\boldsymbol{x}')} \mathbb{1}[(\boldsymbol{x}, \boldsymbol{x}') \text{ bad}] w(\boldsymbol{x}, \boldsymbol{x}') \\
&\geq \sum_{\boldsymbol{x}' \in U} \sum_{\boldsymbol{x} \in \mathcal{N}(\boldsymbol{x}')} w(\boldsymbol{x}, \boldsymbol{x}') - \eta \sum_{\boldsymbol{x}' \in U} \sum_{\boldsymbol{x} \in \mathcal{N}(\boldsymbol{x}')} w(\boldsymbol{x}, \boldsymbol{x}') \\
&= (1 - \eta) \sum_{\boldsymbol{x}' \in U} \sum_{\boldsymbol{x} \in \mathcal{N}(\boldsymbol{x}')} w(\boldsymbol{x}, \boldsymbol{x}') \\
&= (1 - \eta) w(\mathcal{N}(U), U).
\end{aligned}
$$

where we used (2) in the second inequality. $\qquad \square$

**Expanding set family.** Now we construct the set families that must expand for our results in this section. Let $\mathcal{F}$ be the hypothesis class for the strong model, $y$ the ground-truth function, and $B \subset \mathcal{X}$ an arbitrary set. For each $f \in \mathcal{F}$ let $U(B, f) = \{\boldsymbol{x} \in B : f(\boldsymbol{x}) \neq y(\boldsymbol{x})\}$ be the set of $f$'s mistakes on $y$ in $B$. Then we define the family:

$$
\mathcal{M}_\eta(B, \mathcal{F}) = \{R_\eta(f) \cap U(B, f) : f \in \mathcal{F}\}.
$$

That is, $\mathcal{M}_\eta(B, \mathcal{F})$ is the family of $\eta$-robust (Definition 2) mistake sets of $\mathcal{F}$ on the set $B$. This exactly agrees with the definition of $\mathcal{M}(B, \mathcal{F})$ from Section 4 when $\eta = 0$. Similarly, we define:

$$
\mathcal{M}'_\eta(B, \mathcal{F}) = \{R_\eta(f) \cap (B \setminus U(B, f)) : f \in \mathcal{F}\}
$$

as the family of $\eta$-robust non-mistake sets. Again, this exactly agrees with the definition of $\mathcal{M}'(B, \mathcal{F})$ from Section 4 when $\eta = 0$.

## B.1 Pseudolabel correction

We directly prove the "average-case-robust" version of Theorem 4.1. Theorem 4.1 then follows from the equivalence between expansion and robust expansion, and adversarial robustness and $\eta$-robustness, when $\eta = 0$.

**Theorem B.1** (Pseudolabel correction). *Suppose $M'_\eta(S_i^{good}, \mathcal{F})$ satisfies $(c, q, \eta)$-robust expansion on $(S_i^{bad}, S_i^{good})$ for some $c > 0$ and $\eta \geq 0$. Consider an arbitrary classifier $f$ such that $\mathbb{P}(f(\mathbf{x}) \neq \tilde{y}(\mathbf{x})$ or $f$ not $\eta$-robust at $\mathbf{x}|S_i) \leq 1 - q - \alpha_i$. Then $f$ satisfies the following error bound:*

$$err(f, y|S_i) \leq \frac{err(f, \tilde{y}|S_i) - \alpha_i(2c' - 1) + 2c'\alpha_i \mathbb{P}(\overline{R_\eta(f)}|S_i)}{1 - 2c'\alpha_i}$$

*where $c' = c/(1 - \alpha_i + c\alpha_i)$.*

**Corollary B.1** (Simplified bound, average-case-robust version of Theorem 4.1). *Suppose the conditions of Theorem B.1 hold. For any $\Delta$ such that $err(f, \tilde{y}|S_i) \leq c\alpha_i \Delta + (1 - \alpha_i)(1 - \Delta)$, the error of $f$ satisfies:*

$$err(f, y|S_i) \leq \frac{2\alpha_i}{1 - 2\alpha_i}\mathbb{P}(\overline{R_\eta(f)}|S_i) + err(f, \tilde{y}|S_i) + \alpha_i(1 - 2c\Delta).$$

*Moreover, if $f$ satisfies*

$$\mathbb{E}_{\mathbf{x} \sim \mathcal{D}|S_i, \mathbf{x}' \sim \mathcal{D}|\mathcal{N}(\mathbf{x})}[f(\mathbf{x}) \neq f(\mathbf{x}')] \leq \gamma,$$

*then*

$$err(f, y|S_i) \leq \frac{2\alpha_i}{1 - 2\alpha_i}\frac{\gamma}{\eta} + err(f, \tilde{y}|S_i) + \alpha_i(1 - 2c\Delta).$$

*Proof.* The first part directly follows from Theorem B.1 by substituting the value of $c'$, using $c \leq 1$, and using the form of $\Delta$ to simplify the bound. Theorem 4.1 follows from taking $\Delta = 3/4$, which was small enough for the error condition to hold empirically. The condition on $q$ in Theorem 4.1 implies that if $\mathbb{P}(f(\mathbf{x}) \neq \tilde{y}(\mathbf{x})$ or $f$ not $\eta$-robust at $\mathbf{x}|S_i) \leq \frac{1 - \alpha_i + 3c\alpha_i}{4}$, then

$$\mathbb{P}(f(\mathbf{x}) \neq \tilde{y}(\mathbf{x}) \text{ or } f \text{ not } \eta\text{-robust at } \mathbf{x}|S_i) \leq 1 - q - \alpha_i,$$

so the conditions of Theorem B.1 hold. This upper bound on $q$ could be replaced by instead assuming

$$\mathbb{P}(f \neq \tilde{y}(\mathbf{x}) \text{ or } f \text{ not } \eta\text{-robust at } \mathbf{x}|S_i) \leq \min\left(1 - q - \alpha_i, \frac{1 - \alpha_i + 3c\alpha_i}{4}\right).$$

The second part follows directly from Lemma B.1. $\qquad\square$

*Proof of Theorem B.1.* Let $M_i = \{\boldsymbol{x} \in S_i : f(\boldsymbol{x}) \neq y(\boldsymbol{x})\}$ be the set of mistakes of $f$ on the *true* labels in $S_i$. Similarly, let $D_i = \{\boldsymbol{x} \in S_i : f(\boldsymbol{x}) \neq \tilde{y}(\boldsymbol{x})\}$ be the set of mistakes of $f$ on the *weak* labels in $S_i$. Define $U_i = S_i \setminus M_i$ and let $V_i = R_\eta(f) \cap U_i \cap S_i^{good}$. Note that $V_i \in \mathcal{M}'(S_i^{good}, \mathcal{F})$, so if it's large enough, it expands according to our expansion assumption. The following lemma shows that $V_i$ is large enough to expand.

**Lemma B.3.** $\mathbb{P}(V_i|S_i^{good}) > q$.

*Proof.* Suppose for a contradiction that $\mathbb{P}(V_i|S_i^{good}) \leq q$. Then by definition of $V_i$,

$$\begin{aligned}
\mathbb{P}(V_i|S_i^{good}) &= 1 - \mathbb{P}(\overline{V_i}|S_i^{good}) \\
&= 1 - \mathbb{P}(\overline{V_i} \cap S_i^{good}|S_i^{good}) \\
&= 1 - \mathbb{P}(\overline{((S_i \setminus M_i) \cap R_\eta(f))} \cap S_i^{good}|S_i^{good}) \\
&= 1 - \mathbb{P}((M_i \cap S_i^{good}) \cup \overline{R_\eta(f)}|S_i^{good})
\end{aligned}$$

Fix an arbitrary $\boldsymbol{x} \in M_i \cap S_i^{good}$. By definition of $M_i$, $f(\boldsymbol{x}) \neq y(\boldsymbol{x})$. By definition of $S_i^{good}$, $\tilde{y}(\boldsymbol{x}) = y(\boldsymbol{x})$. Hence $f(\boldsymbol{x}) \neq \tilde{y}(\boldsymbol{x})$, so $\boldsymbol{x} \in D_i$, and therefore $M_i \cap S_i^{good} \subset D_i$. Then:

$$
\begin{aligned}
q \geq \mathbb{P}(V_i | S_i^{good}) &\geq 1 - \mathbb{P}(D_i \cup \overline{R_\eta(f)} | S_i^{good}) \\
&= 1 - (1 - \alpha_i) \mathbb{P}((D_i \cup \overline{R_\eta(f)}) \cap S_i^{good} | S_i) \\
&\geq 1 - (1 - \alpha_i) \mathbb{P}((D_i \cup \overline{R_\eta(f)}) | S_i) \\
&= 1 - (1 - \alpha_i) \mathbb{P}(f(\mathbf{x}) \neq \tilde{y}(\mathbf{x}) \text{ or } f \text{ not } \eta\text{-robust at } \mathbf{x} | S_i) \\
&\geq 1 - (1 - \alpha_i)(1 - q - \alpha_i) \\
&= q + \alpha_i(2 - q - \alpha_i) \\
&> q
\end{aligned}
$$

The second-to-last inequality used the assumption on $\mathbb{P}(f(\mathbf{x}) \neq \tilde{y}(\mathbf{x}) \text{ or } f \text{ not } \eta\text{-robust at } \mathbf{x} | S_i)$ and the final inequality used $q < 1$ and $0 < \alpha_i < 1/2$. Assuming $\mathbb{P}(V_i | S_i^{good}) \leq q$ thus leads to $q > q$, a contradiction. So $\mathbb{P}(V_i | S_i^{good}) > q$. $\qquad\square$

Recall that for a set $A \subset \mathcal{X}$, we define $\widetilde{\mathcal{N}}(A) \subset \mathcal{N}(A)$ as the subset of points reachable from $A$ by a *good edge* $f(\boldsymbol{x}) = f(\boldsymbol{x}')$. By Lemma B.2, since $V_i \subset R_\eta(f)$,

$$
w(\widetilde{\mathcal{N}}(V_i), V_i) \geq (1 - \eta) w(\mathcal{N}(V_i), V_i).
$$

Then by Lemma B.3, since $(S_i^{bad}, S_i^{good})$ satisfy $(c, q, \eta)$-robust expansion,

$$
\mathbb{P}(\widetilde{\mathcal{N}}(V_i) | S_i^{bad}) \geq P_{1-\eta}(V_i, S_i^{bad}) \geq c \mathbb{P}(V_i | S_i^{good}).
$$

Fix an arbitrary $\boldsymbol{x} \in \widetilde{\mathcal{N}}(V_i)$. By definition of $\widetilde{\mathcal{N}}$, there exists $\boldsymbol{x}' \in V_i$ such that $f(\boldsymbol{x}) = f(\boldsymbol{x}')$. Since $\boldsymbol{x}' \in V_i \subset U_i$, we have that $f(\boldsymbol{x}) = f(\boldsymbol{x}') = y(\boldsymbol{x}')$ (since $\boldsymbol{x}' \notin M_i$), and $y(\boldsymbol{x}') = y(\boldsymbol{x})$, since $\boldsymbol{x}$ and $\boldsymbol{x}'$ are both in $S_i$. This shows $f(\boldsymbol{x}) = y(\boldsymbol{x})$ and therefore $\boldsymbol{x} \notin M_i$, so $\boldsymbol{x} \in U_i = S_i \setminus M_i$. Since $\boldsymbol{x}$ was arbitrary, $\widetilde{\mathcal{N}}(V_i) \subset U_i$. Thus,

$$
\mathbb{P}(U_i | S_i^{bad}) \geq c \mathbb{P}(V_i | S_i^{good}). \tag{3}
$$

Points $\boldsymbol{x} \in U_i$ are correctly labeled by $f$, and points $\boldsymbol{x} \in S_i^{bad}$ are *incorrectly* labeled by the pseudolabeler $\tilde{y}$. This inequality thus guarantees that there must be some points that $f$ gets correct and $\tilde{y}$ gets incorrect. It also gives a quantitative lower bound on how much probability is assigned to these points. Intuitively, (3) is already a "pseudolabel correction" result, and the remainder of the proof is deriving a final bound on the error from (3).

The following lemma converts (3) into a relationship between $\mathbb{P}(U_i | S_i^{bad})$ and $\mathbb{P}(U_i \cap R_\eta(f) | S_i)$.

**Lemma B.4.** $\mathbb{P}(U_i | S_i^{bad}) \geq c' \mathbb{P}(U_i \cap R_\eta(f) | S_i) + c' \alpha_i \mathbb{P}(U_i \cap \overline{R_\eta(f)} | S_i^{bad})$, where $c' = c/(1 - \alpha_i + c\alpha_i)$.

*Proof.*

$$
\begin{aligned}
\mathbb{P}(U_i | S_i^{bad}) &\geq c \mathbb{P}(V_i | S_i^{good}) \qquad \text{(by (3))} \\
&= \frac{c}{1 - \alpha_i} \mathbb{P}(V_i \cap S_i^{good} | S_i) \\
&= \frac{c}{1 - \alpha_i} \mathbb{P}(U_i \cap R_\eta(f) \cap S_i^{good} | S_i) \\
&= \frac{c}{1 - \alpha_i} \left( \mathbb{P}(U_i \cap R_\eta(f) | S_i) - \mathbb{P}(U_i \cap R_\eta(f) \cap S_i^{bad} | S_i) \right)
\end{aligned}
$$

Rearranging,

$$
\begin{aligned}
\frac{c}{1 - \alpha_i} \mathbb{P}(U_i \cap R_\eta(f) | S_i) &\leq \mathbb{P}(U_i | S_i^{bad}) + \frac{c}{1 - \alpha_i} \mathbb{P}(U_i \cap R_\eta(f) \cap S_i^{bad} | S_i) \\
&= \mathbb{P}(U_i | S_i^{bad}) + \frac{c\alpha_i}{1 - \alpha_i} \mathbb{P}(U_i \cap R_\eta(f) | S_i^{bad}) \\
&= \mathbb{P}(U_i | S_i^{bad}) + \frac{c\alpha_i}{1 - \alpha_i} \left( \mathbb{P}(U_i | S_i^{bad}) - \mathbb{P}(U_i \cap \overline{R_\eta(f)} | S_i^{bad}) \right) \\
&= \mathbb{P}(U_i | S_i^{bad}) \left( 1 + \frac{c\alpha_i}{1 - \alpha_i} \right) - \frac{c\alpha_i}{1 - \alpha_i} \mathbb{P}(U_i \cap \overline{R_\eta(f)} | S_i^{bad})
\end{aligned}
$$

And finally,

$$\frac{c}{1 - \alpha_i + c\alpha_i}\mathbb{P}(U_i \cap R_\eta(f)|S_i) + \frac{c\alpha_i}{1 - \alpha_i + c\alpha_i}\mathbb{P}(U_i \cap \overline{R_\eta(f)}|S_i^{bad}) \leq \mathbb{P}(U_i|S_i^{bad}).$$

$\square$

The following lemma decomposes the disagreement set $D_i$ into its components in $S_i^{good}$ and $S_i^{bad}$.

**Lemma B.5.** $D_i \supset (M_i \cap S_i^{good}) \cup (U_i \cap S_i^{bad})$.

*Proof.* First, fix $\boldsymbol{x} \in M_i \cap S_i^{good}$. Since $\boldsymbol{x} \in M_i$, $f(\boldsymbol{x}) \neq y(\boldsymbol{x})$. Since $\boldsymbol{x} \in S_i^{good}$, $\tilde{y}(\boldsymbol{x}) = y(\boldsymbol{x})$. Thus $f(\boldsymbol{x}) \neq \tilde{y}(\boldsymbol{x})$, so $\boldsymbol{x} \in D_i$. Second, fix $\boldsymbol{x} \in U_i \cap S_i^{bad}$. Since $\boldsymbol{x} \in U_i = S_i \setminus M_i$, $f(\boldsymbol{x}) = y(\boldsymbol{x})$. Since $\boldsymbol{x} \in S_i^{bad}$, $\tilde{y}(\boldsymbol{x}) \neq y(\boldsymbol{x})$. Thus $f(\boldsymbol{x}) \neq \tilde{y}(\boldsymbol{x})$, so $\boldsymbol{x} \in D_i$. $\square$

By Lemma B.5 and using $\mathbb{P}(S_i^{good}|S_i) = 1 - \mathbb{P}(S_i^{bad}|S_i) = 1 - \alpha_i$,

$$\mathbb{P}(D_i|S_i) \geq \mathbb{P}(M_i|S_i^{good})(1 - \alpha_i) + \mathbb{P}(U_i|S_i^{bad})\alpha_i$$

Applying Lemma B.4,

$$\mathbb{P}(D_i|S_i) \geq (1 - \alpha_i)\mathbb{P}(M_i|S_i^{good}) + c'\alpha_i\left(\mathbb{P}(U_i \cap R_\eta(f)|S_i) + \alpha_i\mathbb{P}(U_i \cap \overline{R_\eta(f)}|S_i^{bad})\right). \quad (4)$$

Applying Lemma B.4 again,

$$\mathbb{P}(U_i|S_i) = \alpha_i\mathbb{P}(U_i|S_i^{bad}) + (1 - \alpha_i)\mathbb{P}(U_i|S_i^{good})$$
$$\geq c'\alpha_i\left(\mathbb{P}(U_i \cap R_\eta(f)|S_i) + \alpha_i\mathbb{P}(U_i \cap \overline{R_\eta(f)}|S_i^{bad})\right) + (1 - \alpha_i)\mathbb{P}(U_i|S_i^{good}),$$

so we have

$$\mathbb{P}(U_i|S_i^{good}) \leq \frac{1}{1 - \alpha_i}\left(\mathbb{P}(U_i|S_i) - c'\alpha_i\left(\mathbb{P}(U_i \cap R_\eta(f)|S_i) + \alpha_i\mathbb{P}(U_i \cap \overline{R_\eta(f)}|S_i^{bad})\right)\right)$$

Combining this with $U_i = S_i \setminus M_i$,

$$\mathbb{P}(M_i|S_i^{good}) = 1 - \mathbb{P}(U_i|S_i^{good})$$
$$\geq 1 - \frac{1}{1 - \alpha_i}\left(\mathbb{P}(U_i|S_i) - c'\alpha_i\left(\mathbb{P}(U_i \cap R_\eta(f)|S_i) + \alpha_i\mathbb{P}(U_i \cap \overline{R_\eta(f)}|S_i^{bad})\right)\right)$$

Plugging this into (4) gives:

$$\mathbb{P}(D_i|S_i) \geq (1 - \alpha_i) - \mathbb{P}(U_i|S_i) + 2c'\alpha_i\left(\mathbb{P}(U_i \cap R_\eta(f)|S_i) + \alpha_i\mathbb{P}(U_i \cap \overline{R_\eta(f)}|S_i^{bad})\right)$$
$$\geq (1 - \alpha_i) - (1 - \mathbb{P}(M_i|S_i)) + 2c'\alpha_i\left(1 - \mathbb{P}(M_i \cup \overline{R_\eta(f)}|S_i) + \mathbb{P}(U_i \cap \overline{R_\eta(f)} \cap S_i^{bad}|S_i)\right)$$
$$= \mathbb{P}(M_i|S_i) + \alpha_i(2c' - 1) + 2c'\alpha_i\left(\mathbb{P}(U_i \cap \overline{R_\eta(f)} \cap S_i^{bad}|S_i) - \mathbb{P}(M_i \cup \overline{R_\eta(f)}|S_i)\right)$$
$$\geq \mathbb{P}(M_i|S_i) + \alpha_i(2c' - 1) - 2c'\alpha_i\mathbb{P}(M_i \cup \overline{R_\eta(f)}|S_i)$$

Let $p = \mathbb{P}(M_i|S_i)$. Then, using the union bound for the last term,

$$\mathbb{P}(D_i|S_i) \geq p + \alpha_i(2c' - 1) - 2c'\alpha_i(p + \mathbb{P}(\overline{R_\eta(f)}|S_i))$$
$$= (1 - 2c'\alpha_i)p + \alpha_i(2c' - 1) - 2c'\alpha_i\mathbb{P}(\overline{R_\eta(f)}|S_i)$$

Since $\mathbb{P}(D_i|S_i) = \text{err}(f, \tilde{y}|S_i)$, rearranging terms we get:

$$\text{err}(f, y|S_i) = \mathbb{P}(M_i|S_i) \leq \frac{\text{err}(f, \tilde{y}|S_i) - \alpha_i(2c' - 1) + 2c'\alpha_i\mathbb{P}(\overline{R_\eta(f)}|S_i)}{1 - 2c'\alpha_i}.$$

$\square$

## B.2 Coverage expansion

We directly prove the "average-case-robust" version of Theorem 4.2. Theorem 4.2 then follows from the equivalence between expansion and robust expansion, and adversarial robustness and $\eta$-robustness, when $\eta = 0$. The following theorem also allows for $T_i$ to expand to $S_i^{good}$ and $S_i^{bad}$ in different amounts, controlled by the two expansion parameters $c_1$ and $c_2$. This allows for empirically tighter bounds and is used in our experiments in Section 6. Theorem 4.2 is a special case of Theorem B.2 taking $\eta = 0$ and $c_1 = c_2$.

**Theorem B.2** (Average-case-robust version of Theorem 4.2). *Suppose there exists $c_1, c_2, q, \eta > 0$ such that $M_\eta(T_i, \mathcal{F}, y)$ satisfies $(c_1, q, \eta)$-expansion on $(S_i^{good}, T_i)$ and $M'_\eta(T_i, \mathcal{F}, y)$ satisfies $(c_2, q, \eta)$-expansion on $(S_i^{bad}, T_i)$. Fix an arbitrary classifier $f : \mathcal{X} \to \mathcal{Y}$ that satisfies:*

$$err(f, \tilde{y}|S_i) + \mathbb{P}(\overline{R_\eta(f)}|T_i) < c_1(1 - q - \alpha_i)$$

*Let $\bar{c} = \max\{c_1, c_2\}$ and $\underline{c} = \min\{c_1, c_2\}$. Then the error of $f$ on $T_i$ is at most:*

$$err(f, y|T_i) \le \left(1 + \frac{\alpha_i}{\underline{c}/\bar{c} - 2\alpha_i}\right) \mathbb{P}(R_\eta(f)|T_i) + \max\left(q, \frac{err(g, \tilde{y}|S_i) - c_2\alpha_i}{c_1 - (c_1 + c_2)\alpha_i}\right).$$

*Moreover, if $f$ satisfies*

$$\mathbb{E}_{\mathbf{x} \sim \mathcal{D}|T_i, \mathbf{x}' \sim \mathcal{D}|\mathcal{N}(\mathbf{x})}[f(\mathbf{x}) \ne f(\mathbf{x}')] \le \gamma,$$

*then the error of $f$ on $T_i$ is bounded by:*

$$err(f, y|T_i) \le \left(1 + \frac{\alpha_i}{\underline{c}/\bar{c} - 2\alpha_i}\right) \frac{\gamma}{\eta} + \max\left(q, \frac{err(g, \tilde{y}|S_i) - c_2\alpha_i}{c_1 - (c_1 + c_2)\alpha_i}\right).$$

*Proof.* Let $M_i = \{\mathbf{x} \in T_i : f(\mathbf{x}) \ne y(\mathbf{x})\}$ and $C_i = T_i \setminus M_i = \{\mathbf{x} \in T_i : f(\mathbf{x}) = y(\mathbf{x})\}$. Define their robust subsets as $U_i = M_i \cap R_\eta(f)$ and $V_i = C_i \cap R_\eta(f)$. We have $U_i \in \mathcal{M}_\eta(T_i, \mathcal{F})$ and $V_i \in \mathcal{M}'_\eta(T_i, \mathcal{F})$. Recall that for a set $A$, $\widetilde{\mathcal{N}}(A)$ is the subset of $\mathcal{N}(A)$ consisting of neighbors reachable by good edges (edges where $f$ assigns the same label to both endpoints). Since $U_i \subset R_\eta(f)$, Lemma B.2 implies that $\widetilde{\mathcal{N}}(U_i)$ has enough weight to qualify as one of the sets in the definition of the robust neighborhood size $P_{1-\eta}(U, S_i^{good})$. In particular, it implies that $\mathbb{P}(\widetilde{\mathcal{N}}(U_i)|S_i^{good}) \ge P_{1-\eta}(U_i, S_i^{good})$. Similarly, $\widetilde{\mathcal{N}}(V_i)$ satisfies $\mathbb{P}(\widetilde{\mathcal{N}}(V_i)|S_i^{bad}) \ge P_{1-\eta}(V_i, S_i^{bad})$. There are now three cases to consider:

**Case 1:** $\mathbb{P}(U_i|T_i) > q$, $\mathbb{P}(V_i|T_i) > q$. In this case, the expansion assumptions imply:

$$\mathbb{P}(\widetilde{\mathcal{N}}(U_i)|S_i^{good}) \ge c_1\mathbb{P}(U_i|T_i)$$

$$\mathbb{P}(\widetilde{\mathcal{N}}(V_i)|S_i^{bad}) \ge c_2\mathbb{P}(V_i|T_i)$$

For all $\mathbf{x} \in \widetilde{\mathcal{N}}(U_i) \cap S_i^{good}$ there exists $\mathbf{x}' \in U_i$ with $f(\mathbf{x}) = f(\mathbf{x}')$. Then:

$$f(\mathbf{x}) = \underbrace{f(\mathbf{x}') \ne y(\mathbf{x}')}_{\mathbf{x}' \in U_i} \overbrace{= y(\mathbf{x})}^{\text{Def. of } S_i, T_i} = \underbrace{\tilde{y}(\mathbf{x})}_{\mathbf{x} \in S_i^{good}},$$

so $f(\mathbf{x}) \ne \tilde{y}(\mathbf{x})$. Similarly, for all $\mathbf{x} \in \widetilde{\mathcal{N}}(V_i) \cap S_i^{bad}$ there exists $\mathbf{x}' \in V_i$ with $f(\mathbf{x}) = f(\mathbf{x}')$. Then

$$f(\mathbf{x}) = \underbrace{f(\mathbf{x}') = y(\mathbf{x}')}_{\mathbf{x}' \in V_i} \overbrace{= y(\mathbf{x})}^{\text{Def. of } S_i, T_i} = \underbrace{\tilde{y}(\mathbf{x})}_{\mathbf{x} \in S_i^{bad}} \ne,$$

so $f(\mathbf{x}) \ne \tilde{y}(\mathbf{x})$.

Let $D_i = \{\mathbf{x} \in S_i : f(\mathbf{x}) \ne \tilde{y}(\mathbf{x})\}$ be the set of points in $S_i$ where $f$ and the teacher $\tilde{y}$ disagree. These arguments showed that $\widetilde{\mathcal{N}}(U_i) \cap S_i^{good} \subset D_i$ and $\widetilde{\mathcal{N}}(V_i) \cap S_i^{bad} \subset D_i$. Since these sets are disjoint,

$$\mathbb{P}(D_i|S_i) \ge \mathbb{P}(\widetilde{\mathcal{N}}(U_i) \cap S_i^{good}|S_i) + \mathbb{P}(\widetilde{\mathcal{N}}(U_i) \cap S_i^{bad}|S_i)$$
$$= \mathbb{P}(\widetilde{\mathcal{N}}(U_i)|S_i^{good})(1 - \alpha_i) + \mathbb{P}(\widetilde{\mathcal{N}}(U_i)|S_i^{bad})\alpha_i$$
$$\ge c_1\mathbb{P}(U_i|T_i)(1 - \alpha_i) + c_2\mathbb{P}(V_i|T_i)\alpha_i.$$

Since $U_i \cup V_i = R_\eta(f) \cap T_i$ and $U_i$ and $V_i$ are disjoint subsets of $T_i$,
$$\mathbb{P}(V_i|T_i) = \mathbb{P}(R_\eta(f)|T_i) - \mathbb{P}(U_i|T_i).$$
Plugging this into the previous inequality and simplifying gives:
$$\mathbb{P}(U_i|T_i) \leq \frac{\mathbb{P}(D_i|S_i) - c_2\alpha_i\mathbb{P}(R_\eta(f)|T_i)}{c_1 - (c_1 + c_2)\alpha_i} = \frac{\mathbb{P}(D_i|S_i) - c_2\alpha_i + c_2\alpha_i\mathbb{P}(\overline{R_\eta(f)}|T_i)}{c_1 - (c_1 + c_2)\alpha_i}$$
$$\leq \frac{\mathbb{P}(D_i|S_i) - c_2\alpha_i}{c_1 - (c_1 + c_2)\alpha_i} + \frac{\alpha_i}{\underline{c}/\bar{c} - 2\alpha_i}\mathbb{P}(\overline{R_\eta(f)}|T_i).$$
We can then bound $\mathbb{P}(M_i|T_i) = \mathbb{P}(U_i|T_i) + \mathbb{P}(M_i \cap \overline{R_\eta(f)}|T_i) \leq \mathbb{P}(U_i|T_i) + \mathbb{P}(\overline{R_\eta(f)}|T_i)$, so all combined, we have:
$$\mathbb{P}(M_i|T_i) \leq \left(1 + \frac{\alpha_i}{\underline{c}/\bar{c} - 2\alpha_i}\right)\mathbb{P}(\overline{R_\eta(f)}|T_i) + \frac{\mathbb{P}(D_i|S_i) - c_2\alpha_i}{c_1 - (c_1 + c_2)\alpha_i}$$

**Case 2:** $\mathbb{P}(U_i|T_i) \leq q$. Here we directly upper-bound $\mathbb{P}(U_i|T_i)$ with $q$ and $\mathbb{P}(M_i|T_i) \leq \mathbb{P}(U_i|T_i) + \mathbb{P}(\overline{R_\eta(f)}|T_i) \leq q + \mathbb{P}(\overline{R_\eta(f)}|T_i)$.

**Case 3:** $\mathbb{P}(U_i|T_i) > q$, $\mathbb{P}(V_i|T_i) \leq q$. Since $\mathbb{P}(U_i|T_i) + \mathbb{P}(V_i|T_i) = \mathbb{P}(R_\eta(f)|T_i)$, in this case we have $\mathbb{P}(U_i|T_i) \geq \mathbb{P}(R_\eta(f)|T_i) - q$. Since $\mathbb{P}(U_i|T_i) > q$, $U_i$ expands, so as in Case 1 we have:
$$\mathbb{P}(\widetilde{\mathcal{N}}(U_i)|S_i^{good}) \geq c_1\mathbb{P}(U_i|T_i),$$
and therefore:
$$\mathbb{P}(D_i|S_i) \geq \mathbb{P}(\widetilde{\mathcal{N}}(U_i)|S_i^{good})(1-\alpha_i) \geq c_1(1-\alpha_i)\mathbb{P}(U_i|T_i) \geq c_1(1-\alpha_i)(\mathbb{P}(R_\eta(f)|T_i) - q).$$
But $\mathbb{P}(D_i|S_i) = \text{err}(g, \tilde{y}|S_i)$ and we assumed:
$$\text{err}(g, \tilde{y}|S_i) + \mathbb{P}(\overline{R_\eta(f)}|T_i) < c_1(1 - q - \alpha_i) \leq c_1(1-q)(1-\alpha_i),$$
since $c_1 \leq 1$, this implies
$$\text{err}(g, \tilde{y}|S_i) + c_1(1-\alpha_i)\mathbb{P}(\overline{R_\eta(f)}|T_i) < c_1(1-q)(1-\alpha_i),$$
so:
$$\text{err}(g, \tilde{y}|S_i) < c_1(1 - \mathbb{P}(\overline{R_\eta(f)}|T_i) - q)(1-\alpha_i)$$
$$= c_1(\mathbb{P}(R_\eta(f)|T_i) - q)(1-\alpha_i).$$
So Case 3 leads to a contradiction. Combining Cases 1 and 2 yields the final bound. The second part follows directly from Lemma B.1. $\qquad \square$

If $c_1 = c_2 \equiv c$, we have $\underline{c}/\bar{c} = 1$ and $c_1 - (c_1 + c_2)\alpha_i = c(1 - 2\alpha_i)$, which gives a simpler bound for coverage expansion of average-case-robust classifiers. When $\eta = 0$, $\mathbb{P}(R_\eta(f)|T_i) := \mathbb{P}(R(f)|T_i)$ by definition, and $(c, q, \eta)$-expansion is equivalent to $(c, q)$-expansion. These two simplifications directly yield Theorem 4.2 for coverage expansion of adversarially robust classifiers.

The following theorem gives a more loose bound that only assumes expansion from $T_i$ to $S_i^{good}$.

**Theorem B.3** (Coverage expansion, weaker assumptions). *Suppose $\mathcal{M}_\eta(T_i, \mathcal{F})$ satisfies $(c, q, \eta)$-robust expansion on $(S_i^{good}, T_i)$ for some $c > 0$. Fix an arbitrary classifier $f : \mathcal{X} \to \mathcal{Y}$. The error of $f$ on $T_i$ is bounded by:*
$$\text{err}(f, y|T_i) \leq \mathbb{P}(\overline{R_\eta(f)}|T_i) + \max\left(q, \frac{\text{err}(f, \tilde{y}|S_i)}{c(1-\alpha_i)}\right).$$

*Proof.* Define $M_i = \{\boldsymbol{x} : f(\boldsymbol{x}) \neq y(\boldsymbol{x})\} \cap T_i$ as the set of mistakes of $f$ in $T_i$, and let $U_i = M_i \cap R_\eta(f)$. Let $D_i = \{\boldsymbol{x} \in S_i : f(\boldsymbol{x}) \neq \tilde{y}(\boldsymbol{x})\}$ be the set of points in $S_i$ where $f$ disagrees with the weak labels. Note that $\text{err}(f, \tilde{y}|S_i) = \mathbb{P}(D_i|S_i)$ and $\text{err}(f, y|T_i) = \mathbb{P}(M_i|T_i) \leq \mathbb{P}(U_i|T_i) + \mathbb{P}(\overline{R_\eta(f)}|T_i)$. So the goal is to bound $\mathbb{P}(U_i|T_i)$. Recall that $\widetilde{\mathcal{N}}(U)$ is the subset of $\mathcal{N}(U)$ consisting of neighbors reachable by good edges (edges where $f$ assigns the same label to both endpoints). Since $U_i \subset R_\eta(f)$, Lemma B.2 implies that $\widetilde{\mathcal{N}}(U_i)$ has enough weight to qualify as one of the sets $V$ in the definition of the robust neighborhood size $P_{1-\eta}(U, S_i^{good})$. In particular, it implies that $\mathbb{P}(\widetilde{\mathcal{N}}(U_i)|S_i^{good}) \geq P_{1-\eta}(U, S_i^{good})$. Note that $U_i \in \mathcal{M}_\eta(T_i, \mathcal{F})$ since it is a mistake set intersected with a robust set. Then, if $\mathbb{P}(U_i|T_i) > q$, $(c, q, \eta)$-robust expansion implies that
$$\mathbb{P}(\widetilde{\mathcal{N}}(U_i)|S_i^{good}) \geq P_{1-\eta}(U, S_i^{good}) > c\mathbb{P}(U_i|T_i).$$
We now proceed in two cases.

**Case 1:** $\mathbb{P}(U_i|T_i) > q$. In this case, because $(S_i^{good}, T_i)$ satisfy $(c, q, \eta)$-robust expansion, as discussed above, we know that $\mathbb{P}(\widetilde{\mathcal{N}}(U_i)|S_i^{good}) \geq c\mathbb{P}(U_i|T_i)$, and thus

$$\mathbb{P}(\widetilde{\mathcal{N}}(U_i) \cap S_i^{good}|S_i) \geq c(1 - \alpha_i)\mathbb{P}(U_i|T_i).$$

Fix $\boldsymbol{x} \in \widetilde{\mathcal{N}}(U_i) \cap S_i^{good}$. By the definition of $\widetilde{\mathcal{N}}(U_i)$, there must be a point $\boldsymbol{x}' \in U_i$ reachable from $\boldsymbol{x}$ by a good edge. That is, there is a point $\boldsymbol{x}' \in U_i$ such that $f(\boldsymbol{x}) = f(\boldsymbol{x}')$. Then the following set of inequalities holds:

$$\overbrace{\tilde{y}(\boldsymbol{x}) = \underbrace{y(\boldsymbol{x}) = y(\boldsymbol{x}')}_{\text{Defn of } S_i, T_i}}^{\boldsymbol{x} \in S_i^{good}} \overbrace{\neq \underbrace{f(\boldsymbol{x}') = f(\boldsymbol{x})}_{\text{Defn of } \widetilde{\mathcal{N}}}}^{\boldsymbol{x}' \in M_i}.$$

This shows $f(\boldsymbol{x}) \neq \tilde{y}(\boldsymbol{x})$, so $\boldsymbol{x} \in D_i$. Hence $\widetilde{\mathcal{N}}(U_i) \cap S_i^{good} \subset D_i$ and we have:

$$\text{err}(f, \tilde{y}|S_i) = \mathbb{P}(D_i|S_i) \geq \mathbb{P}(\widetilde{\mathcal{N}}(U_i) \cap S_i^{good}|S_i) \geq c(1 - \alpha_i)\mathbb{P}(U_i|T_i),$$

so

$$\text{err}(f, y|T_i) \leq \mathbb{P}(\overline{R_\eta(f)}|T_i) + \frac{\text{err}(f, \tilde{y}|S_i)}{c(1 - \alpha_i)}.$$

**Case 2:** $\mathbb{P}(U_i|T_i) \leq q$. In this case we directly use the bound $\mathbb{P}(U_i|T_i) \leq q$, so we get $\text{err}(g, y|T_i) \leq \mathbb{P}(\overline{R_\eta(f)}|T_i) + q$. Combining the two cases yields the theorem. $\qquad\square$

### B.3 Proposition 3.1

To prove Proposition 3.1, we reproduce (a tighter, intermediate version of) Theorem 3 from Fu et al. [21] and translate it to our notation, then simplify the bound in the setting of the example from Section 3.

The goal of Fu et al. [21] is to estimate a set of graphical model parameters $\boldsymbol{\mu}$ such that, given a vector of labeling function outputs $\boldsymbol{\lambda}(\boldsymbol{x})$, the graphical model $\mathbb{P}_{\boldsymbol{\mu}}(\mathbf{y}|\boldsymbol{\lambda}(\mathbf{x}))$ approximates $\mathbb{P}(\mathbf{y}|\mathbf{x})$. A classifier is then trained on weak labels sampled from $\mathbb{P}_{\boldsymbol{\mu}}(\mathbf{y}|\boldsymbol{\lambda}(\mathbf{x}))$. Define $\text{err}(f) = \mathbb{P}_{(\mathbf{x},\mathbf{y})\sim\mathcal{D}}[f(\mathbf{x}) \neq \mathbf{y}]$ and $\text{err}_{\boldsymbol{\mu}}(f) = \mathbb{E}_{(\mathbf{x},\mathbf{y})\sim\mathcal{D}}[\mathbb{P}_{\tilde{\mathbf{y}}\sim\mathbb{P}_{\boldsymbol{\mu}}(\cdot|\boldsymbol{\lambda}(\mathbf{x}))}[f(\mathbf{x}) \neq \tilde{\mathbf{y}}]]$.

**Theorem B.4.** *Fu et al. [21, Theorem 3, intermediate step, p.39] Let $f : \mathcal{X} \to \mathcal{Y}$ be an arbitrary classifier. Then:*

$$err(f) \leq err_{\boldsymbol{\mu}}(f) + \mathbb{E}_{(\mathbf{x}',\mathbf{y}')\sim\mathcal{D}}\left[\sum_y |\mathbb{P}(\mathbf{y} = y|\mathbf{x} = \mathbf{x}') - \mathbb{P}_{\boldsymbol{\mu}}(\mathbf{y} = y|\boldsymbol{\lambda}(\mathbf{x}) = \lambda(\mathbf{x}'))|\right]. \quad (5)$$

The remainder of the theorem accounts for finite-sample issues and estimation error for the optimal graphical model parameters $\boldsymbol{\mu}$, but this is the fundamental relationship proven between a classifier's clean error and weak-label error. The second term in the RHS of (5) is exactly twice the total variation distance between $\mathbb{P}(\mathbf{y}|\mathbf{x})$ and $\mathbb{P}_{\boldsymbol{\mu}}(\mathbf{y}|\boldsymbol{\lambda}(\mathbf{x}))$. For the final theorem statement, Pinsker's inequality is used to upper bound this term using the KL divergence between the true probability $\mathbb{P}(\mathbf{y}|\mathbf{x})$ and the estimated graphical model probability $\mathbb{P}_{\boldsymbol{\mu}}(\mathbf{y}|\boldsymbol{\lambda}(\mathbf{x}))$, but the TV version is tighter, so we proceed by analyzing the intermediate bound (5).

In the example from Section 3, there is only one labeling function:

$$\lambda(\boldsymbol{x}) = \tilde{y}(\boldsymbol{x}) = \begin{cases} 1 & \texttt{good} \in \boldsymbol{x} \\ 0 & \texttt{bad} \in \boldsymbol{x} \\ \varnothing & \text{otherwise,} \end{cases}$$

and we have assumed that the true label $\mathbf{y}$ is a deterministic function of $\mathbf{x}$, i.e., $\mathbf{y} = y(\mathbf{x})$. We can lower-bound (5) by using $\text{err}_{\boldsymbol{\mu}}(f) \geq 0$ and exactly computing the total-variation distance term assuming the optimal graphical model parameters $\boldsymbol{\mu}$ are known, in which case the joint distribution

$\mathbb{P}_{\boldsymbol{\mu}}(\mathbf{y}, \lambda(\mathbf{x}))$ exactly matches the true joint distribution $\mathbb{P}(\mathbf{y}, \lambda(\mathbf{x}))$:

$$\mathbb{P}_{\boldsymbol{\mu}}(\mathbf{y} = 0, \lambda(\mathbf{x}) = 0) = \mathbb{P}(\mathbf{y} = 0, \lambda(\mathbf{x}) = 0) = \mathbb{P}(\mathbf{y} = 0)\mathbb{P}(\lambda(\mathbf{x}) = 0|\mathbf{y} = 0) = \frac{1}{2}(1 - \alpha)$$

$$\mathbb{P}_{\boldsymbol{\mu}}(\mathbf{y} = 0, \lambda(\mathbf{x}) = 1) = \mathbb{P}(\mathbf{y} = 0, \lambda(\mathbf{x}) = 1) = \mathbb{P}(\mathbf{y} = 0)\mathbb{P}(\lambda(\mathbf{x}) = 1|\mathbf{y} = 0) = \frac{1}{2}\alpha$$

$$\mathbb{P}_{\boldsymbol{\mu}}(\mathbf{y} = 1, \lambda(\mathbf{x}) = 1) = \mathbb{P}(\mathbf{y} = 1, \lambda(\mathbf{x}) = 1) = \mathbb{P}(\mathbf{y} = 1)\mathbb{P}(\lambda(\mathbf{x}) = 1|\mathbf{y} = 1) = \frac{1}{2}(1 - \alpha)$$

$$\mathbb{P}_{\boldsymbol{\mu}}(\mathbf{y} = 1, \lambda(\mathbf{x}) = 0) = \mathbb{P}(\mathbf{y} = 1, \lambda(\mathbf{x}) = 0) = \mathbb{P}(\mathbf{y} = 1)\mathbb{P}(\lambda(\mathbf{x}) = 0|\mathbf{y} = 1) = \frac{1}{2}\alpha$$

$$\mathbb{P}_{\boldsymbol{\mu}}(\mathbf{y} = 0, \lambda(\mathbf{x}) = \varnothing) = \mathbb{P}(\mathbf{y} = 0, \lambda(\mathbf{x}) = \varnothing) = \mathbb{P}(\mathbf{y} = 0)\mathbb{P}(\lambda(\mathbf{x}) = \varnothing|\mathbf{y} = 0) = \frac{1}{2}\mathbb{P}(\lambda(\mathbf{x}) = \varnothing)$$

$$\mathbb{P}_{\boldsymbol{\mu}}(\mathbf{y} = 1, \lambda(\mathbf{x}) = \varnothing) = \mathbb{P}(\mathbf{y} = 1, \lambda(\mathbf{x}) = \varnothing) = \mathbb{P}(\mathbf{y} = 1)\mathbb{P}(\lambda(\mathbf{x}) = \varnothing|\mathbf{y} = 1) = \frac{1}{2}\mathbb{P}(\lambda(\mathbf{x}) = \varnothing),$$

where we used the assumptions from Proposition 3.1 that $\mathbb{P}(\tilde{y} = 0|\mathbf{y} = 1) = \mathbb{P}(\tilde{y} = 1|\mathbf{y} = 0) = \alpha$ (the error rates of the weak labels are equal for both classes) and $\mathbb{P}(\lambda(\mathbf{x}) = \varnothing|\mathbf{y} = y) = \mathbb{P}(\lambda(\mathbf{x}) = \varnothing)$ (the weak labels cover each class equally often).

Now we can use this joint distribution to compute the value of the total variation distance term in (5). There are six cases.

**Case 1: $\boldsymbol{x} \in S_0^{good}$.** Here $\boldsymbol{x}$ is a true negative ($\boldsymbol{x} \in S_0$) so $y(\boldsymbol{x}) = 0$. Since $\boldsymbol{x} \in S_0^{good}$ (i.e., the pseudolabel agrees with the true label), $\lambda(\boldsymbol{x}) = 0$. Here we have:

$$\mathbb{P}(\mathbf{y} = 1|\mathbf{x} = \boldsymbol{x}) = 0$$
$$\mathbb{P}(\mathbf{y} = 0|\mathbf{x} = \boldsymbol{x}) = 1$$

$$\mathbb{P}_{\boldsymbol{\mu}}(\mathbf{y} = 0|\lambda(\mathbf{x}) = 0) = \frac{\mathbb{P}_{\boldsymbol{\mu}}(\mathbf{y} = 0, \lambda(\mathbf{x}) = 0)}{\mathbb{P}_{\boldsymbol{\mu}}(\mathbf{y} = 0, \lambda(\mathbf{x}) = 0) + \mathbb{P}_{\boldsymbol{\mu}}(\mathbf{y} = 1, \lambda(\mathbf{x}) = 0)} = \frac{\frac{1}{2}(1 - \alpha)}{\frac{1}{2}(1 - \alpha) + \frac{1}{2}\alpha} = 1 - \alpha$$

$$\mathbb{P}_{\boldsymbol{\mu}}(\mathbf{y} = 1|\lambda(\mathbf{x}) = 0) = \frac{\mathbb{P}_{\boldsymbol{\mu}}(\mathbf{y} = 1, \lambda(\mathbf{x}) = 0)}{\mathbb{P}_{\boldsymbol{\mu}}(\mathbf{y} = 0, \lambda(\mathbf{x}) = 0) + \mathbb{P}_{\boldsymbol{\mu}}(\mathbf{y} = 1, \lambda(\mathbf{x}) = 0)} = \frac{\frac{1}{2}\alpha}{\frac{1}{2}(1 - \alpha) + \frac{1}{2}\alpha} = \alpha$$

And so the (doubled) total variation distance for this $\boldsymbol{x}$ is

$$|1 - \mathbb{P}_{\boldsymbol{\mu}}(\mathbf{y} = 0|\lambda(\mathbf{x}) = 0)| + |0 - \mathbb{P}_{\boldsymbol{\mu}}(\mathbf{y} = 1|\lambda(\mathbf{x}) = 0)| = |1 - (1 - \alpha)| + |0 - \alpha| = 2\alpha.$$

**Case 2: $\boldsymbol{x} \in S_1^{good}$.** Here $\boldsymbol{x}$ is a true positive ($\boldsymbol{x} \in S_1$) so $y(\boldsymbol{x}) = 1$. Since $\boldsymbol{x} \in S_1^{good}$ (i.e., the pseudolabel agrees with the true label), $\lambda(\boldsymbol{x}) = 1$. By the analogous argument to Case 1, the doubled TV distance term for this $\boldsymbol{x}$ is $2\alpha$.

**Case 3: $\boldsymbol{x} \in S_0^{bad}$.** Here $\boldsymbol{x}$ is a true negative ($\boldsymbol{x} \in S_0$) so $y(\boldsymbol{x}) = 0$. Since $\boldsymbol{x} \in S_0^{bad}$ (i.e., the pseudolabel disagrees with the true label), $\lambda(\boldsymbol{x}) = 1$. Here we have:

$$\mathbb{P}(\mathbf{y} = 1|\mathbf{x} = \boldsymbol{x}) = 0$$
$$\mathbb{P}(\mathbf{y} = 0|\mathbf{x} = \boldsymbol{x}) = 1$$

$$\mathbb{P}_{\boldsymbol{\mu}}(\mathbf{y} = 0|\lambda(\mathbf{x}) = 1) = \frac{\mathbb{P}_{\boldsymbol{\mu}}(\mathbf{y} = 0, \lambda(\mathbf{x}) = 1)}{\mathbb{P}_{\boldsymbol{\mu}}(\mathbf{y} = 0, \lambda(\mathbf{x}) = 1) + \mathbb{P}_{\boldsymbol{\mu}}(\mathbf{y} = 1, \lambda(\mathbf{x}) = 1)} = \frac{\frac{1}{2}\alpha}{\frac{1}{2}\alpha + \frac{1}{2}(1 - \alpha)} = \alpha$$

$$\mathbb{P}_{\boldsymbol{\mu}}(\mathbf{y} = 1|\lambda(\mathbf{x}) = 1) = \frac{\mathbb{P}_{\boldsymbol{\mu}}(\mathbf{y} = 1, \lambda(\mathbf{x}) = 1)}{\mathbb{P}_{\boldsymbol{\mu}}(\mathbf{y} = 0, \lambda(\mathbf{x}) = 1) + \mathbb{P}_{\boldsymbol{\mu}}(\mathbf{y} = 1, \lambda(\mathbf{x}) = 1)} = 1 - \alpha$$

And so the (doubled) total variation distance for this $\boldsymbol{x}$ is

$$|1 - \mathbb{P}_{\boldsymbol{\mu}}(\mathbf{y} = 0|\lambda(\mathbf{x}) = 1)| + |0 - \mathbb{P}_{\boldsymbol{\mu}}(\mathbf{y} = 1|\lambda(\mathbf{x}) = 1)| = |1 - \alpha| + |0 - (1 - \alpha)| = 2(1 - \alpha).$$

**Case 4: $\boldsymbol{x} \in S_1^{bad}$.** Here $\boldsymbol{x}$ is a true positive ($\boldsymbol{x} \in S_1$) so $y(\boldsymbol{x}) = 1$. Since $\boldsymbol{x} \in S_1^{bad}$ (i.e., the pseudolabel disagrees with the true label), $\lambda(\boldsymbol{x}) = 0$. By symmetry with Case 3, the (doubled) TV distance for this $\boldsymbol{x}$ is $2(1 - \alpha)$.

**Case 5: $\boldsymbol{x} \in T_0$.** Here $\boldsymbol{x}$ is a true negative by definition of $T_0$, so $y(\boldsymbol{x}) = 0$. Thus $\mathbb{P}(\mathbf{y} = 0|\mathbf{x} = \boldsymbol{x}) = 1$. Since $\boldsymbol{x} \in T$, the pseudolabeler abstains, i.e. $\lambda(\boldsymbol{x}) = \varnothing$. The (doubled) total variation distance for this $\boldsymbol{x}$ is thus:

$$|1 - \mathbb{P}_{\boldsymbol{\mu}}(\mathbf{y} = 0|\lambda(\mathbf{x}) = \varnothing)| + |0 - \mathbb{P}_{\boldsymbol{\mu}}(\mathbf{y} = 1|\lambda(\mathbf{x}) = \varnothing)| = |1 - \frac{1}{2}| + |0 - \frac{1}{2}| = 1$$

**Case 6: $\boldsymbol{x} \in T_1$.** By symmetry with Case 5, in this case the doubled TV term is 1.

**Finishing the proof.** Finally, we can simplify (5). For any partition $\{V_i\}$ of $\mathcal{X}$,

$$\mathbb{E}_{(\mathbf{x}',\mathbf{y}')\sim\mathcal{D}}\left[\sum_y |\mathbb{P}(\mathbf{y} = y|\mathbf{x} = \mathbf{x}') - \mathbb{P}_{\boldsymbol{\mu}}(\mathbf{y} = y|\lambda(\mathbf{x}) = \lambda(\mathbf{x}'))|\right]$$

$$= \sum_i \mathbb{P}(\mathbf{x} \in V_i)\mathbb{E}_{(\mathbf{x}',\mathbf{y}')}\left[\sum_y |\mathbb{P}(\mathbf{y} = y|\mathbf{x} = \mathbf{x}') - \mathbb{P}_{\boldsymbol{\mu}}(\mathbf{y} = y|\lambda(\mathbf{x}) = \lambda(\mathbf{x}'))|\,\Big|\,\mathbf{x}' \in V_i\right]$$

Applying this with our 6 cases above, we obtain:

$$\mathbb{E}_{(\mathbf{x}',\mathbf{y}')\sim\mathcal{D}}\left[\sum_y |\mathbb{P}(\mathbf{y} = y|\mathbf{x} = \mathbf{x}') - \mathbb{P}_{\boldsymbol{\mu}}(\mathbf{y} = y|\lambda(\mathbf{x}) = \lambda(\mathbf{x}'))|\right] = \mathbb{P}(S_0^{good}) \cdot 2\alpha$$

$$+ \mathbb{P}(S_1^{good}) \cdot 2\alpha$$
$$+ \mathbb{P}(S_0^{bad}) \cdot 2(1 - \alpha)$$
$$+ \mathbb{P}(S_1^{bad}) \cdot 2(1 - \alpha)$$
$$+ \mathbb{P}(T_0)$$
$$+ \mathbb{P}(T_1).$$

Now we can group terms together using

$$P(S_y^{good}) = \mathbb{P}(S_y)\mathbb{P}(S_y^{good}|S_y) = \mathbb{P}(S_y)(1 - \alpha)$$
$$P(S_y^{bad}) = \mathbb{P}(S_y)\mathbb{P}(S_y^{bad}|S_y) = \mathbb{P}(S_y)\alpha$$
$$P(S_0) + P(S_1) = P(S)$$
$$P(T_0) + P(T_1) = P(T)$$

for $y \in \{0, 1\}$ to obtain:

$$\mathbb{E}_{(\mathbf{x}',\mathbf{y}')\sim\mathcal{D}}\left[\sum_y |\mathbb{P}(\mathbf{y} = y|\mathbf{x} = \mathbf{x}') - \mathbb{P}_{\boldsymbol{\mu}}(\mathbf{y} = y|\lambda(\mathbf{x}) = \lambda(\mathbf{x}'))|\right] = \mathbb{P}(S)4\alpha(1 - \alpha) + \mathbb{P}(T).$$

Since we ignored the nonnegative $\text{err}_\mu(f)$ term in (5), the following bound is tighter than Fu et al. [21, Theorem 3]:

$$\text{err}(f) \leq \mathbb{P}(S)4\alpha(1 - \alpha) + \mathbb{P}(T).$$

However, this is looser than $\alpha$ on $S$ (since $\alpha < 3/4$), so it can never account for pseudolabel correction, and charges every point in $T$ as an error, so it can't account for coverage expansion.

### B.4 Proposition 3.2

Here we reproduce Theorem 4.3 from Wei et al. [69] in our notation and show how to apply their bound (initially designed for the full-coverage $\mathbb{P}(S) = 1$ pseudolabel correction setting) to the weak supervision setup. The essential idea of the reduction is to treat points $\boldsymbol{x} \in T$, where $\tilde{y}(\boldsymbol{x}) = \varnothing$, as mistakes from the teacher. Unfortunately, this effectively limits the application of the bound to cases where $\mathbb{P}(S)$ is large.

**Definitions.** For a classifier $g$, let $M_i(g) = \{\boldsymbol{x} \in \mathcal{X}_i : g(\boldsymbol{x}) \neq y(\boldsymbol{x})\}$ be the mistakes of $g$ on $\mathcal{X}_i$ (with respect to the true labels $y(\boldsymbol{x})$). In particular, $M_i(g) = \{\boldsymbol{x} \in \mathcal{X}_i : g(\boldsymbol{x}) \neq i\}$. Recall that the pseudolabeler is a classifier $\tilde{y} : \mathcal{X} \to \mathcal{Y} \cup \{\varnothing\}$. In our notation, $M_i(\tilde{y}) = T_i \cup S_i^{bad}$. $M_i(\tilde{y})$ contains $T_i$ because for all $\boldsymbol{x} \in T_i$, $\tilde{y}(\boldsymbol{x}) = \varnothing \neq y(\boldsymbol{x})$.

**Definition** (Expansion, Wei et al. [69]). $\mathbb{P}(\cdot|\mathcal{X}_i)$ *satisfies* $(c, a)$-*expansion if for all* $U \subset M_i(\tilde{y})$ *with* $\mathbb{P}(U|\mathcal{X}_i) \leq a$, $\mathbb{P}(\mathcal{N}(U)|\mathcal{X}_i) \geq \min\{c\mathbb{P}(U|\mathcal{X}_i), 1\}$.

**Theorem** (Wei et al. [69, Theorem 4.3], our notation). *Let* $\bar{a} = \max_i \mathbb{P}(M_i(\tilde{y})|\mathcal{X}_i)$ *and suppose that* $\bar{a} < 1/3$ *and for all* $i$, $\mathbb{P}(\cdot|\mathcal{X}_i)$ *satisfies* $(c, \bar{a})$-*expansion for* $c > 3$. *Suppose the classifier* $\hat{g}$ *minimizes:*

$$L(g, \tilde{y}) := \frac{c+1}{c-1}err(g, \tilde{y}) + \frac{2c}{c-1}\mathbb{P}[\exists \boldsymbol{x}' \in \mathcal{N}(\mathbf{x}) : g(\boldsymbol{x}) \neq g(\boldsymbol{x}')].$$

*Then* $\hat{g}$ *satisfies the following error bound:*

$$err(\hat{g}, y) \leq \frac{2}{c-1}err(\tilde{y}, y) + \frac{2c}{c-1}\mathbb{P}[\exists \boldsymbol{x}' \in \mathcal{N}(\mathbf{x}) : y(\boldsymbol{x}) \neq y(\boldsymbol{x}')].$$

This bound shows that when the expansion (according to their definition) is large and the ground-truth classifier $\tilde{y}$ is adversarially robust over $\mathcal{N}$ on most points, a classifer $\hat{g}$ that minimizes $L(g, \tilde{y})$ enjoys a good upper bound on it error. In particular, this bound can be lower than the error of the pseudolabels $err(\tilde{y}, y)$. However, the assumptions require that $\mathbb{P}(M_i(\tilde{y})|\mathcal{X}_i) < 1/3$ for all $i$. Since $T_i \subset M_i(\tilde{y})$, this requires that $\mathbb{P}(T_i|\mathcal{X}_i) < 1/3$ for all $i$.

In addition to requiring high coverage, this application of Wei et al. [69, Theorem 4.3] also gives one unified bound for the classifier error on the full input space $\mathcal{X}$ instead of dealing with coverage expansion and pseudolabel correction effects separately. Empirically, in real weak supervision settings, the two effects don't always occur together, so ideally a theory for weak supervision would treat the two effects separately. We take this approach with our bounds.

### B.5 Note on finite-sample error bounds

Frei et al. [19] give a finite-sample analysis that shows self-training can give weak-to-strong generalization effects under more strict distributional assumptions than what we consider here. Wei et al. [69] and Cai et al. [12] both provide finite-sample error bounds for the student model in addition to using expansion to prove relationships between the weak label error and true label error on the population distribution. They follow a modular analysis framework that is also followed (for example) by Blum and Mitchell [6], Balcan et al. [3], Lang et al. [37]: first, the population error on the clean labels is related via structural assumptions to the population error on the weak labels. Second, and almost orthogonally, off-the-shelf concentration/finite-sample arguments are applied to the weak label population error. This approach is modular in the sense that once the population error relation is established, the finite-sample issues can be dealt with by purely considering $err(f, \tilde{y}|S)$, and the remainder of the analysis is almost unrelated to each paper's specific setup of self-training, co-training, domain adaptation, or weak supervision. For example, Wei et al. [69], Cai et al. [12] both apply an off-the-shelf generalization result for deep networks from Wei and Ma [68] to $err(f, \tilde{y}|S)$ in this step. The key novelty in each paper is thus to establish reasonable structural assumptions that relate the clean and weak errors on the population data, so this is also the focus of our results. Other work with expansion assumptions, such as HaoChen et al. [23], which gives error bounds for linear classifiers trained on top of contrastive representations, also focuses exclusively on the population case because of the same argument.

## C Connections and comparisons to existing bounds

### C.1 Co-training with conditionally independent views

Suppose that $\mathcal{X} = \mathcal{X}_1 \times \mathcal{X}_2$, so each example $\boldsymbol{x} = (\boldsymbol{x}_1, \boldsymbol{x}_2)$ has two "views". Assume that the weak labels $\tilde{y}$ can be written as a function of $\boldsymbol{x}_1$ alone, and the hypothesis class $\mathcal{F}$ is such that each $f \in \mathcal{F}$ is a function of $\boldsymbol{x}_2$ alone. Finally, suppose the distribution of $\mathbf{x}$ is such that for all $A \subset \mathcal{X}_1$, $B \subset \mathcal{X}_2$, and $i \in \mathcal{Y}$,

$$\mathbb{P}[\mathbf{x}_1 \in A, \mathbf{x}_2 \in B|\mathbf{y}(\mathbf{x}) = i] = \mathbb{P}[\mathbf{x}_1 \in A|\mathbf{y}(\mathbf{x}) = i]\mathbb{P}[\mathbf{x}_2 \in B|\mathbf{y}(\mathbf{x}) = i].$$

That is, the two views are conditionally independent given the (unobserved) true label.

This is precisely the conditionally independent view setup of Blum and Mitchell [6]. They prove the following relationship between the clean error and weak label error of any classifier.

**Theorem C.1** (Blum and Mitchell [6], rephrased). *Fix $f : \mathcal{X}_2 \to \mathcal{Y}$. Let $\alpha_i = \mathbb{P}(S_i^{bad}|S_i)$. In the conditionally independent view setting, the errors of $f$ satisfy:*

$$err(f, y|S_i) \leq \frac{err(f, \tilde{y}|S_i) - \alpha_i}{(1 - 2\alpha_i)} \qquad err(f, y|T_i) \leq \frac{err(f, \tilde{y}|S_i) - \alpha_i}{(1 - 2\alpha_i)}$$

This exactly matches our pseudolabel correction result in Theorem B.1 and our coverage expansion result from Theorem 4.2 when $c = 1$, $q = 0$, and $\mathbb{P}(\overline{R(f)}|T_i) = 0$. We now show how to set $\mathcal{N}$ such that the conditionally independent view distribution provably expands with these parameters and any classifier has $\mathbb{P}(\overline{R(f)}|T_i) = 0$. Our results thus directly generalize those of Blum and Mitchell [6] in three directions, allowing for the views to be dependent, allowing small sets to not have good expansion, and allowing for the case when the classifier is not perfectly robust over $\mathcal{N}$.

**Lemma C.1.** *Suppose the hypothesis class $\mathcal{G}$ is such that $g : \mathcal{X}_2 \to \mathcal{Y}$ for all $g \in \mathcal{G}$. That is, the hypotheses are functions of $\boldsymbol{x}_2$ alone. In the conditionally-independent view setup, the appropriate families of sets $\mathcal{M}(\cdot, \mathcal{G})$ and $\mathcal{M}'(\cdot, \mathcal{G})$ satisfy $(c, q)$-expansion on the pairs of sets $(S_i^{good}, T_i)$, $(S_i^{bad}, T_i)$, $(S_i^{good}, S_i^{bad})$, and $(S_i^{bad}, S_i^{good})$, all with $c = 1$ and $q = 0$. Additionally, any classifier $g \in \mathcal{G}$ is adversarially robust at all $\boldsymbol{x} \in \mathcal{X}$, so $\mathbb{P}(R(g)|A) = 0$ for all $g \in \mathcal{G}$ and all $A \subset \mathcal{X}$. Plugging these coefficients into our Theorems B.1 and 4.2 yields Theorem C.1. Our bounds therefore exactly recover the Blum and Mitchell [6]'s bounds in the conditionally-independent-view setting.*

*Proof.* Let $\mathcal{N}(\boldsymbol{x}) = \mathcal{N}(\boldsymbol{x}_1, \boldsymbol{x}_2) := \{(\boldsymbol{x}', \boldsymbol{x}_2)\}$ be the set of all points in $\mathcal{X}$ that share the same value of $\boldsymbol{x}_2$ as $\boldsymbol{x}$. That is, $\boldsymbol{x}' \in \mathcal{N}(\boldsymbol{x})$ iff $\boldsymbol{x}'_2 = \boldsymbol{x}_2$. By construction, any $g : \mathcal{X}_2 \to \mathcal{Y}$ has $\mathbb{P}(R(g)) = 1$: all points in the same neighborhood have the same value of $\boldsymbol{x}_2$, so $g$ must be constant on every neighborhood. This proves $r(f, \boldsymbol{x}) = 0$ for all $\boldsymbol{x}$.

We will show $c = 1$ for the pair $(S_i^{good}, T_i)$. Completely analogous arguments hold for the three other pairs. The expansion amount $c$ between $S_i^{good}$ and $T_i$ for set size $q \geq 0$ is given by:

$$c = \min_{U : \mathbb{P}(U|T_i) > q} \frac{\mathbb{P}(\mathcal{N}(U)|S_i^{good})}{\mathbb{P}(U|T_i)} = \frac{\mathbb{P}(\mathcal{N}(U), S_i^{good}|\mathrm{y} = i)}{\mathbb{P}(S_i^{good}|\mathrm{y} = i)} \frac{\mathbb{P}(T_i|\mathrm{y} = i)}{\mathbb{P}(U, T_i|\mathrm{y} = i)}$$

Fix an arbitrary $g \in \mathcal{G}$ and let set $U \subset T_i$ be the set of $\boldsymbol{x} \in T_i$ such that $g(\boldsymbol{x}) \neq y(\boldsymbol{x})$. A point $\boldsymbol{x} \in S_i^{good}$ is in $\mathcal{N}(U)$ iff there exists $\boldsymbol{x}' \in U$ with $\boldsymbol{x} \in \mathcal{N}(\boldsymbol{x}')$. By our neighborhood construction, this is equivalent to $\boldsymbol{x}_2 = \boldsymbol{x}'_2$. Let $U_2 = \{\boldsymbol{x}_2 : \boldsymbol{x} \in U\}$ and consider a point $\boldsymbol{x} \in S_i^{good}$. Then $\boldsymbol{x} \in S_i^{good}$ satisfies $\boldsymbol{x} \in \mathcal{N}(U)$ iff $\boldsymbol{x}_2 \in U_2$. Likewise, a point $\boldsymbol{x} \in T_i$ is in $U$ iff $\boldsymbol{x}_2 \in U_2$, since the errors of $g$ (and thus, membership in $U$) only depend on $\boldsymbol{x}_2$. Now consider membership in $S_i^{good}$: $\boldsymbol{x} \in S_i^{good}$ iff $\tilde{y}(\boldsymbol{x}_1) = i$. Conditioned on $\mathrm{y} = i$, $\boldsymbol{x} \in T_i$ iff $\tilde{y}(\boldsymbol{x}_1) = \varnothing$. We can thus rewrite the expression above as:

$$\frac{\mathbb{P}(\mathcal{N}(U), S_i^{good}|\mathrm{y} = i)}{\mathbb{P}(S_i^{good}|\mathrm{y} = i)} \frac{\mathbb{P}(T_i|\mathrm{y} = i)}{\mathbb{P}(U, T_i|\mathrm{y} = i)} = \frac{\mathbb{P}(\boldsymbol{x}_2 \in U_2, \tilde{y}(\boldsymbol{x}_1) = i|\mathrm{y} = i)}{\mathbb{P}(\tilde{y}(\boldsymbol{x}_1) = i|\mathrm{y} = i)} \frac{\mathbb{P}(\tilde{y}(\boldsymbol{x}_1) = \varnothing|\mathrm{y} = i)}{\mathbb{P}(\boldsymbol{x}_2 \in U_2, \tilde{y}(\boldsymbol{x}_1) = \varnothing|\mathrm{y} = i)}$$

By the conditional independence assumption on the two views, the numerator of the first term and the denominator of the second term factor and cancel out their denominator and numerator, respectively. This yields:

$$\frac{\mathbb{P}(\boldsymbol{x}_2 \in U_2|\mathrm{y} = i)}{1} \frac{1}{\mathbb{P}(\boldsymbol{x}_2 \in U_2|\mathrm{y} = i)} = 1.$$

Since $U$ was arbitrary, this shows that in the conditionally-independent view setup has $(c, q)$-expansion for the pair $(S_i^{good}, T_i)$ with $c = 1$ and $q = 0$. The argument for the remaining three pairs of sets is completely analogous. $\square$

### C.2 Wei et al. [69]'s pseudolabel correction bound

Proposition 3.2 shows that Wei et al. [69]'s error bound can't easily be ported to the full weak supervision setting (i.e., pseudolabel correction *and* coverage expansion) because it requires high coverage $\mathbb{P}(S)$. In this section, we show that even for pseudolabel correction, our bound is significantly different from Wei et al.'s bound and has three desirable properties that Wei et al. [69, Theorem

A.2] (the more general version of Wei et al. [69, Theorem 4.3]) lacks: (a) our bound more directly generalizes the Blum and Mitchell [6] bounds in the conditionally-independent-view setting from Section C.1, (b) our expansion parameter naturally appears in our bound, making it clear that more expansion yields a tighter bound, and (c) our bound empirically applies to more classifiers since it places less strict conditions on the student model.

First, we prove a direct analogue of Wei et al. [69, Theorem A.2] using our multiplicative expansion assumptions, since the original uses a slightly different additive expansion assumption. The proofs are almost identical.

**Theorem C.2** (Wei et al. [69] Theorem A.2, restated). *Suppose $\mathcal{M}(S_i^{bad}, \mathcal{F})$ satisfies $(c, q)$-expansion on $(S_i^{good}, S_i^{bad})$ for some $c > \alpha_i/(1-\alpha_i)$. Let $\tilde{c} = c\frac{1-\alpha_i}{\alpha_i}$ and suppose that the classifier $f$ satisfies:*

$$\mathbb{P}(f(\boldsymbol{x}) \neq \tilde{y}(\boldsymbol{x}) \text{ or } f \text{ not robust at } \boldsymbol{x}|S_i) \leq \alpha_i(1 + q(\tilde{c} - 1))$$

*Then:*

$$err(f, y|S_i) \leq 2\left(q\alpha_i + \mathbb{P}(\overline{R(f)}|S_i)\right) + err(f, \tilde{y}|S_i) - \alpha_i.$$

**Note:** The following result shows our robust expansion technique can also be used to directly generalize this result to average-case-robust classifiers, whereas Wei et al. [69, Theorem A.2] only applies to adversarially robust classifiers. Theorem C.2 is a special case of Theorem C.3 with $\eta = 0$.

**Theorem C.3.** *Suppose $M_\eta(S_i^{bad}, \mathcal{F})$ satisfies $(c, q, \eta)$-expansion on $(S_i^{good}, S_i^{bad})$ for some $c > \alpha_i/(1-\alpha_i)$. Let $\tilde{c} = c\frac{1-\alpha_i}{\alpha_i}$ and additionally suppose:*

$$\mathbb{P}(f(\boldsymbol{x}) \neq \tilde{y}(\boldsymbol{x}) \text{ or } f \text{ not } \eta\text{-robust at } \boldsymbol{x}|S_i) \leq \alpha_i(1 + q(\tilde{c} - 1))$$

*Then:*

$$err(f, y|S_i) \leq 2\left(q\alpha_i + \mathbb{P}(\overline{R_\eta(f)}|S_i)\right) + err(f, \tilde{y}|S_i) - \alpha_i.$$

*Furthermore, if $f$ satisfies:*

$$\mathbb{E}_{\mathbf{x}'\sim\mathcal{D}|S_i, \mathbf{x}\sim\mathcal{D}|\mathcal{N}(\mathbf{x}')}[f(\mathbf{x}) \neq f(\mathbf{x}')] \leq \gamma$$

*for some $\gamma \geq 0$, then $\mathbb{P}(\overline{R_\eta(f)}|S_i) \leq \frac{\gamma}{\eta}$.*

*Proof.* We directly prove Theorem C.3 since Theorem C.2 is a special case with $\eta = 0$. The proof is largely similar to Wei et al. [69, Theorem A.2] but with our multiplicative version of the expansion assumptions and our generalization to robust expansion. Let $M_i = \{\boldsymbol{x} \in S_i | f(\boldsymbol{x}) \neq y(\boldsymbol{x})\}$ be the set of mistakes of $f$ in $S_i$; note that $err(f, y|S_i) = \mathbb{P}(M_i|S_i)$. Let $D_i = \{\boldsymbol{x} \in S_i | f(\boldsymbol{x}) \neq \tilde{y}(\boldsymbol{x})\}$ be the set of disagreements between $f$ and $\tilde{y}$ in $S_i$; note that $err(f, \tilde{y}|S_i) = \mathbb{P}(D_i|S_i)$.

We can partition $M_i$ into three sets:

$$A_1 = (S_i \setminus D_i) \cap S_i^{bad}$$
$$A_2 = D_i \cap M_i \cap S_i^{bad}$$
$$A_3 = D_i \cap S_i^{good}$$

Points $\boldsymbol{x} \in A_1$ must be in $M_i$ since $f(\boldsymbol{x}) = \tilde{y}(\boldsymbol{x}) \neq y(\boldsymbol{x})$ by the definitions of $S_i \setminus D_i$ and $S_i^{bad}$, respectively, so $A_1 = A_1 \cap M_i$. Points $\boldsymbol{x} \in A_3$ must be in $M_i$ since $f(\boldsymbol{x}) \neq \tilde{y}(\boldsymbol{x}) = y(\boldsymbol{x})$ by the definitions of $D_i$ and $S_i^{good}$, respectively, so $A_3 \subset M_i \cap S_i^{good}$. Finally, any point in $M_i \cap S_i^{good}$ must be in $D_i$ since otherwise we'd have $f(\boldsymbol{x}) = \tilde{y}(\boldsymbol{x}) = y(\boldsymbol{x})$ and thus $\boldsymbol{x} \notin M_i$, so $M_i \cap S_i^{good} \subset A_3$. Together, this indicates that $\{A_1, A_2, A_3\}$ is a partition of $M_i$, since we've shown:

$$\begin{aligned}
A_1 \cup A_2 \cup A_3 = \quad & (S_i \setminus D_i) \cap M_i \cap S_i^{bad} \\
& \cup D_i \cap M_i \cap S_i^{bad} \\
& \cup D_i \cap M_i \cap S_i^{good} \\
= \; & M_i \cap S_i^{bad} \cup M_i \cap S_i^{good} \\
= \; & M_i
\end{aligned}$$

Now define $U_i = M_i \cap S_i^{bad} \cap R_\eta(f)$. Note that $A_1 \cap R_\eta(f)$ and $A_2 \cap R_\eta(f)$ are disjoint subsets of $U_i$. Since $U_i \subset R_\eta(f)$, Lemma B.2 implies that $w(\widetilde{\mathcal{N}}(U_i), U_i) \geq (1 - \eta)w(\mathcal{N}(U_i), U_i)$, so $\mathbb{P}(\widetilde{\mathcal{N}}(U_i)|S_i^{good}) \geq P_{1-\eta}(S_i^{good}, U_i)$. Since $U_i$ is a mistake set intersected with a robust set, $U_i \in \mathcal{M}(S_i^{bad}, \mathcal{F}, y)$.

Suppose for the sake of contradiction that $\mathbb{P}(U_i|S_i^{bad}) > q$. By the expansion assumption, we have:

$$\mathbb{P}(\widetilde{\mathcal{N}}(U_i) \cap S_i^{good}|S_i) = (1 - \alpha_i)\mathbb{P}(\widetilde{\mathcal{N}}(U_i)|S_i^{good}) \geq (1 - \alpha_i)P_{1-\eta}(S_i^{good}, U_i)$$
$$> c(1 - \alpha_i)\mathbb{P}(U_i|S_i^{bad}) = c\frac{(1 - \alpha_i)}{\alpha_i}\mathbb{P}(U_i|S_i).$$

Since we assumed $c > \frac{\alpha_i}{1-\alpha_i}$, for some $\tilde{c} > 1$ we have

$$\mathbb{P}(\widetilde{\mathcal{N}}(U_i) \cap S_i^{good}|S_i) > \tilde{c} \cdot \mathbb{P}(U_i|S_i).$$

Now we decompose $S_i$ into three disjoint sets (note the differences between $V_j$'s and $A_j$'s):

$$V_1 = S_i \setminus D_i$$
$$V_2 = D_i \cap S_i^{bad}$$
$$V_3 = D_i \cap S_i^{good}$$

Consider a point $\boldsymbol{x} \in \widetilde{\mathcal{N}}(U_i) \cap S_i^{good}$. By definition of $\widetilde{\mathcal{N}}$, there exists $\boldsymbol{x}' \in U_i$ reachable from $\boldsymbol{x}$ by a good edge, so $f(\boldsymbol{x}) = f(\boldsymbol{x}')$. Since $\boldsymbol{x}$ and $\boldsymbol{x}'$ are both in $S_i$, $y(\boldsymbol{x}) = y(\boldsymbol{x}')$. Since $U_i \subset M_i$, $f(\boldsymbol{x}') \neq y(\boldsymbol{x}')$, so we must have $f(\boldsymbol{x}) \neq y(\boldsymbol{x}) = \tilde{y}(\boldsymbol{x})$. Hence $\boldsymbol{x} \in D_i \cap S_i^{good}$. This shows that $\widetilde{\mathcal{N}}(U_i) \cap S_i^{good} \subset V_3$. Using the decomposition above, we have:

$$1 = \mathbb{P}(S_i|S_i) = \mathbb{P}(V_1|S_i) + \mathbb{P}(V_2|S_i) + \mathbb{P}(V_3|S_i)$$
$$\geq \mathbb{P}(V_1|S_i) + \mathbb{P}(V_2|S_i) + \mathbb{P}(\widetilde{\mathcal{N}}(U_i) \cap S_i^{good}|S_i)$$
$$> \mathbb{P}(V_1|S_i) + \mathbb{P}(V_2|S_i) + \tilde{c}\mathbb{P}(U_i|S_i). \tag{6}$$

**Lemma C.2.**
$$\alpha_i \leq \mathbb{P}(V_2|S_i) + \mathbb{P}(U_i|S_i) + \mathbb{P}(S_i^{bad} \cap V_1 \cap \overline{R_\eta(f)}|S_i)$$

*Proof.*

$$\alpha_i = \mathbb{P}(S_i^{bad}|S_i) = \mathbb{P}(S_i^{bad} \cap D_i|S_i) + \mathbb{P}(S_i^{bad} \cap (S_i \setminus D_i)|S_i)$$
$$= \mathbb{P}(V_2|S_i) + \mathbb{P}(S_i^{bad} \cap V_1 \cap R_\eta(f)|S_i) + \mathbb{P}(S_i^{bad} \cap V_1 \cap \overline{R_\eta(f)}|S_i).$$

Consider $\boldsymbol{x} \in S_i^{bad} \cap V_1 \cap R_\eta(f)$. Since $\boldsymbol{x} \in S_i^{bad}$, $\tilde{y}(\boldsymbol{x}) \neq f(\boldsymbol{x})$. Since $\boldsymbol{x} \in V_1 = S_i \setminus D_i$, $f(\boldsymbol{x}) = \tilde{y}(\boldsymbol{x})$. Thus $f(\boldsymbol{x}) \neq y(\boldsymbol{x})$ and hence $\boldsymbol{x} \in M_i$. Because $U_i = M_i \cap S_i^{bad} \cap R_\eta(f)$, this shows that $S_i^{bad} \cap V_1 \cap R_\eta(f) \subset U_i$. The lemma follows.

$\square$

Plugging Lemma C.2 into (6) yields:

$$1 > \mathbb{P}(V_1|S_i) + \alpha_i + (\tilde{c} - 1)\mathbb{P}(U_i|S_i) - \mathbb{P}(V_1 \cap S_i^{bad} \cap \overline{R_\eta(f)}|S_i)$$
$$= \mathbb{P}(V_1 \cap \overline{(S_i^{bad} \cap \overline{R_\eta(f)})}|S_i) + \alpha_i + (\tilde{c} - 1)\mathbb{P}(U_i|S_i)$$
$$= \mathbb{P}(V_1 \cap (S_i^{good} \cup R_\eta(f))|S_i) + \alpha_i + (\tilde{c} - 1)\mathbb{P}(U_i|S_i)$$
$$\geq \mathbb{P}(V_1 \cap R_\eta(f)|S_i) + \alpha_i + (\tilde{c} - 1)\mathbb{P}(U_i|S_i). \tag{7}$$

Now recall that $V_1 = S_i \setminus D_i$, so:

$$\mathbb{P}(V_1 \cap R_\eta(f)|S_i) = 1 - \mathbb{P}(D_i \cap \overline{R_\eta(f)}|S_i)$$
$$= 1 - \mathbb{P}(f(\boldsymbol{x}) \neq \tilde{y}(\boldsymbol{x}) \text{ or } f \text{ not } \eta\text{-robust at } \boldsymbol{x}|S_i)$$

We assumed in the theorem statement that:

$$\mathbb{P}(f(\boldsymbol{x}) \neq \tilde{y}(\boldsymbol{x}) \text{ or } f \text{ not } \eta\text{-robust at } \boldsymbol{x}|S_i) \leq \alpha_i(1 + q(\tilde{c} - 1))$$
$$\leq \alpha_i + (\tilde{c} - 1)\alpha_i\mathbb{P}(U_i|S_i^{bad})$$
$$= \alpha_i + (\tilde{c} - 1)\mathbb{P}(U_i|S_i).$$

And hence (7) gives $1 > 1$, a contradiction. It must therefore be the case that $\mathbb{P}(U_i|S_i^{bad}) < q$. Since $A_1 \cap R_\eta(f)$ and $A_2 \cap R_\eta(f)$ are disjoint subsets of $U_i$, we have that $\mathbb{P}(A_1 \cap R_\eta(f)|S_i) + \mathbb{P}(A_2 \cap R_\eta(f)|S_i) \leq q\alpha_i$.

We can now finish the bound.

$$\text{err}(f, y|S_i) = \mathbb{P}(M_i|S_i) = \mathbb{P}(A_1 \cup A_2 \cup A_3|S_i)$$
$$= \mathbb{P}((A_1 \cup A_2 \cup A_3) \cap R_\eta(f)|S_i) + \mathbb{P}((A_1 \cup A_2 \cup A_3) \cap \overline{R_\eta(f)}|S_i)$$
$$\leq \mathbb{P}((A_1 \cup A_2 \cup A_3) \cap R_\eta(f)|S_i) + \mathbb{P}(\overline{R_\eta(f)}|S_i)$$
$$\leq q\alpha_i + \mathbb{P}(A_3 \cap R_\eta(f)|S_i) + \mathbb{P}(\overline{R_\eta(f)}|S_i).$$

Since $(A_3 \cup (S_i \setminus D_i)) \cap R_\eta(f) = (A_1 \cup S_i^{good}) \cap R_\eta(f)$, we have:

$$(A_3 \cap R_\eta(f)) \cup (S_i \setminus D_i) \cap R_\eta(f) = A_1 \cap R_\eta(f) \cup S_i^{good} \cap R_\eta(f).$$

All sets that are arguments to the unions above are disjoint, so:

$$\mathbb{P}(A_3 \cap R_\eta(f)|S_i) + \mathbb{P}(\overline{D_i} \cap R_\eta(f)|S_i) = \mathbb{P}(A_1 \cap R_\eta(f)|S_i) + \mathbb{P}(S_i^{good} \cap R_\eta(f)|S_i)$$

Using $\mathbb{P}(\overline{D_i} \cap R_\eta(f)|S_i) = 1 - \mathbb{P}(D_i \cup \overline{R_\eta(f)}|S_i)$, $\mathbb{P}(S_i^{good} \cap R_\eta(f)) \leq 1 - \alpha_i$, and $\mathbb{P}(A_1 \cap R_\eta(f)|S_i) \leq q\alpha_i$, we obtain:

$$\mathbb{P}(A_3 \cap R_\eta(f)|S_i) \leq \mathbb{P}(D_i \cup \overline{R_\eta(f)}|S_i) + q\alpha_i - \alpha_i.$$

Plugging this in to the earlier bound gives:

$$\text{err}(g, y|S_i) \leq \mathbb{P}(\overline{R_\eta(f)}|S_i) + 2q\alpha_i + \mathbb{P}(D_i \cup \overline{R_\eta(f)}|S_i) - \alpha_i,$$

which we can leave as-is:

$$\text{err}(f, y|S_i) \leq \mathbb{P}(\overline{R_\eta(f)}|S_i) + 2q\alpha_i + \mathbb{P}(f(\boldsymbol{x}) \neq y(\boldsymbol{x}) \text{ or } f \text{ not } \eta\text{-robust at } \boldsymbol{x}|S_i) - \alpha_i,$$

or union-bound to obtain:

$$\text{err}(f, y|S_i) \leq 2\left(q\alpha_i + \mathbb{P}(\overline{R_\eta(f)}|S_i)\right) + \text{err}(f, \tilde{y}|S_i) - \alpha_i.$$

The second part of the theorem follows directly from Lemma B.1.

$\square$

**Comparing Theorems 4.1 and C.2 in the Co-Training setting.** In Section C.1 we argued that the conditionally-independent-view setup of Blum and Mitchell [6] satisfies our expansion assumptions with $c = 1$ and $q = 0$, and any model is inherently perfectly robust because it's only trained on one of the views, so $\mathbb{P}(\overline{R(f)}|S_i) = 0$ for all $f \in \mathcal{F}$. We can plug these values into the conditions of Theorem C.2. In particular, Theorem C.2 only applies to classifiers $f$ that satisfy:

$$\text{err}(f, \tilde{y}|S_i) \leq \alpha_i\left(1 + q(\tilde{c} - 1)\right) = \alpha_i,$$

But in the conditionally independent view setting, $\min_{f \in \mathcal{F}} \text{err}(f, \tilde{y}|S_i) = \alpha_i$. In this case, the error bound in Theorem C.2 simplifies to:

$$\text{err}(f, y|S_i) \leq \text{err}(f, \tilde{y}|S_i) - \alpha_i = 0$$

Thus, in this setting, Theorem C.2 only applies to classifiers that minimize the weak error as much as possible, and therefore have 0 error on the true labels. In contrast, our Theorem B.1 applies to any $f$ with $\text{err}(f, \tilde{y}|S_i) \leq 1 - q - \alpha_i = 1 - \alpha_i$. When $c = 1$, the value of $c'$ in Theorem B.1 reduces to $c' = 1$, and our bound exactly recovers the Blum and Mitchell bound for any classifier with error at most $1 - \alpha_i$. This relaxation of which classifiers qualify for the bound would be important for finite-sample

Table 2: Comparison of measured values of the amount of expansion $c$ ("exp. $c$ val.") on the data described in Section 6. Theorem B.1 (our pseudolabel correction result) requires $\mathcal{M}'(S_i^{good}, \mathcal{F})$ to expand on the pair $(S_i^{bad}, S_i^{good})$ for some $c > 0$. That is, it requires robust non-mistakes to expand to points with the wrong pseudolabels. We call this "good-to-bad" or G2B expansion. On the other hand, Theorem C.2 requires $\mathcal{M}(S_i^{bad}, \mathcal{F})$ to expand on the pair $(S_i^{good}, S_i^{bad})$ for some $c > \alpha_i/(1 - \alpha_i)$. In other words, it requires robust mistakes to expand to points with the correct pseudolabel. We call this "bad-to-good" or B2G expansion. These results show that empirically, we may have the G2B $c > 0$, so our bounds apply, but the B2G $c < \alpha_i/(1 - \alpha_i)$, so Theorem C.2 does not apply. This is the case for the $i = 1$ values.

| $i$ | $\alpha_i$ | $\alpha_i/(1 - \alpha_i)$ | $(S_i^{bad}, S_i^{good})$ (G2B) exp. $c$ val. | $(S_i^{good}, S_i^{bad})$ (B2G) exp. $c$ val. |
|---|---|---|---|---|
| 0 | 0.11 | 0.12 | 0.85 | 0.17 |
| 1 | 0.33 | **0.49** | 0.50 | **0.32** |

guarantees, since Theorem C.2 only applies to $f$ for which $\mathrm{err}(f, \tilde{y}|S_i) - \min_{f' \in \mathcal{F}} \mathrm{err}(f', \tilde{y}|S_i) = 0$. It is not possible to guarantee that we can obtain such an $f$ from a finite sample using standard improper PAC learning techniques. In contrast, our bound still applies to student models $f$ with $\mathrm{err}(f, \tilde{y}|S_i) - \min_{f' \in \mathcal{F}} \mathrm{err}(f', \tilde{y}|S_i) > 0$.

Our result can also recover the same bound as Theorem C.2 in this setting: when $c = 1$, we can take $\Delta = 1$ in Corollary B.1. The condition on the classifier then becomes $\mathrm{err}(f, \tilde{y}|S_i) \leq \alpha_i$, and in that case the bound in Corollary B.1 simplifies to

$$\mathrm{err}(f, y|S_i) \leq \mathrm{err}(f, \tilde{y}|S_i) + \alpha_i(1 - 2c\Delta)$$
$$= \mathrm{err}(f, \tilde{y}|S_i) - \alpha_i = 0$$

so Theorem B.1 can exactly match Theorem C.2 in this case. This discussion shows that our result can match the bound from Theorem C.2 in this setting, but is more stable, since it applies to classifiers that do not attain the exact optimal weak label error on the population.

**Non-Robust coefficient.** The bound in Theorem B.1 has a coefficient of $\frac{2\alpha_i}{1 - 2\alpha_i}$ on the probability of non-robust points $\mathbb{P}(\overline{R(f)}|S_i)$. The bound in Theorem C.3 has a coefficient of 2. Our dependence on the probability of non-robust points is better when $\alpha < 1/3$.

**Expansion term not in the bound.** Theorem C.2 does not have the amount of expansion $c$ in the error bound—instead, the expansion is only present in the pre-conditions that the classifier needs to meet for the bound to apply. In Wei et al. [69], the authors avoid this by introducing a loss function that contains the expansion $c$ and assuming the classifier $f$ minimizes that loss, essentially re-introducing a dependence on $c$. However, achieving this bound (Wei et al. [69, Theorem 4.3]) therefore requires *knowing the expansion up-front* in order to minimize the suitably-scaled loss function. In contrast, our pseudolabel correction bound directly has the amount of expansion $c$ on the right-hand-side.

**Empirically less sensitive.** Theorem C.2 requires $c > \alpha_i/(1 - \alpha_i)$ for it to apply. This is not an artifact of our conversion from Wei et al. [69]'s additive expansion to our multiplicative expansion, as shown by the following lemma.

**Lemma C.3.** *Suppose that the distribution $\mathbb{P}(\cdot|S_i)$ satisfies $(q, \delta)$-additive expansion on $S_i^{bad}$ [69, Definition A.1] for some $\delta > 0$. Then $\mathbb{P}(\cdot|S_i)$ satisfies $(c, \frac{q}{\alpha_i})$-expansion on $(S_i^{good}, S_i^{bad})$ with $c > \alpha_i/(1 - \alpha_i)$.*

*Proof.* $(q, \delta)$-additive expansion implies that for any $U \subset S_i^{bad}$ with $P(U|S_i) > q$,

$$\mathbb{P}(\mathcal{N}(U) \setminus S_i^{bad}|S_i) \geq \mathbb{P}(U|S_i) + \delta$$
$$\mathbb{P}(\mathcal{N}(U) \cap S_i^{good}|S_i) \geq \mathbb{P}(U \cap S_i^{bad}|S_i) + \delta$$
$$\mathbb{P}(\mathcal{N}(U)|S_i^{good})(1 - \alpha_i) \geq \mathbb{P}(U|S_i^{bad})\alpha_i + \delta,$$

so $\mathbb{P}(\mathcal{N}(U)|S_i^{good}) > \frac{\alpha_i}{1-\alpha_i}\mathbb{P}(U|S_i^{bad})$. We assumed $U \subset S_i^{bad}$ was an arbitrary set with $\mathbb{P}(U|S_i) = \mathbb{P}(U|S_i^{bad})\alpha_i > q$, so $\mathbb{P}(U|S_i^{bad}) > q/\alpha_i$. Hence this example satisfies $(c, q/\alpha_i)$-expansion on the pair of sets $(S_i^{good}, S_i^{bad})$. $\qquad\square$

Lemma C.3 shows that if Wei et al. [69]'s additive expansion assumptions hold, our $(S_i^{good}, S_i^{bad})$ expansion assumption holds with $c > \alpha_i/(1 - \alpha_i)$. However, our experiments suggest that this amount of expansion may not be present empirically for the error sets of actual trained classifiers. In contrast, our pseudolabel correction result only requires $c > 0$ and the bounds computed using our empirical measurements are close to the true error of the classifiers. Table 2 shows an example of this on the data from Section 6. For label 1, $\alpha_i/(1 - \alpha_i) = 0.49$ but the measured value of the expansion between $(S_i^{good}, S_i^{bad})$ is only 0.32, so Theorem C.2 does not apply. Of course, this does not rule out the possibility that C.2 may apply with a different choice of the neighborhood $\mathcal{N}$, but for at least one natural setup ($\mathcal{N}$ the set of paraphrases of the text input $\boldsymbol{x}$), there is not enough measured expansion for it to apply.

# D  Checking expansion: Statistical theory

**Theorem D.1** (Theorem 5.1, formal). *For $U \in \mathcal{M}$, define:*

$$c(U) := \frac{\mathbb{P}(n(\mathbf{x}) \in U | \mathbf{x} \in A)}{\mathbb{P}(\mathbf{x} \in U | \mathbf{x} \in B)}$$

*and its empirical version:*

$$\hat{c}(U) := \frac{\frac{1}{n_A}\sum_{i=1}^{n_A} \mathbb{1}[n(\boldsymbol{x}_i) \in U]}{\frac{1}{m}\sum_{j=1}^{m} \mathbb{1}[\boldsymbol{x}_i \in U]}$$

*Suppose there is a lower bound $\bar{q} > 0$ such that $q(U) \geq \bar{q}$ for all $U \in \mathcal{M}$. Define $\gamma = n_B\bar{q}/n_A$ and let $m = n_A + n_B$. Then for any $\delta \in (0, 1]$, with probability at least $1 - \delta$ over the sampling of $\mathcal{S}_A$ and $\mathcal{S}_B$,*

$$\sup_{U \in \mathcal{M}} \hat{c}(U) - c(U) \leq 4(4 + \sqrt{\gamma})\sqrt{\frac{\text{VC}(\mathcal{M})\log\frac{2em}{\text{VC}(\mathcal{M})} + \log\frac{8}{\delta}}{n_A\bar{q}^2}}.$$

*Proof of Theorem D.1.* We start off with a lemma providing concentration results for a fixed set $U \in \mathcal{M}$.

**Lemma D.1** (Concentration for a fixed $U$). *Fix an arbitrary set $U \in \mathcal{M}$. Define:*

$$p(U) := \mathbb{P}(n(\mathbf{x}) \in U | \mathbf{x} \in A) \qquad q(U) := \mathbb{P}(\mathbf{x} \in U | \mathbf{x} \in B)$$

*And their empirical versions:*

$$\hat{p}(U) := \frac{1}{n_A}\sum_{\boldsymbol{x}_i \in \mathcal{S}_A} \mathbb{1}[n(\boldsymbol{x}_i) \in U] \qquad \hat{q}(U) := \frac{1}{n_B}\sum_{\boldsymbol{x}_i \in \mathcal{S}_B} \mathbb{1}[\boldsymbol{x}_i \in U].$$

*Suppose there is a lower bound $\bar{q} > 0$ such that $q(U) \geq \bar{q}$ for all $U \in \mathcal{M}$. Define $\gamma = n_B\bar{q}/n_A$. Then for any $\delta \in (0, 1]$,*

$$\mathbb{P}\left[\frac{\hat{p}(U)}{\hat{q}(U)} - \frac{p(U)}{q(U)} > \delta\right] \leq 2\exp\left(-2n_A\delta^2 \cdot \frac{\gamma\bar{q}^2}{\left(4 + \sqrt{\gamma}\right)^2}\right),$$

where the probability is with respect to the sampling of $\mathcal{S}_A$ and $\mathcal{S}_B$.

*Proof.* Fix $\epsilon > 0$ and let $E$ be the event that $(1 - \epsilon)q(U) \leq \hat{q}(U)$. Let $W$ refer to the event that $\frac{\hat{p}(U)}{\hat{q}(U)} - \frac{p(U)}{q(U)} > \delta$. Then

$$\mathbb{P}[W] = \mathbb{P}[W|E]\mathbb{P}[E] + \mathbb{P}[W|\overline{E}]\mathbb{P}[\overline{E}]$$
$$\leq \mathbb{P}[W|E] + \mathbb{P}[\overline{E}].$$

By the multiplicative Chernoff bound,

$$\mathbb{P}[\overline{E}] = \mathbb{P}[\hat{q}(U) < (1 - \epsilon)q(U)] \leq \exp(-\epsilon^2 n_B \cdot q(U)/2).$$

Now the goal is to bound $\mathbb{P}[W|E]$.

$$\mathbb{P}[W|E] = \mathbb{P}\left[\frac{\hat{p}(U)}{\hat{q}(U)} - \frac{p(U)}{q(U)} > \delta \Big| E\right] \leq \mathbb{P}\left[\frac{\hat{p}(U)}{(1-\epsilon)q(U)} - \frac{p(U)}{q(U)} > \delta \Big| E\right]$$

$$= \mathbb{P}\left[\frac{\hat{p}(U)}{(1-\epsilon)q(U)} - \frac{p(U)}{q(U)} > \delta\right]$$

$$= \mathbb{P}[\hat{p} > p + q\delta - \epsilon(p + q\delta)]$$

$$\leq \mathbb{P}[\hat{p} > p + \bar{q}\delta - 2\epsilon],$$

where we used in the second line that the samples from $A$ and $B$ are independent (and thus, $\hat{p}(U)$ and $\hat{q}(U)$ are independent), and in the last line that $q \geq \bar{q}$, $0 \leq p \leq 1$, $0 \leq q \leq 1$, and $\delta \in (0, 1]$. Setting $\epsilon' = \bar{q}\delta - 2\epsilon$, and using the Chernoff-Hoeffding theorem, we obtain:

$$\mathbb{P}[W|E] \leq \mathbb{P}[\hat{p} > \mathbb{E}[\hat{p}] + \epsilon'] \leq \exp(-2(\epsilon')^2 \cdot n_A)$$

Combining results, we have that for any $\epsilon > 0$:

$$\mathbb{P}[W] \leq \exp(-2(\epsilon')^2 \cdot n_A) + \exp(-\epsilon^2 n_B \cdot \bar{q}/2).$$

Let $\gamma = n_B \bar{q}/n_A$. Setting $\epsilon = \frac{2\bar{q}\delta}{4+\sqrt{\gamma}}$ (this is valid since $\epsilon > 0$) yields $\epsilon' = \frac{\bar{q}\delta\sqrt{\gamma}}{4+\sqrt{\gamma}}$, so we obtain

$$\mathbb{P}[W] \leq 2\exp\left(-2n_A\delta^2 \cdot \frac{\gamma\bar{q}^2}{\left(4 + \sqrt{\gamma}\right)^2}\right).$$

$\square$

**Corollary D.1.** *An almost exactly symmetric argument yields:*

$$\mathbb{P}\left[\frac{p(U)}{q(U)} - \frac{\hat{p}(U)}{\hat{q}(U)} > \delta\right] \leq 2\exp\left(-n_A\delta^2 \cdot \frac{\gamma\bar{q}^2}{\left(2 + \sqrt{\gamma}\right)^2}\right),$$

*and thus:*

$$\mathbb{P}\left[\left|\frac{p(U)}{q(U)} - \frac{\hat{p}(U)}{\hat{q}(U)}\right| > \delta\right] \leq 4\exp\left(-n_A\delta^2 \cdot \frac{\gamma\bar{q}^2}{\left(4 + \sqrt{\gamma}\right)^2}\right)$$

Now we show how to apply this deviation bound for $\hat{p}(U)/\hat{q}(U)$ in place of Hoeffding's inequality in the symmetrization argument from Bousquet et al. [8].

**Lemma D.2** (Symmetrization). *Suppose we have a ghost sample of $n_A$ additional points $\boldsymbol{x}_i'$ drawn i.i.d. from $P(\cdot|A)$ and $n_B$ additional points $\boldsymbol{z}_i'$ drawn i.i.d. from $P(\cdot|B)$. Let $\hat{c}'(U) = \hat{p}'(U)/\hat{q}'(U)$ denote the empirical expansion of set $U$ on the ghost sample, and let $\hat{c}(U) = \hat{p}(U)/\hat{q}(U)$ denote the empirical expansion of set $U$ on the original sample. Then for any $t > 0$ such that $n_A \cdot t^2 \geq \frac{(4+\sqrt{\gamma})^2 \log 16}{\gamma\bar{q}^2}$:*

$$\mathbb{P}\left[\sup_{U \in \mathcal{M}} \hat{c}(U) - c(U) > t\right] \leq 2\mathbb{P}\left[\sup_{U \in \mathcal{M}} \hat{c}(U) - \hat{c}'(U) > t/2\right].$$

*Proof of Lemma D.2.* This follows Bousquet et al. [8] exactly, except we replace the application of one inequality with the deviation bound derived above. Let $U_n$ be the set achieving the supremum on the left-hand-side. This depends on the sample $(\boldsymbol{x}_1, \ldots, \boldsymbol{x}_{n_A+n_B})$.

$$\mathbb{1}_{\hat{c}(U_n)-c(U_n)>t}\mathbb{1}_{\hat{c}'(U_n)-c(U_n)<t/2} = \mathbb{1}_{\hat{c}(U_n)-c(U_n)>t \wedge c(U_n)-\hat{c}'(U_n)\geq -t/2}$$

$$\leq \mathbb{1}_{\hat{c}(U_n)-\hat{c}'(U_n)>t/2}$$

Taking the expectation over the ghost sample $(\boldsymbol{x}_1', \ldots, \boldsymbol{x}_{n_A+n_B}')$,

$$\mathbb{1}_{\hat{c}(U_n)-c(U_n)>t}\mathbb{P}'[\hat{c}'(U_n) - c(U_n) < t/2] \leq \mathbb{P}'[\hat{c}(U_n) - \hat{c}'(U_n) > t/2]$$

From Lemma D.1,

$$P'[\hat{c}(U_n) - c(U_n) \geq t/2] \leq 2\exp\left(-\frac{n_A t^2}{2} \cdot \frac{\gamma \bar{q}^2}{\left(4 + \sqrt{\gamma}\right)^2}\right)$$

$$\leq \frac{1}{2}$$

by the condition on $n_A \cdot t^2$. Hence:

$$\mathbb{1}_{\hat{c}(U_n)-c(U_n)>t} \leq 2\mathbb{P}'[\hat{c}(h_n) - \hat{c}'(h_n) > t/2],$$

and taking the expectation over the original sample $(\boldsymbol{x}_1, \ldots, \boldsymbol{x}_{n_A+n_B})$ finishes the proof. $\qquad\square$

Recall that the (deterministic) neighborhood oracle $n$ is fixed ahead of training and thus does not depend on the random sample(s). For $\boldsymbol{x}_i \in \mathcal{S}_A$, and $\boldsymbol{x}'_i \in \mathcal{S}'_A$, we can redefine $\boldsymbol{x}_i \leftarrow n(\boldsymbol{x}_i)$—that is, we apply the neighborhood oracle as part of the sampling process for points in $A$. Let $m = n_A + n_B$ and define $\mathcal{M}_{\boldsymbol{x}_1,\ldots,\boldsymbol{x}_m} = \{(\mathbb{1}_{\boldsymbol{x}_1 \in U}, \ldots \mathbb{1}_{\boldsymbol{x}_n \in U}) : U \in \mathcal{M}\}$. Recall that the *growth function* of class $\mathcal{M}$ is defined as $\mathcal{S}_\mathcal{M}(m) = \sup_{(\boldsymbol{x}_1,\ldots,\boldsymbol{x}_m)} |\mathcal{M}_{\boldsymbol{x}_1,\ldots,\boldsymbol{x}_m}|$. Now to finish the proof, observe that the sup in the right-hand-side of the Lemma D.2 result only depends on the *finite* set of vectors $\mathcal{M}_{\boldsymbol{x}_1,\ldots,\boldsymbol{x}_m,\boldsymbol{x}'_1,\ldots,\boldsymbol{x}'_m}$, due to our simplification $\boldsymbol{x}_i \leftarrow n(\boldsymbol{x}_i)$ for $\boldsymbol{x}_i \in \mathcal{S}_A, \mathcal{S}'_A$. That is,

$$\mathbb{P}\left[\sup_{U \in \mathcal{M}} \hat{c}(U) - c(U) > t\right] \leq 2\mathbb{P}\left[\sup_{U \in \mathcal{M}_{\boldsymbol{x}_1,\ldots,\boldsymbol{x}_m,\boldsymbol{x}'_1,\ldots,\boldsymbol{x}'_m}} \hat{c}(U) - \hat{c}'(U) > t/2\right]$$

$$\leq \sum_{U \in \mathcal{M}_{\boldsymbol{x}_1,\ldots,\boldsymbol{x}_m,\boldsymbol{x}'_1,\ldots,\boldsymbol{x}'_m}} \mathbb{P}[\hat{c}(U) - \hat{c}'(U) > t/2]$$

$$\leq 2 \sum_{U \in \mathcal{M}_{\boldsymbol{x}_1,\ldots,\boldsymbol{x}_m,\boldsymbol{x}'_1,\ldots,\boldsymbol{x}'_m}} \mathbb{P}[|\hat{c}(U) - c(U)| > t/4]$$

$$\leq 8\mathcal{S}_\mathcal{H}(2m) \exp\left(-n_A \cdot t^2 \frac{\gamma \bar{q}^2}{16(4 + \sqrt{\gamma})^2}\right).$$

The first line simply applies the union bound over $\mathcal{M}_{\boldsymbol{x}_1,\ldots,\boldsymbol{x}_m,\boldsymbol{x}'_1,\ldots,\boldsymbol{x}'_m}$. The second line uses the fact that if $a - b > c$ then for any real $x$, either $|a - x|$ or $|b - x|$ or both are larger than $c/2$, applies a union bound, and uses that $\hat{c}(U)$ and $\hat{c}'(U)$ are identically distributed. The last line applies Corollary D.1 and uses the definition of the growth function. The Sauer-Shelah lemma [61, 56, 57] implies that for any class $\mathcal{H}$ with $\mathrm{VC}(\mathcal{H}) = d$, $\mathcal{S}_\mathcal{H}(m) \leq \left(\frac{em}{d}\right)^d$. Then setting:

$$t \geq 4(4 + \sqrt{\gamma})\sqrt{\frac{\mathrm{VC}(\mathcal{M})\log\frac{2em}{\mathrm{VC}(\mathcal{M})} + \log\frac{8}{\delta}}{n_A \bar{q}^2}}$$

guarantees that

$$\mathbb{P}\left[\sup_{U \in \mathcal{M}} \hat{c}(U) - c(U) > t\right] \leq \delta$$

and completes the proof. $\qquad\square$

The following proposition shows the complexity of the class of mistake sets generated by $f \in \mathcal{F}$ is bounded by complexity of the hypothesis class $\mathcal{F}$ itself.

**Proposition D.1.** *Suppose $\mathcal{F}$ is a binary hypothesis class and $y : \mathcal{X} \to \{0, 1\}$ is the ground-truth classifier (we do not assume $y \in \mathcal{F}$). Fix a set $A \subset \mathcal{X}$ and for each $f \in \mathcal{F}$ let $U(A, f, y) = \{\boldsymbol{x} \in A : f(\boldsymbol{x}) \neq y(\boldsymbol{x})\}$ be $f$'s mistakes on $A$. Define $\mathcal{M} = \{U(A, f, y) : f \in \mathcal{F}\}$. Then $\mathrm{VC}(\mathcal{M}) \leq \mathrm{VC}(\mathcal{F})$.*

*Proof.* Fix a finite set of points $V \subset A$ with $|V| = d$, so $V = (\boldsymbol{x}_1, \ldots, \boldsymbol{x}_d)$. Suppose that $\mathcal{M}$ shatters $V$, so:

$$|\{U \cap V : U \in \mathcal{M}\}| = 2^d$$

Fix an arbitrary labeling $y' : \mathcal{X} \to \{0, 1\}$. Let $W = \{\boldsymbol{x}_i \in V : y'(\boldsymbol{x}_i) \neq y(\boldsymbol{x}_i)\}$. Since $W \in \mathcal{M}$, by definition of $\mathcal{M}$ there exists $f \in \mathcal{F}$ such that $f(\boldsymbol{x}_i) \neq y(\boldsymbol{x}_i)$ if and only if $\boldsymbol{x}_i \in W$. If $\boldsymbol{x}_i \in W$, then $f(\boldsymbol{x}_i) \neq y(\boldsymbol{x}_i)$ and $y'(\boldsymbol{x}_i) \neq y(\boldsymbol{x}_i)$; since there are only two labels, it must be the case that $f(\boldsymbol{x}_i) = y'(\boldsymbol{x}_i)$. If $\boldsymbol{x}_i \notin W$, then $f(\boldsymbol{x}_i) = y(\boldsymbol{x}_i) = y'(\boldsymbol{x}_i)$. This shows that for any labeling $y'$ of the points in $V$, there exists $f \in \mathcal{F}$ that has zero error on that labeling. Hence any set shattered by $\mathcal{M}$ is shattered by $\mathcal{F}$, so $\mathrm{VC}(\mathcal{M}) \leq \mathrm{VC}(\mathcal{F})$. $\qquad\square$

# E   Experiment Details

**Dataset details.** We train on the IMDb dataset of movie reviews [41] (HuggingFaceHub ID `stanfordnlp/imdb`), which has 25000 training examples and 25000 test examples, each with exactly 50/50 positive/negative split, and an unspecified license. For the teacher model described in Section 6, based on the presence of unigrams 'horrible' and 'incredible', the coverage rate is $\mathbb{P}(\mathbf{x} \in S) = 0.06$. The fully-supervised (gold) error of a linear classifier trained on top of the SentenceBERT representation we use is 10.6% (when trained using the optimization procedure and hyperparameters described below; we did not perform hyperparameter tuning, since this number is merely used as an ideal lower bound for the performance of a weakly-supervised classifier). For $i \in \{0, 1\}$ and points $\boldsymbol{x} \in S_i^{good}$, we obtained neighbors in $T_i$ and $S_i^{bad}$ using the oracle procedure described in Section 6 and shown graphically in Figure 3. In each neighbor sampling step, we enforce the constraint $y(\boldsymbol{x}') = i$ using the model trained on the gold labels to perform rejection sampling. This ensures (as much as possible) that all neighbors have the same ground-truth label, as required by the definitions. We include these sampled neighbors in the supplementary material for reproducability.

**Model training.** We train the linear classifiers using the AdamW optimizer [40] with global learning rate 0.01 and a weight decay of 0.1, and linear learning rate decay over 500 optimizer steps. For the heuristic in Section 5, we retrain a model 5 times on 5 different random subsets of the covered training samples, each 80% of the original, and use the other 20% of covered samples as a validation set to perform early stopping with the *weak label*. We do *not* perform early stopping with the true gold label, which is common practice in many works on programmatic weak supervision and gives a large performance gain [73, 10]. Since the goal of our experiments is to give evidence that expansion holds in practice, we do not tune the learning rate or weight decay parameters at all, since the initial values we chose already led to weak-to-strong generalization effects.

**Compute.** We used an internal machine with 4xA100 80GB GPUs to extract all deep network embeddings and to train the linear classifiers on top of those embeddings. Embeddings were only extracted once. Classifiers were trained 5 times with different random seeds and data subsets to perform the heuristic procedure in Section 5. This step was repeated several times as we developed different versions of our bounds and recomputed different notions of expansion w.r.t. the mistake sets of the trained classifiers.

**Results.** Table 3 shows the results of training using the SentenceBERT representations on the pseudolabels described in Section 6. The table also shows the accuracy of the weak labels themselves in the $\tilde{y}(\boldsymbol{x})$ row. Pseudolabel correction and coverage expansion occur in different amounts depending on the class. For example, the student consistently improves over $\tilde{y}$ on $S_0$ but not on $S_1$. This justifies our choice to analyze these effects separately for different classes. As discussed in Section 6, Tables 1 and 4 show the measured values of expansion, the worst-case weak label error of the student across the 5 training runs, and the values of our bounds from Theorems B.1 and B.2, respectively. Since the goal is to give evidence for the presence of expansion, we ignore $\mathbb{P}(R_\eta(f)|\cdot)$ when computing the values of the bounds and focus on the terms involving amount of expansion. The bounds in Table 1 show that our results can effectively distinguish between cases where pseudolabel correction does (on $S_0$) and does not (on $S_1$) occur.

Table 4 measures the expansion of the set families $\mathcal{M}(T_i, \mathcal{F})$ and $\mathcal{M}'(T_i, \mathcal{F})$ on the set pairs $(S_i^{good}, T_i)$ and $(S_i^{bad}, T_i)$, respectively. First, the fact that both expansion numbers are nonzero gives evidence for the extra structure described in Section 4.1 and assumed in Theorem 4.2/Theorem B.2. As mentioned in Section 4.1, we could have also just assumed expansion on $(S_i^{good}, T_i)$, but this extra structure gives a tighter bound, and these results suggest that it is present empirically. Table 4 shows that even though the coverage $\mathbb{P}(\boldsymbol{x} \in S) = 0.06$ is very low in this example, good representations (in this case, representations such that $\mathcal{M}(\cdot, \mathcal{G} \circ \varphi)$ expands for $\mathcal{G}$ linear), can have good coverage expansion. We use the tighter bound from Theorem B.2, which allows for different amounts of expansion between $(S_i^{good}, T_i)$ and $(S_i^{bad}, T_i)$.

Table 3: Test accuracy breakdown for linear probe trained with true (gold) and weak labels on the IMDb data. Performance of the weakly-trained model is broken down across the covered sets ($S_0$, $S_1$) and the uncovered sets ($T_0$, $T_1$). One standard deviation across five training folds is shown in parentheses. Pseudolabel correction and coverage expansion occur in different amounts depending on the class. For example, the student consistently improves over $\tilde{y}$ on $S_0$ but not on $S_1$. This justifies our choice to analyze these effects separately for different classes.

| Model | Test Acc. (gold train) | Test Acc. (weak train) | $S_0$ (4%) | $S_1$ (2%) | $T_0$ (46%) | $T_1$ (48%) |
|---|---|---|---|---|---|---|
| $\tilde{y}(\boldsymbol{x})$ | | | 89.5 | 67.1 | n/a | n/a |
| BoW | 84.9 | 50.1 | 89.5 | 67.1 | 100.0 | 0.0 |
| SentenceBERT | 89.4 | 81.8 (1.4) | 95.5 (1.4) | 69.1 (4.6) | 85.8 (3.1) | 74.9 (4.8) |

Table 4: Measured expansion values and error bounds for the uncovered sets $T_i$. Expansion values for the families $\mathcal{M}(S_i^{bad}, \mathcal{F})$ on $(S_i^{good}, S_i^{bad})$ and $\mathcal{M}'(S_i^{good}, \mathcal{F})$ on $(S_i^{bad}, S_i^{good})$, are measured using the heuristic described in Section 5. The detection of *both* types of expansion (expansion from $T_i$ to $S_i^{good}$ *and* to $S_i^{bad}$) gives evidence for the extra structure we described in Section 4.1 and justifies our use of Theorem B.2, which uses this structure, instead of Theorem B.3, which only uses expansion from $T_i$ to $S_i^{good}$ and gives a looser bound. Worst-case value of the error bound in Theorem B.2, computed using the smallest expansion values and largest weak errors $\text{err}(f, \tilde{y}|S_i)$ from the 5 training runs. The $\text{err}(f, \tilde{y}|S_i)$ and $\alpha_i$ values used in the bound computation are identical to the values in Table 1. Unlike in Table 1, where the "baseline" for pseudolabel correction effects is to have error bounds strictly better than $\alpha_i$, for coverage expansion, the more relevant comparison is against random/arbitrary guessing. The actual worst-case error of the student on each $T_i$ is shown as $\text{err}(f, \tilde{y}|T_i)$. As suggested by our bound values, the errors on each $T_i$ are non-trivial (much better than random or arbitrary guessing).

| Model | $i$ | $(S_i^{good}, T_i)$ | $(S_i^{bad}, T_i)$ | $\text{err}(f, \tilde{y}|S_i)$ | Bd. val | $\text{err}(f, y|T_i)$ |
|---|---|---|---|---|---|---|
| SentenceBERT | 0 | 0.16 | 0.98 | 0.12 | 0.37 | 0.16 |
| | 1 | 0.75 | 0.55 | 0.29 | 0.33 | 0.29 |

Figure 3: Example of our neighborhood oracle $n$, constructed using a targeted paraphrase procedure. For a covered point $\boldsymbol{x} \in S$ (in this case, $\boldsymbol{x} \in S_0^{bad}$, since it is a true negative point mislabeled by our example weak rules $\tilde{y}$), we first generate an uncovered point $\boldsymbol{x}' \in T_i$ using a constrained paraphrase model and rejection sampling to ensure the ground-truth label remains negative (we use a model trained on the gold labels as a stand-in for the ground truth $y$). Next, we use GPT-4 to rewrite $\boldsymbol{x}'$ using the *opposite* word from {horrible, incredible} than the one that originally appeared. This generates another point $\boldsymbol{x}'' \in S_i$. Since we enforce that $\boldsymbol{x}$ and $\boldsymbol{x}''$ are covered by different words, we know that if $\boldsymbol{x} \in S_i^{good}$ (resp. $S_i^{bad}$), $\boldsymbol{x}'' \in S_i^{bad}$ (resp. $S_i^{good}$).

