# OpenReview forum: "Theoretical Analysis of Weak-to-Strong Generalization"
_NeurIPS.cc/2024/Conference — NeurIPS 2024 poster_

### Official Review · Reviewer_cokD · 2024-07-09

**Soundness:** 3
**Presentation:** 3
**Contribution:** 3
**Rating:** 7
**Confidence:** 3

**Summary:**

The paper provides a theoretical analysis of weak-to-strong generalization, a phenomenon where strong models can learn from weak supervisors and outperform them [[Burns et al 2023](https://arxiv.org/abs/2312.09390)].

The authors make precise assumptions about the nature of the strong student model family and the weak supervisor mistakes. Specifically, (1) near any datapoint (where neighborhoods are defined according to the strong student model class) that is incorrectly labeled by the weak supervisor, there should be many correctly labeled datapoints and (2) datapoints that are unlabeled (i.e. in the test set) also have a correct number of correctly and incorrectly labeled datapoints in the neighborhood. The authors make these ideas mathematically precise, and derive generalization bounds for empirical risk minimization for the strong student model.

Importantly these bounds describe two phenomena: (1) pseudolabel correction, i.e. correcting for the mistakes of the weak supervisor and (2) coverage expansion, i.e. generalization to unlabeled datapoints.

Finally, the authors provide a way of statistically testing their assumptions empirically, and show that they are applicable in one setting: sentiment analysis with a bag-of-words supervisor.

**Strengths:**

S1: The theoretical approach used by the authors makes sense intuitively. The authors often provide an informal intuition for the definitions and results that they derive.

S2: The derived generalization bounds are to the best of my understanding novel and non-trivial. The authors provide detailed comparisons to existing generalization bounds for weak supervision.

S3: The authors provide a rigorous statistical method, as well as a heuristic for testing their assumptions in practice.

S4: The authors evaluate their assumptions and bounds in a simple empirical setting.

**Weaknesses:**

W1: While the core of the paper makes sense intuitively, the paper becomes more mathematically dense and hard to follow towards the end.

W2: It is not obvious to me that the paper provides a novel empirical insight. As authors admit in the limitations section, they do not provide a new training method with improved weak-to-strong generalization or generally make practical recommendations. However, the paper makes a valuable contribution in formally describing conditions under which we can provably get weak-to-strong generalization.

W3: The empirical evaluation is limited, and only covers one simple case. It would be very interesting to apply the bound to several settings, with varying weak-to-strong generalization. For example, can the bound predict the result in [[Burns et al 2023](https://arxiv.org/abs/2312.09390)] that weak-to-strong generalization doesn't work well on the reward modeling task? Does the bound provide an intuition for why that would be the case?

**Questions:**

Q1. See questions in W3.

Q2. Which of the assumptions specifically captures the intuitive notion that the strong student should not be able to fit the mistakes of the weak model too well?

**Limitations:**

The limitations are adequately discussed.

---

> ### Author Rebuttal · Authors · 2024-08-06
>
> > __For example, can the bound predict the result in [Burns et al 2023] that weak-to-strong generalization doesn't work well on the reward modeling task?__
>
> Great question! Our analysis is limited to classification for now, since we essentially assume that the class-conditional sets $\mathcal{X}_i$ have some neighborhood structure that is respected by the hypothesis class of the strong model. This type of assumption makes the most sense when $\mathcal{X}_i$ has some consistent semantic meaning. In the reward modeling case, the labels (chosen/rejected) are not semantically meaningful in the same way, so different structural assumptions are likely required. While our bounds don't explain the lack of weak-to-strong generalization in reward modeling, we are encouraged that they seem to plausibly capture these effects in classification problems. We will add reward modeling / ranking losses as an interesting direction for future work!
>
> > __Which of the assumptions specifically captures the intuitive notion that the strong student should not be able to fit the mistakes of the weak model too well?__
> >
> > [Related question from Reviewer 3HaP] __What happens in the case of self-training when the function class of strong and weak models are the same?__
>
> If the strong model class can exactly fit the weak model, the expansion assumption is not satisfied, as we now show.
>
> Suppose we have a weak model $\tilde{y} \in \mathcal{F}$ and aim to fit a strong model $f \in \mathcal{F}$. In this case we can have $f = \tilde{y}$ (i.e., the strong model exactly fits the weak model), so suppose that is the case. The class of sets we need to expand for pseudolabel correction is $\mathcal{M}’ = \\{R(g) \cap S_i^{good} \setminus \text{mistakes}(g) : g \in \mathcal{F}\\}.$
>
> Since $f = \tilde{y}$, $S_i^{good} \setminus \text{mistakes}(f) = S_i^{good}$. Now consider a pair $(x,x')$ with $x\in S_i^{good}$, $x' \in \mathcal{N}(x)\cap S_i^{bad}$. Since $x$ and $x'$ are both in $S_i$, $y(x) = y(x')$. Then we have $f(x) = \tilde{y}(x) = y(x) = y(x') \ne \tilde{y}(x') = f(x')$, so $f(x) \ne f(x')$. Thus $x$ is not in $R(f)$. Since $x$ was an arbitrary point in $S_i^{good}$, this shows $R(f) \cap S_i^{good} = \emptyset$, so $\mathcal{M}’$ contains $\emptyset$. In this case, expansion requires: $0 = P(\mathcal{N}(\emptyset) | S_i^{bad}) > c P(\emptyset | S_i^{good}) = 0,$ which is not possible.
>
> Working out this example hopefully gives more intuition for the expansion assumption, and we will include it as an example in the paper. Thanks for the question!

---

### Official Review · Reviewer_3HaP · 2024-07-13

**Soundness:** 3
**Presentation:** 2
**Contribution:** 3
**Rating:** 6
**Confidence:** 3

**Summary:**

The paper provides a theoretical explanation for weak-to-strong generalization. A weak model produces pseudolabels that can have errors and may not cover the entire input space. The paper argues current weak supervision theory fails to explain how and when the strong model can correct the psuedolabels (pseudolabel correction) and expand the coverage of pseudolabels (coverage expansion). The authors derive a new bound based on the expansion properties of the data distribution  and the hypothesis class of the strong model. Their bound suggests that generalization occurs when the strong model is unable to fit the mistakes of the weak model without incurring additional errors. They provide experiments to show the expansion properties are verifiable in practice.

**Strengths:**

1. The paper shows the gaps in the existing theory of learning from weak labels ( potentially erroneous labels available for part of the input space). In programmatic weak supervision (PWS) the focus has been on aggregating several weak labeling sources and showing that learning with aggregated labels is as good as learning from clean labels when the weak labeling sources satisfy certain conditions. The paper shows shortcomings of the previous results on PWS – failing to explain pseudolabel correction and coverage expansion.

2. It provides upper bounds on the expected error of the learned strong model on the part of the space that is covered by the psueodlabels and the part that is not covered. The first bounds suggests the error decreases in the covered part implying pseudolabel correction. A non-trivial error bound is provided on the error of the strong model in the uncovered set by utilizing the expansion property i.e. many points in the uncovered set have many correctly pseudo-labeled points.

3. The authors also provide a theoretical result suggesting that the expansion property is statistically checkable with finite samples but is computationally hard. They provide a heuristic approximation to verify the property in practice.

**Weaknesses:**

1. It is not clear from the bounds how the size of covered sets (pseudolabel coverage) affects the results and why are the results in terms of subsets of the covered set. Is it possible to provide a final result conditioned on the entire covered set?

2. The improvements in terms of coverage and error correction are not immediately clear. For instance, the following paper [1] explains the improvement in coverage with self-training in a specific setting. It would be useful to have a discussion and instantiation to some specific settings like in [1] to understand the results better.

[1] https://arxiv.org/abs/2006.11006

**Questions:**

1. How do the bounds depend on the size of the covered points and by how much does the coverage improve?

2. Where does the complexity of the strong model's function class come into play, how do the bounds depend on it? What happens in the case of self-training when the function class of strong and weak models are the same.

**Limitations:**

Yes.

---

> ### Author Rebuttal · Authors · 2024-08-06
>
> > __Is it possible to provide a final result conditioned on the entire covered set?__
>
> Yes, we just left it out for space and because getting a combined bound with a simple functional form requires more definitions (max/min of the weak label errors $\alpha_i$, the minimum expansion parameter across all the $S_i$'s, etc). Given these definitions, the combined bound essentially follows from averaging the bounds on each $S_i$. Similarly, we can give a combined coverage expansion bound conditioned on the entire uncovered set $T$ instead of the individual covered sets $T_i$. These two combined bounds can also be averaged together to get a final combined bound on the unconditioned error $err(f,y)$. We will include combined bounds in the Appendix.
>
> > __It is not clear from the bounds how the size of covered sets (pseudolabel coverage) affects the results__
>
> Thanks, this is a great point. We had a discussion on the role of coverage and had to cut it for space since it affects our bounds in a somewhat subtle way. The most direct way in which coverage enters the bounds would be in a combined error bound that uses
> $err(f,y) = err(f,y|S) P(coverage) + err(f,y | T) (1-P(coverage))$ and then applies the combined-source and combined-target bounds discussed in our reply to your previous question to $err(f,y|S)$ and $err(f,y|T)$, respectively. However, this doesn’t capture the full story.
>
> The “coverage rate” $P(S)$ also enters the picture implicitly in the $S$–$T$ expansion parameter. For a fixed neighborhood $\mathcal{N}$ (e.g., all points with similar embedding), it will be qualitatively easier to have expansion from $T_i$ to $S_i$ when $S_i$ is larger, since (informally) for each uncovered point in $T_i$, there are more possible covered neighbors in $S_i$. But increasing the coverage by including more points in $S_i$ might also affect the weak label accuracy parameters $\alpha_i$, which also affect the bounds.
>
> In practice, there is not a clear tradeoff between coverage and performance, as explored recently in, e.g., [37], so it qualitatively makes sense that our bounds do not prescribe a functional dependence of the error on the amount of coverage and instead allow that dependence to enter through data-dependent parameters (expansion $c$ and weak error rate conditioned on coverage $\alpha_i$). We hope this at least partially answers your question and we will include a more detailed discussion of the role of coverage in the final draft.
>
> > __[How do the bounds depend on] the complexity of the strong model's function class?__
>
> Great question! Following related work on expansion-based bounds (e.g., [23]), our error bounds are expressed as relationships between population quantities (e.g., expansion, population error on the weak/ground-truth labels). There is no statistical estimation aspect, which is where the complexity of the strong model's function class directly enters the picture. We focus on population quantities because relating the population error on the weak and ground-truth labels is the key problem-specific component. Once these quantities are related, the sample complexity aspect is more standard and can be dealt with by applying existing generalization bounds. We discuss this topic in more detail in Appendix B.5.
>
> The strong model hypothesis class can also enter the bounds indirectly via the expansion parameter, since the amount of expansion depends on that class. A richer class for the strong model may decrease the amount of expansion. For example, if the strong model class is rich enough to exactly fit the weak labels, there is zero expansion, as worked out in our response to Reviewer cokD. At the same time, the error of the strong model on the weak labels appears as a term in the bounds ($err(f,\tilde{y} | S_i)$), and a richer class might decrease this term. So these two terms capture a potential tradeoff—a stronger hypothesis class may decrease the expansion, but it may also decrease the error on the weak labels. Whether this makes the bounds tighter or looser depends on the sizes of these decreases. As with the coverage $P(S)$, our bounds do not prescribe a functional form for this dependence and instead allow it to enter via data-dependent parameters. This should make the bounds flexible enough to capture seemingly conflicting empirical results, where sometimes a stronger hypothesis class works better and sometimes a weaker one works better, as seen for example in the WRENCH weak supervision benchmarks [73].
>
> We will highlight this implicit dependence in Section 4. An interesting direction for future work is to see whether these tradeoffs can reveal how to pick a strong model hypothesis class for a given weak labeler, or pick a weak labeler for a given strong model hypothesis class. Thanks for the insightful question.
>
> > __It would be useful to have a discussion and instantiation to some specific settings like in [1] to understand the results better.__
>
> We wanted to include an instantiation of the results for special cases in the main text, but were limited by space. Appendix C.1 has a worked example under the distributional assumptions of co-training (where each data point consists of two conditionally independent views). We can mention this as a simple example where the expansion assumptions are satisfied with good values for $c$. There are other distributional assumptions that lead to good expansion, such as the Gaussian Mixture Model style distributions studied in Oymak and Cihad Gulcu---Wei et al. [69,  Example 3.4] showed that GMMs satisfy their expansion assumption, which is very related to ours. We will extend our existing discussion of Oymak and Cihad Gulcu in Appendix A to comment on its relationship to our assumptions.
>
> > __What happens in the case of self-training when the function class of strong and weak models are the same?__
>
> Please see our response to Reviewer cokD! We will include this in the paper as a worked example of a case where expansion does not hold.

---

> > ### Comment · Reviewer_3HaP · 2024-08-11
> >
> > Thanks for the response. I have no further questions and I'll keep my current scores.

---

> > > ### Author Response · Authors · 2024-08-12
> > >
> > > Thanks for reading our reply! We tried to address all of your questions and concerns (combined bounds, the role of coverage, the role of the strong hypothesis class, results for specific settings, what happens to the bounds when the strong model can exactly fit the weak labels). Please let us know if there is anything else we can do to improve your view of the paper!

---

### Official Review · Reviewer_KtpZ · 2024-07-14

**Soundness:** 3
**Presentation:** 3
**Contribution:** 4
**Rating:** 7
**Confidence:** 4

**Summary:**

This paper proposes a theoretical framework to interpret the weak-to-strong generalization phenomenon. It shows that strong student models trained on noisy labels from weak teacher models can outperform the weak teacher models, correcting their errors and generalizing well to examples where the weak teacher models are not confident or abstain. The paper demonstrates that existing bounds for learning from noisy labels do not explain this phenomenon and derives new bounds based on assumptions about the strong model’s robustness in its neighborhood and its expansion property. An empirical study validates the generalization bound in a practical setting.

**Strengths:**

- Weak-to-strong generalization has gained renewed attention despite being well-known in weak supervision. However, there has been no theoretical analysis of why a model trained with noisy labels can be more accurate than the labels themselves. This paper’s contribution is crucial as it provides a theoretical framework for this problem.
- The theory explains under what conditions weak-to-strong generalization can occur, which has practical implications.
- The paper is technically solid, with reasonable assumptions for derivation. Sections 4.2 and 5 add to its practicality.
- The writing is clear and the main ideas are easy to follow.

**Weaknesses:**

- The experiment is limited to only one setting, though it is already mentioned in the limitations section. This is a minor concern since the paper is primarily theoretical.

**Questions:**

- Adding comparisons with recent concurrent works ([1, 2, 3]) would be beneficial. I am curious about the authors' views on these papers and whether there are any conflicting points in theory or if the conclusions are well-aligned.
- Including simpler or synthetic experiments could enhance the paper. The (c, q) expansion property in a practical setting is not intuitively straightforward.

[1] Somerstep, Seamus, et al. "A statistical framework for weak-to-strong generalization." *arXiv preprint arXiv:2405.16236* (2024).

[2] Charikar, Moses, Chirag Pabbaraju, and Kirankumar Shiragur. "Quantifying the Gain in Weak-to-Strong Generalization." *arXiv preprint arXiv:2405.15116* (2024).

[3] Zhang, Edwin, et al. "Transcendence: Generative Models Can Outperform The Experts That Train Them." *arXiv preprint arXiv:2406.11741* (2024).

**Limitations:**

Authors adequately addressed the limitations in Conclusion.

---

> ### Author Rebuttal · Authors · 2024-08-06
>
> > __Adding comparisons with recent concurrent works ([1, 2, 3]) would be beneficial.__
>
> Thanks for pointing these out! First, we just want to note that all 3 of these works appeared on arXiv after the NeurIPS submission deadline, so we were not aware of them at the time of submission.
>
> Broadly, [1] and [2] are similar to our paper in that they make some structural assumptions under which weak-to-strong generalization can provably occur. [1] uses a framing of weak supervision as transfer learning to give pseudolabel correction results when the data is distributed according to a Gaussian Mixture Model, whereas we do not make specific distributional assumptions and also focus on coverage expansion in addition to pseudolabel correction. As discussed in our response to Reviewer 3HaP, Gaussian Mixture Models can be shown to satisfy notions of expansion. [2] is the most similar to our paper, but focuses on pseudolabel correction for regression problems, whereas our work tries to explain both coverage expansion and pseudolabel correction for classification problems—both phenomena are important in weak supervision settings that we are interested in. Additionally, the underlying assumptions and conceptual ideas are very different. In our work, we use structural notions of expansion to explain these two phenomena, and our bounds directly generalize existing results from other literature (co-training, self-training), and extend this theory to more general settings. In [2] they assume convexity of the hypothesis space for fine-tuning, and show that projections onto this space can explain pseudolabel correction effects. We feel these two explanations are very different but potentially complementary, and it would be very interesting future work to develop a common framework that incorporates both. [3] considers a qualitatively very different setting, showing that when the training data consists of a mixture of data generated by different experts, a strong student model can be better than the best single expert that generated the data. Their results are more related to classical work on crowdsourcing, where by observing generations from many experts the learner can outperform the best single expert. Our paper and [1] and [2] consider the case where there is a single weak teacher model (which itself may be an ensemble of many models, but this is not exposed to the learner).
>
>
> > __Including simpler or synthetic experiments could enhance the paper. The (c, q) expansion property in a practical setting is not intuitively straightforward.__
>
> Thanks for the suggestion! Per our responses to the other reviewers, we will include a worked example that shows (c,q)-expansion does not hold when the strong model can exactly fit the weak model’s errors. Subject to space limitations, we can also include a version of the content in Appendix C.1, which gives a very straightforward setting (“conditionally independent views”) where the expansion assumptions hold. These examples should give more intuition for how the assumption works.

---

> > ### Comment · Reviewer_KtpZ · 2024-08-09
> >
> > Thank you for your response! I was aware that [1, 2, 3] were published after the NeurIPS deadline, but I was curious to hear the authors' perspective, as these works seem to offer somewhat different interpretations of the same phenomenon. I thoroughly enjoyed reading this paper, and I will maintain my initial score, as I had already given it a high rating.

---

> > > ### Author Response · Authors · 2024-08-12
> > >
> > > Thanks for reading our reply and for the encouraging feedback!

---

### Author Rebuttal · Authors · 2024-08-06

Overall comments:
--

Thanks to all the reviewers for their time, effort, and helpful feedback. We are encouraged that all the reviewers found our work technically sound, novel, and potentially impactful. We have replied to individual points below. We hope the reviewers will consider raising their scores if our replies address their questions and concerns!

---

### Decision · Program_Chairs · 2024-09-25

**Decision:**

Accept (poster)

**Comment:**

This paper addresses some of the failures of current weak-to-strong generalization results to account for phenomena that occur in practice, namely correction of the teacher’s errors and generalization where the teacher is not confident. The reviewers found the writing clear and the contributions valuable. However, they called for illustrating the results with basic experiments, even if synthetic. Also, several concurrent manuscripts were pointed out, and comparison with them would help the community better appreciate the contributions.